# From mapped faults to Fault-Length Earthquake Magnitude (FLEM): A test on Italy with methodological implications

Fabio Trippetta[1], Patrizio Petricca[1], Andrea Billi[2], Cristiano Collettini[1,3], Marco Cuffaro[4], Anna Maria Lombardi[3], Davide Scrocca[4], Giancarlo Ventura[5], Andrea Morgante[5], and Carlo Doglioni[1,3]

[1]Dipartimento di Scienze della Terra, Sapienza Universitá di Roma, Rome, Italy
[2]Consiglio Nazionale delle Ricerche, Rome, Italy
[3]Istituto Nazionale di Geofisica e Vulcanologia, Rome, Italy
[4]Istituto di Geologia Ambientale e Geoingegneria, CNR, Rome, Italy
[5]Sogin (Società Gestione Impianti Nucleari) S.p.a., Rome, Italy

**Correspondence:** Andrea Billi (andrea.billi@cnr.it)

**Abstract.** Empirical scaling laws between fault/slip dimensions and earthquake magnitudes are often used to assess the maximum possible earthquake magnitude of a territory. In this paper, upon the assumption of the reactivability of any fault, these seismic scaling relationships are ~~benchmarked~~ compared at the national scale in Italy against catalogued magnitudes, considering ~~all known faults regardless of their~~ a comprehensive fault dataset regardless fault age, stress field orientation, strain rate, or else. Italy is a suitable case for comparing the fault-size-derived seismic magnitudes with the existing accurate catalogues of historical-instrumental earthquakes. To do so~~,~~: (1) a comprehensive catalogue of all known faults is compiled by merging the most complete databases available~~,~~; (2) the earthquakes magnitude (FLEM) is simply derived from fault length~~,~~; and (3) the resulting FLEMs are compared (i.e., the mathematical difference) with catalogued earthquake magnitudes. Results show that the largest FLEMs as well as the largest differences between FLEMs and catalogued magnitudes are observed for poorly constrained faults, mainly inferred from subsurface data. It is suggested that these areas have to be further characterized to better estimate fault dimension and segmentation and hence properly assess the FLEM. Where, in contrast, the knowledge of faults is geologically well constrained, the calculated FLEM is often consistent with the catalogued seismicity, with the $2\sigma$ value of the distribution of differences being 1.47 and reducing to 0.53 when considering only the M $\geq$ 6.5 earthquakes. ~~This overall consistency gives credibility to the used empirical scaling laws; however, some large differences between the two datasets suggest further validation of this experiment in Italy and elsewhere. The main advantages~~ The main advantage of this method is its independence from temporal and (paleo)seismological information, whereas the main novelty is its use at the national scale also for faults considered inactive. ~~Our work can provide a perspective time-independent seismic potential of faults; however, it cannot be a substitute for time-dependent (paleo)seismological methods for seismic hazard assessments.~~

## 1 Introduction

In some seismically active regions like California, information derived from geodesy, the geology of active faults, and seismology (both historical and instrumental) have been combined to develop a comprehensive method that predicts 99% of the

chances of having one or more M $\geq$ 6.7 earthquakes in the following 30 years (USGS Fact Sheet, 2008). Along the North Anatolian Fault (Turkey), fault geometry and slip accumulated during strong earthquakes have been used to infer the transfer of stress during seismic sequences and to estimate the increase of earthquake probability with M $\geq$ 6.7 (Stein, 1997; Parsons et al., 2000; Bohnhoff et al., 2016). These case studies involve areas characterized by high strain rates and affected by large

plate boundary faults that can be easily recognized in geological and/or geophysical records.

In contrast, other areas in the world where strain rates are low-to-intermediate and faults show smaller dimensions and unclear surface expression of recent activity, the connection between faults and earthquakes is not straightforward~~, particularly~~. This, in particular, applies to the relationship between potentially seismogenic faults and the maximum possible earthquake magnitude. Moreover, where lands were largely uninhabited during historical times (but are now densely populated and, there-

fore, potentially exposed at the seismic threat), the recurrence time of strong earthquakes is significantly longer with respect to the age of the seismological network~~, or~~. Further, in these areas information on historical and instrumental seismicity can be largely incomplete ~~,~~ and the assessment of the maximum possible earthquake magnitude can be difficult (Camelbeeck et al., 2007; Kafka, 2007; Stein and Mazzotti, 2007; Swafford et al., 2007; Dawson et al., 2008; Braun et al., 2009; Boyd et al., 2013; Leonard et al., 2014; Talwani, 2014; Campbell et al., 2015; Calais et al., 2016; Christophersen et al., 2017; Wang et al., 2019).

Where frequent earthquakes are absent and seismic history is unknown, the seismic potential and the temporal knowledge of seismic activity of faults can be determined using geological studies such as paleoseismological trenching or radiometric dating of slip indicators (Galli, 2000; Rockwell et al., 2000; Palumbo et al., 2004; Dixon et al., 2003; Galli et al., 2008; McCalpin, 2009; Sherlock et al., 2009; Nuriel et al., 2012; Viete et al., 2018)~~, but these studies~~. These studies however, are usually time-consuming and expensive. Alternatively, the maximum potential magnitude of earthquakes can be assessed using

empirical scaling laws of active faults, that is, using fault length and/or fault slip (Wells and Coppersmith, 1994; Pegler and Das, 1996; Mai and Beroza, 2000; Henry and Das, 2001; Liu-Zeng et al., 2005; Leonard, 2010; Thingbaijam et al., 2017). In this latter case, however, a lack of information on fault age and earthquake recurrence time may induce to neglect some faults as potential seismogenic sources. One method to overcome the problems connected with the lack of information on the age of (seismogenic) faulting may be to apply the existing seismic scaling relationships, which directly link fault/slip dimensions

and earthquake magnitudes (e.g., Wells and Coppersmith, 1994; Leonard, 2010; Thingbaijam et al., 2017), to the whole set of known faults including the (presumably-)inactive and undetermined faults. This method would rely on the long-held concept of fault reactivation (Sibson, 1985) due to the weakness of the fault surface compared with the host rock (Zoback et al., 1987; Collettini et al., 2009). This concept is particularly relevant when considering the future behaviour of faults over long terms. For nuclear plants, for instance, it is important to assess the occurrence of events, even if very unlikely, also in areas historically

free of damage, where normally these plants are built. In these sites (nuclear plants) indeed only a very low level of risk can be accepted. In other cases such as the geological disposals of radioactive wastes, the lifetimes to be considered are on the order of 105 years at least (up to 1 Ma according to prescriptions by the Nuclear Energy Agency and International Atomic Energy Agency, www.oecd-nea.org, www.iaea.com; NEA, 2004; IAEA, 2016). Accordingly, very long seismic return times must be considered, far exceeding the lifespan of historical seismic catalogues (and also paleoseismological records) normally used in

standard seismic hazard assessment practices.

For the richness of data (both tectonic and catalogued historical/instrumental-seismology; Fig. 1), Italy is a suitable country for deriving and comparing the potential maximum magnitude from earthquake data and from scaling relationships. Italy has, indeed, a long tradition of geological-structural mapping over the entire territory (since at least Murchison, 1849; von Zittel, 1869; Viola, 1893; Pagani, 1907, to the recent national geological maps available online at http://sgi.isprambiente.it/geoportal/) as well as a dense seismological network operating since at least the 1980s (available online at http://cnt.rm.ingv.it/ Iside Working Group, 2016), and . Moreover, Italy has a rare, if not unique, historic record of earthquakes (Rovida et al., 2016).

This work includes: (1) the composition of a comprehensive catalogue of all mapped faults in Italy, merging materials from previous documents; (2) the calculations of earthquake magnitudes (FLEM, Fault-Length Earthquake Magnitude) from the fault length in map view; (3) a comparison (i.e., the difference) of these calculations (FLEM) with catalogued seismological data (historical and instrumental earthquake magnitudes); and (4) a discussion of the agreements and differences between the calculated (FLEM) and catalogued magnitudes to determine the strengths and weaknesses of the proposed method.

We anticipate that, with this work, we do not intend to propose an alternative method for seismic hazard assessment or to better previous methods (e.g., Giardini, 1999; Jiménez et al., 2001; Michetti et al., 2005; Field et al., 2009, 2015; Reicherter et al., 2009). Our main aim is to test whether solely considering the known mapped faults (both active, inactive, and undetermined) and disregarding further information (e.g., historically- and instrumentally-recorded earthquakes as well as the regional stress field and strain rate) it is possible to provide, through existing seismic scaling relationships of faults and earthquakes, reasonable assessments of the maximum possible earthquake magnitude over an entire nation. The resulting (assessed) magnitudes (FLEMs) are compared (i.e., the mathematical difference) with catalogued earthquake magnitudes that are the only existing points of reference against which assessed magnitudes can be compared. Note that these results should be considered more in a theoretical and methodological perspective for comparison with future similar studies rather than in an applicative perspective for the case of Italy. In particular, our assessed earthquake magnitudes (FLEMs) for the Italian territory are proposed in this paper for scientific reasons and not for their use for civil protection and prevention purposes. Moreover, in this article, we do not consider or estimate at all the probability of earthquake occurrence. Yet, we would like to acknowledge that some large magnitudes of earthquakes (FLEMs) calculated in this article are considered very unlikely in the existing literature of seismic hazard in Italy (e.g., Cinti et al., 2004; Slejko et al., 2010).

## 2   Seismotectonic setting

The present setting of the Italian peninsula (Fig. 1) derives from the interaction (mostly convergence) of at least three main plates, namely, Eurasia to the north, Africa to the south, and Adria to the east and southeast. During Cenozoic-Quaternary times, this tectonic interaction has led to the formation of two major mountain chains - the Alps to the north and the Apennines along the peninsula - and to the opening of two major oceanic basins - the Ligurian-Provencal basin to the west of Sardinia and Corsica and the Tyrrhenian Sea basin between the Italian peninsula and the Sicily and Sardinia major islands (Dewey et al., 1989; Malinverno and Ryan, 1986; Doglioni, 1991; Doglioni et al., 1999; Faccenna et al., 2004; Rosenbaum and Lister, 2004; Carminati et al., 2010; Carminati and Doglioni, 2012).

The Alps and Apennines are characterized by very different tectonic styles. ~~Whereas the~~ The Alps show double-verging growth (northward and southward), with the involvement of large volumes of crystalline basement and the exhumation of metamorphic rocks (Nicolas et al., 1990; Schmid et al., 1996)~~, the~~ . The Apennines form a single-verging chain characterized by a radial vergence and a strong curvature from northwest to southeast. The Apennines are characterized by thin-skinned tectonics with rare exposures of the crystalline basement and metamorphic rocks (Barchi et al., 1998; Patacca et al., 2008; Scrocca et al., 2005). The development of the Apennines (contraction) occurred partly at the same time with the development of the Ligurian-Provencal and Tyrrhenian oceanic basins (extension). The different tectonic styles of Alps and Apennines are also well represented by the different settings of the related foreland monoclines and basins. The foreland basin is shallow in front of the Alps with a monocline dipping by 2-4° and deep in front of the Apennines with a monocline dipping by 6-15° (Mariotti and Doglioni, 2000).

The present seismotectonic setting of Italy is still mainly ruled by the interaction between the African and Eurasian plates, presently converging at a rate of c. 10 mm/yr along a NNW-SSE direction (DeMets et al., 1990; Serpelloni et al., 2005; Billi et al., 2011). This kinematic setting is complicated by the Adria plate (an independent microplate or a promontory of the African plate), which contributes to cause contractional deformations along the central-northern Adriatic margins (both toward the Apennines to the west and toward the Dinarides to the east) and in the Po Plain between the northern Apennines and the southern Alps. Contractional deformations are in contrast absent or poorly active in the southern Apennines where these deformations have been inactive or poorly active since about mid-Pleistocene times. In synthesis, for the aims of this work, it is important to know that most of the Italian territory has been subject to numerous deformation phases with related activations, re-activations, and/or inversions of faults. These phases are schematically amenable to one or two main complex tectonic mechanisms: the Alpine orogenesis (Paleozoic to present times) and the Hercinian orogenesis (Paleozoic times) (Vai, 2001). At present, within the Alpine orogenesis, back-arc extensional seismic mechanisms prevail in large areas of the Italian territory (e.g., the axial portion of the Apennines), whereas compressional seismic mechanisms are less frequent and prevail elsewhere (e.g., the eastern or Adriatic portion of the Apennines) (e.g., Basili and Meghraoui, 2001; Neri et al., 2002, 2005; Roberts and Michetti, 2004; Chiarabba et al., 2005, 2015; Palano et al., 2012; Presti et al., 2013; Ferranti et al., 2014; Cowie et al., 2017; Orecchio et al., 2017). Strike-slip tectonics is limited to a few major fault zones (e.g., Grasso and Reuther, 1988; Billi et al., 2003; Viganò et al., 2015; Polonia et al., 2016, 2017).

Italy has a widespread crustal seismicity (depth < 35 km) that concentrates along some portions of the Alps and mostly on the Apennines, including Calabria and Sicily (Fig. 1). Deep earthquakes are mainly located beneath the Calabrian Arc and, secondarily, beneath the northern Apennines and eastern Alps, where sparse seismicity is recorded (Chiarabba et al., 2005). The eastern and western Alps show a clustered crustal seismicity (Fig. 1)~~, whereas in~~ . In the central Alps, the number of recorded events seems to be less densely distributed. This is probably related to the lack of an appropriately-dense seismic network in this area (Amato, 2004; Chiarabba et al., 2005).

The western Alps are particularly active in their southern portion, where active N-S-striking faults accommodate most of the extensional and wrench deformations that characterize the present-day tectonics of the area (Chiarabba et al., 2005; Sanchez et al., 2010; Sue et al., 1999; Sue and Tricart, 2002). Focal mechanisms of the sparse seismicity in the central Alps show

extensional kinematics on N-S-striking fault planes (Chiarabba et al., 2005); however, stress field studies in this area suggest mainly active N-S compression (Montone and Mariucci, 2016). The clustered seismicity recorded in the eastern Alps is mainly located along E-W trending structures~~, showing~~. This seismicity shows an overall compressive tectonics generating large thrust earthquakes in response to a N-S trending compression, like the 1976 Friuli, Mw = 6.4 earthquake (Bressan et al., 1998;

Cheloni et al., 2012; Michetti et al., 2012).

The Apennines seismicity is densely distributed along the whole chain (Fig. 1). Focal mechanisms of the chain show a rotation of the fault strike from NW-trending alignments to NNE-trending ones moving from north to south, along the arc-like shape of the Apennines. In general, in the middle portion (axis) of the Apennines, earthquakes are characterized by extensional kinematics, whereas the Adriatic front is mainly affected by compressive kinematics, which is in agreement with the regional

stress field (Montone and Mariucci, 2016). In particular, the north-western portion of the Apennines shows a NW-SE trending cluster of seismicity. Moving eastward, beneath the Po Plain, seismicity involves the northern Apennines outer front (Fig. 1), where focal mechanisms are mainly compressive~~, highlighting~~. These mechanisms highlight E-W-striking fault planes consistent with the attitude of the main structures of the area. Here, the largest instrumentally measured earthquakes (Mw = 6.1) occurred in 2012 during the Emilia (N-Apennines) sequence (Govoni et al., 2014). Moving southward, crustal seismicity

(M 4-6.5) is densely distributed and follows the Apennine chain axis. In this zone (Umbria-Marche), the largest instrumental earthquakes with extensional focal mechanisms occurred in the 1997 Colfiorito, Mw = 6.0 (Amato et al., 1998); 2016 Norcia, Mw = 6.5 (Chiaraluce et al., 2017); 2016 Amatrice, Mw = 6.0 (Chiaraluce et al., 2017); 2009 L'Aquila (Abruzzi), Mw = 6.3 (Chiarabba et al., 2009); and 1980 Irpinia (Campania), Mw = 6.9 (Nostro et al., 1997) earthquakes. A cluster of moderate (M 4.0-5.0) seismicity is recognized close to the Tyrrhenian coast in relation to active geothermal and volcanic districts (Gasparini

et al., 1985). Larger earthquakes characterize the Apennines southern portion (Calabria), with historical seismic events that reached magnitudes up to $\simeq$7.1 (Rovida et al., 2016; Guidoboni et al., 2018).

The Calabrian Arc (Fig. 1) is mostly characterized by deep instrumental seismicity related to the NW dipping Benioff plane (Polonia et al., 2016; Selvaggi and Chiarabba, 1995). Both the Etna area and the northern on-shore portion of Sicily show clustered seismicity, where the latter is mainly characterized by extensional and oblique-extensional earthquakes (Azzaro

and Barbano, 2000; Billi et al., 2006a). The strongest earthquakes (some of them with associated destructive tsunamis; Billi et al., 2010) from historical catalogues in the area occurred predominantly on extensional faults and are: the 1908 (Mw 7.2) earthquake of Messina and Reggio Calabria (Messina Straits, Sicily), the 1638 and 1905 (Mw 7.0) earthquakes of Nicastro (Calabria), and the 1693 (Mw 7.4) earthquake of eastern Sicily (Rovida et al., 2016).

The Southern Tyrrhenian (Fig. 1) is characterized by intense seismicity, with magnitudes up to 7.1, which occurred in 1938

at 290 km depth (Selvaggi and Chiarabba, 1995). The seismicity of shallow levels (< 30 km) below the southern Tyrrhenian Sea indicates the presence of two adjacent domains characterized by different tectonic environments: (1) to the northwest of the Aeolian Islands, a N-S compressive tectonics is present, whereas (2) to the east and southeast of these same islands, a NW-SE extensional tectonics occurs (Fig. 2) (Goes et al., 2004; Pondrelli et al., 2004; Billi et al., 2006a; Cuffaro et al., 2010). Intermediate and deep seismicity concentrates along a roughly uninterrupted, narrow, and steep (70°) Benioff zone, which

strikes SW-NE, dips toward NW, and reaches a depth of about 500 km~~; only~~. Only one earthquake occurred inland at depth of

350 km (Pondrelli et al., 2004; Selvaggi and Chiarabba, 1995). Shallow compressive seismicity (< 30 km deep) is characterized by epicentres mainly aligned E-W (Pondrelli et al., 2004; Presti et al., 2013), off-shore northern Sicily, and it rotates . These epicentres align roughly NNW-SSE moving eastward (Fig. 1). Sparse compressive events have also been recorded off-shore north-eastern and southern Sardinia.

## 3   Input data

### 3.1   Fault data

To build a comprehensive dataset of faults in Italy (Figs. 1-3; supplement; Petricca et al., 2018), the following databases were merged: (1) the entire fault collection of the Italian Geological Maps at the 1:100,000 scale (i.e., Carta Geologica d'Italia available online at www.isprambiente.it); (2) the fault compilation from the Structural Model of Italy at the 1:500,000 scale (Bigi et al., 1989); (3) all faults provided in the ITHACA-Italian Catalogue of Capable Faults (Michetti et al., 2000); and (4) the inventory of active faults from the GNDT (Gruppo Nazionale per la Difesa dai Terremoti; Galadini et al., 2000) database. The strength point of our approach is the assemblage of different fault datasets heterogeneously built for different purposes and based on different primary information and methods. In this approach, we consider all known faults (see above)to form a dataset as comprehensive as possibleAlthough different, the common point of all used datasets is that they have faults mapped and therefore measurable over the Earth's surface. Geological studies and related mapping are ongoing in Italy and, frequently, new geological-structural maps are produced and published. We considered these further maps only for those areas that are, in our opinion, poorly covered by the main four databases mentioned above (see below for further explanations).

Faults from the 1:100,000 Italian Geological Maps and 1:500,000 Structural Model of Italy (black and black dashed lines, respectively, in Fig. 2) are essentially based on field surveys integrated with subsurface geophysical data and, therefore, were drawn and constrained by geological and geophysical observations.

The ITHACA-Italian Catalogue of Capable Faults (Michetti et al., 2000) is a database developed by the ISPRA (Istituto Superiore per la Protezione e la Ricerca Ambientale) containing cartographic and parametric information of active faults - i.e., faults with evidence of repeated reactivation during the last 40,000 years - capable of rupturing the ground surface in Italy. The database is available as a layer in a web GIS (http://www.isprambiente.gov.it/it/progetti/suolo-e-territorio-1/ithaca-catalogo-delle-faglie-capaci) and contains both the geographic location and a text description of each fault. The entire set of capable faults is included in this compilation (blue lines in Fig. 2).

The inventory of active faults of the GNDT database (Galadini et al., 2000) represents a collection of the Italian active faults. The activity of these faults is mainly deduced through surface geological evidence and (paleo)seismological data including historical information (red lines in Fig. 2).

To improve and implement these fault databases, we selected published complementary studies for some specific areas considered to not be exhaustively covered by the aforementioned collection of faults including Sardinia, SW Alps, Tuscany, the Adriatic front, Puglia, and the Calabrian Arc. For these areas, we selected faults on the grounds of scientific contributions that documented faults the fault presence based on field, seismic, and paleoseismological data (pink lines in Fig. 2). In particular,

for the Campidano area (southern Sardinia), we used the fault pattern proposed by Casula et al. (2001), who reconstructed fault geometry with recent tectonic activity based on field and seismic reflection profiles. For the SW Alps, we followed the works of Augliera et al. (1994), Courboulex et al. (1998), Larroque et al. (2001), Christophe et al. (2012), Sue et al. (2007), Capponi et al. (2009), Turino et al. (2009) and Sanchez et al. (2010). For the Tuscany area, we consulted Brogi et al. (2003), Brogi et al. (2005), Brogi (2006), Brogi (2008), Brogi (2011) and Brogi and Fabbrini (2009). For the buried northern Apennines and Adriatic front, we used the fault datasets provided by Scrocca (2006), Cuffaro et al. (2010) and Fantoni and Franciosi (2010). For the Puglia region, we used data from Patacca and Scandone (2004) and Del Gaudio et al. (2007), whereas, for the Calabrian Arc, we used data from Polonia et al. (2016) and Polonia et al. (2017).

Furthermore, we are aware of the DISS compilation of seismogenic sources in Italy (Basili et al., 2008; DISS Working Group, 2018) as one of the most important integrated datasets of the Italian territory in this field. However, we did not use this dataset in this work since, as described in Basili et al. (2008), the dataset aims to identify the "seismogenic sources" rather than the actual faults. A composite seismogenic source (CSS in the DISS nomenclature) is a complex fault system showing homogeneous kinematic and geometric parameters and contains an unspecified number of aligned possible ruptures (i.e., faults) that cannot be isolated. Hence, a CSS, for its own nature, cannot be considered a real fault as declared in our scope.

The above-mentioned fault datasets form a ~~comprehensive~~ tectonic image of the Italian territory (Fig. 2). These fault data and the spatial grid are available for download as ASCII files in the supplement (see also Petricca et al., 2018). The major thrust faults occur along: (1) the N-verging western and north-western external front of the northern Alps; (2) the S-verging frontal ramp of the southern Alps (Po Plain and Veneto-Friuli regions); (3) the E- to N-verging external front of the central-northern Apennines from the Po Plain down to the central Adriatic off-shore; (4) the SE-verging outer front of the Calabrian Arc; (5) the outer (southern) front of the Maghrebian-Apennines chain in Sicily; and (6) the E-W-trending contractional belt located in the southern Tyrrhenian Sea, close to the northern Sicily coast.

The major normal faults occur along the median zone of the Apennines fold-thrust belt (Fig. 2), namely: (1) in northern Tuscany; (2) in central Italy including the Tuscany, Umbria, Marche, and Abruzzi regions; (3) in the southern Apennines including the Molise, Campania, Basilicata, and Calabria regions; and (4) in eastern Sicily including the Messina Straits, part of the Ionian Sea areas, and the Hyblean foreland. In particular, SW-dipping, high-angle normal faults (NW-striking) host the strongest seismicity recorded along the northern-central Apennines belt (Figs. 1 and 3). This fault pattern rotates to a NE-trending direction toward the southern Calabria region consistently with the focal solutions of the area (Fig. 1). Moreover, extensional faults have also been mapped in the Sardinian region, specifically in the Campidano graben, mainly based on subsurface data (Casula et al., 2001).

Strike-slip faults are located in some areas of the Italian territory (e.g., Billi et al., 2003, 2007), in particular: (1) in the southern Alps (Veneto) with NNW-striking structures; (2) across the external front of the central-southern Apennines foreland-fold-thrust belt (e.g., south of the Montemurro area, Fig. 1); (3) along the central Adriatic-Gargano-Molise belt with E-W-striking structures; (4) across the Calabrian Arc with radial structures cutting through the accretionary wedge; (5) in eastern Sicily from the Aeolian Islands (Tyrrhenian Sea) and southward into the Ionian Sea; and (6) in south-western Sicily and in the Sicily Channel with structures striking between N-S and NNE-SSW (Figs. 1 and 2).

## 3.2 Earthquake data

To obtain a complete earthquake dataset for the Italian territory (Figs. 1 and 3; supplement; Petricca et al., 2018), we integrated the existing most comprehensive catalogues of instrumental and historical seismicity: (1) the CSI1.1 instrumental database (csi.rm.ingv.it; Castello et al., 2006) for the 1981–2002 period; (2) the ISIDe instrumental database (iside.rm.ingv.it; Iside Working Group, 2016) for the 2003–2017 period; and (3) the CPTI15 historical-instrumental database (emidius.mi.ingv.it; Rovida et al., 2016) for the 1000-1981 period.

The CSI 1.1 database (Castello et al., 2006) is a catalogue of Italian relocated earthquakes for the 1997–2002 period. This collection derives from the work by Chiarabba et al. (2005), who relocated, using a homogeneous procedure, approximately 45,000 events provided by several seismological networks (both national, regional, and local) operating in the Italian territory. Most seismic events are lower than 4.0 in magnitude and are mostly located in the upper 12 km of the crust. A few earthquakes exceed magnitude 5.0, whereas the largest event is Mw 6.0. The time-span of this compilation is 1981–2002. From the CSI 1.1 database, we selected events with Mw > 4.0 (Fig. 3).

The ISIDe database (Iside Working Group, 2016) provides the parameters of earthquakes from real-time recordings and from the Italian Seismic Bulletin. The main aim of this database is to supply information on the seismicity as soon as it becomes available by integrating it with updated information on past seismicity. The time-span of this compilation begins in 1985 and lasts up to the present day. To avoid an overlap with the CSI database (1981–2002), from the ISIDe database, we considered events with Mw > 4.0 only from the 2003–2017 period (Fig. 3).

The CPTI15 historic database integrates the Italian macroseismic database version 2015 (DBMI15; Locati et al., 2016) and instrumental data from 26 different catalogues, databases, and regional studies (including the CSI and ISIDe databases) for the Italian territory, starting from 1000 A.D. until 2014. This catalogue provides moment magnitudes from macroseismic determination for more than 3200 earthquakes with values in the 4-7 range of Mw. To avoid any overlapping of data with the aforementioned instrumental datasets, we used data from the 1000-1981 period from the CPTI2015 catalogue.

We acknowledge that the earthquake magnitude used in this paper is the moment magnitude (Mw) and it should be noted that, for a few earthquakes from the aforementioned datasets, only the Ml (local magnitude) is available. However, according to Grünthal and Wahlström (2003), the difference between Mw an Ml can be ignored for magnitudes above 4, which represents the main focus of this study.

## 4 Method

### 4.1 FLEM computation

Starting from the entire dataset of faults in Italy, as a first step, we measured the length of each fault as the real fault trace length in map view, i.e., the length of the vertical projection of the fault trace as observed on the Earth's surface over a horizontal plane (Fig. 2; supplement; Petricca et al., 2018). Our complete dataset includes 12467 faults. Specifically, it includes 9169 A-class faults and 3298 B-class faults. Explanations for the classification into A- and B-class faults are given in the following sections.

As most faults have a horizontal length that is less than 25 km regardless of the selected database (Fig. 4), we divided the Italian territory into a grid with square cells of 25x25 km (Fig. 5; i.e., see also the explanation below). The length of the longest fault crossing each cell determined the parameter "fault length" (Lf) of the considered cell. In the second step, we used these lengths (Lf) as input parameters to empirically derive the earthquake magnitude (i.e., FLEM) of each cell containing at least one fault (Supplement; Petricca et al., 2018). Our only criterion to choose the fault from which the FLEM of the considered cell is computed is the greatest fault length in map view. In such a way, only one fault (the longest one) will provide the FLEM in a given cell. Therefore, faults cannot be and are not double-counted. Several studies investigated the scaling relationship between earthquake magnitudes and various geometric-kinematic parameters (i.e., fault dimensions and slip) of causative faults (e.g., Wells and Coppersmith, 1994; Leonard, 2010; Thingbaijam et al., 2017, and references therein), providing similar empirical equations. We used the equation proposed by Leonard (2010) (hereafter named as L10) that is expressed as:

$$FLEM = M = a + b * log(Lf) \tag{1}$$

where $a$ and $b$ are constants related to fault kinematics and are equal to 4.24 and 1.67, respectively (see table 6 in Leonard, 2010), and $Lf$ is the fault length as explained above. From Eq. 1, in our study, FLEM values range from 4.74 to 7.40, where the two end members are obviously imposed by the minimum and maximum fault length of the dataset, i.e., $\simeq 2$ and $\simeq 75$ km, respectively (Supplement; Petricca et al., 2018). Reasons for the choice of L10 are explained below.

## 4.2  Limits and assumptions

The method used in this study is based on some approximations and assumptions regarding the following points: (1) the arbitrary dimension of the cells (25x25 km); (2) the choice of the Leonard's equation and parameters (L10; Leonard, 2010) instead of others (e.g., Wells and Coppersmith, 1994); (3) the reliability of fault lengths; and (4) the assumption that all considered faults are active or can be potentially reactivated.

(1) Cell dimensions: as explained above, we choose a cell size of 25x25 km after considering the mean length (Figs. 4 and 5) of our fault dataset (Supplement; Petricca et al., 2018). Moreover, crustal earthquakes in Italy are mostly generated at a depth of 4-10 km. Within this depth range, the attitude of faults determines the distance between the fault trace and the epicentre over the Earth's surface. For instance, a hypocentral depth of 10 km along a causative fault dipping $45°$ implies a horizontal shift of about 10 km between the fault trace on the Earth's surface and the earthquake epicentre. Since the hypocentral depth is usually less than 10 km, a cell dimension of 25 km favours the occurrence of causative faults and related earthquake epicentres in the same cell. Moreover, to test the suitability of the size (25 km) of our square cells, we tested the sensitivity of the FLEM and the difference between the FLEM and the catalogued earthquake magnitudes on the cell size. We performed this sensitivity test on central Italy that is one of the most seismic areas of Italy. In this area, we changed our grid (i.e., cell size 25x25 km) both into a finer grid characterized by a cell size of 12.5x12.5 km and into a coarser grid characterized by a cell size of 50x50 km (Figs. S1 and S2). Each 50 km side square cell includes four 25 km side square cells and sixteen 12.5 km side square cells. For all cells, we computed the FLEM (Fig. S1) and the difference between the FLEM and the catalogued earthquake magnitudes

(Fig. S2). Then, cell by cell, we compared results (i.e., the FLEM) obtained for the largest and smallest grids (i.e., with square cell size of 12.5 and 50 km) against the FLEM obtained in the same geographical position using the intermediate grid (i.e., with square cell size of 25 km) (Fig. S3). This analysis shows that a change of grid size between 12.5 and 50 km does not provoke dramatic changes in the spatial distribution of both the FLEM (Fig. S1) and the difference between the FLEM and

the catalogued earthquake magnitudes (Fig. S2). In particular, using the largest grid (cell size = 50 km), the FLEM tends to be smoothed toward an upper boundary between magnitude 6 and 7 (Fig. S3a). This is due to the fact that only the longest faults of the dataset are considered for the FLEM computation. On the contrary, using the smallest grid (cell size = 12.5 km), the FLEM tends to be rather scattered and depressed around a magnitude 6 (Fig. S3a). This is due to the fact that reducing the cell size, in many cases, minor (small) faults result as being the longest ones in small cells. These results are also synthetically

expressed by the arithmetic averages of non-null FLEMs in the three different grids that are: $\text{FLEM}_{average}$ = 6.04, 6.19, and 6.62 for the grids with 12.5, 25, and 50 km sized cells, respectively. This sensitivity analysis shows that a microzonation (12.5 km sized cells) tends to locally overvalue small faults that are clearly poorly relevant when compared with adjacent (10-20 km distant) longer faults. On the contrary, a macrozonation (50 km sized cells) could overvalue long faults whose effect may be strongly reduced in a large area such as that included in a 50x50 km cell. Our choice (cell size = 25 km) appears therefore a

good compromise between significant faults and their potential areal influence.

     (2) Adopted equation parameters: The most popular scaling ~~equations~~ relationships that directly relate fault length and earthquake magnitude are provided by Wells and Coppersmith (1994), Leonard (2010), and Thingbaijam et al. (2017), among others. In this work, we used the equation (L10) by Leonard (2010) for dip-slip faults (both extensional and compressional). This equation is particularly suitable for Italy, where most earthquakes are generated by dip-slip faults (e.g., Chiarabba et al.,

2005). The main advantage of using Eq. 1 is that L10 is valid for both buried and outcropping fault as well as for normal and reverse (dip-slip) faults. To assess the difference (i.e., Fig. 6) between L10 and the equations provided by others, in Fig. 6a, we plotted the computed earthquake magnitude as a function of fault length for all scaling ~~equations~~ relationships by Wells and Coppersmith (1994), Leonard (2010), and Thingbaijam et al. (2017). Fig. 6b shows the difference between L10 and all other equations. For large fault lengths (i.e., corresponding to large earthquake magnitudes), the difference is mostly less than

5%. This difference increases to about 15% for smaller faults (i.e., for fault length $\leq$ 2 km) corresponding to earthquake magnitudes less than about 4.8 (Fig. 6a). These fault lengths ($\leq$ c. 2 km) and earthquake magnitudes ($\leq$ 4.8) represent the lower boundary of this study. For this reason, we assessed the difference of results from L10 with results from other scaling ~~equations~~ relationships (Fig. 6) as acceptable for the aims of our study. In particular, also considering the version of L10 given for strike-slip faults, the difference with the version of L10 that we used (i.e., for dip-slip faults) is less than 3% (Fig. 6b). Note

also that, due to the lower boundaries of the magnitude range for which the equations considered in Fig. 6 are valid (i.e., Mw $\geq$ 4.8-5.0 forWells and Coppersmith (1994) and Mw $\geq$ 5.8 for Thingbaijam et al. (2017)), Fig. 6 is relevant for Mw higher than these lower boundaries.

     (3) Fault length reliability: FLEM is calculated through Eq. 1 from the longest fault falling in each cell (Fig. 5). For faults located on-shore and well exposed at the surface, detailed studies allow for a well-constrained and reliable characterization of

fault length (class A faults). On the contrary, for some specific regions like the south-western Alps, the eastern front of the Po

Plain, the eastern Alps, the off-shore Adriatic front and Calabrian Arc, where fault planes are not exposed at the surface (i.e., buried and/or off-shore faults), fault geometry has been mainly constrained from regional seismic reflection profiles or from earthquake sequences. Therefore, fault length and along strike continuity are not well constrained, ~~hence,~~ producing the longest faults of the whole dataset. We assess the dimension of these latter faults as poorly constrained and reliable (class B faults).

Following this analysis on the accuracy on fault dimension, we decided to divide the dataset into two quality classes. Class A (high quality) includes exposed faults where subsurface and surface data allow for a detailed and reliable characterization of fault length (light purple cells in Fig. 5), whereas class B (low quality) contains buried and off-shore faults investigated mainly by seismic surveys, for which a precise characterization of fault length cannot be achieved (dark grey cells in Fig. 5). Moreover, we acknowledge that FLEM (fault length earthquake magnitude) is indeed the maximum magnitude expectable

from the actual size of the causal faults based on Eq. 1. A co-seismic lengthening of the causal faults through, for instance, the rupture of a bridge separating two adjacent faults to form one may produce earthquakes with a magnitude greater than that expected from the length of each of the two faults as measured before the fault co-seismic junction. Moreover, rupture jumps and coseismic slip distribution can happen during earthquakes thus producing earthquakes more energetic than what expected from the activation of a single fault or fault segment. An explicatory example in this sense is the Mw 7.8, 2016, Kaikoura

earthquake, New Zealand, when the coseismic rupture nucleated as a weak strike-slip event along the Humps Fault. This rupture progressively moved northward onto a shallow contractional fault, where most seismic moment was delivered, before it activated slip on a further system of strike-slip faults at the northern tip of the rupture (e.g., Cesca et al., 2017). Another explicatory example of the uncertainty connected with the adopted method is the Fucino Fault in central Italy, which generated the destructive Mw 7.0 (estimated from damages and a maximum Mercalli intensity of XI) earthquake of 1915. The Fucino

Fault in our database is 15.86 km long, corresponding to a FLEM (Mw) of 6.25. It is also true, however, that the 1915 coseismic rupture occurred along at least two parallel normal faults (Michetti et al., 1996). Hence, also the Fucino case shows that the method adopted in this work is poorly suitable in cases of multiple or complex coseismic ruptures.

(4) Fault orientation and activity: ~~although~~ it is well known that each earthquake is usually associated with a precise slip on a fault plane and that the potential of a fault to undergo (seismic) slip depends on its orientation within the stress field (Morris

et al., 1996; Collettini and Trippetta, 2007)~~, it~~. It is also true, however, that the seismic history of the Earth is characterized by many examples of unexpected earthquakes occurring where plate boundaries are far away (i.e., intraplate earthquakes), and/or the stress field is apparently badly oriented to trigger earthquakes (Bouchon et al., 1998; Camelbeeck et al., 2007; Kafka, 2007; Stein and Mazzotti, 2007; Swafford et al., 2007; Braun et al., 2009; Boyd et al., 2013; Leonard et al., 2014; Talwani, 2014; Campbell et al., 2015; Walsh III and Zoback, 2016; Christophersen et al., 2017). Since our study aims to define the Fault-Length

Earthquake Magnitude (FLEM) of each fault and cell, based on the aforementioned instances, we assumed that all faults are potentially reactivable. This notion becomes increasingly relevant when considering the prospective behaviour of faults over long terms. As mentioned above, indeed, for some societal challenges such as the safety of nuclear waste repositories, the recommendations are to consider the behaviour of faults in the future up to even 1 Ma (see, for instance, prescriptions by the Nuclear Energy Agency and International Atomic Energy Agency, www.oecd-nea.org, www.iaea.com; NEA, 2004; IAEA,

35 2016).

## 5 Results~~and discussion~~

Bearing in mind the limits mentioned above, we firstly calculated the FLEM for the Italian territory using the faults of class A (Fig. 7; supplement; Petricca et al., 2018). Most of the Italian territory, including the entire Apennine belt, the north-eastern Alps, and the central part of the Po Plain, is characterized by $6.0 \leq FLEM \leq 7.0$. The largest FLEMs have been obtained for the north-eastern Alps (FLEMs $\leq 7.4$) and are mainly related to the E-W-striking thrusts of the area responsible for the 1976 Friuli earthquake (Mw = 6.4; Bressan et al., 1998; Cheloni et al., 2012). In this area, we recall also the Carinthian great earthquake of 1348 (Villach, Mw 7.0; Rohr, 2003). In the central Alps, long and continuous thrust faults are reported in the ITHACA dataset (e.g., Maurer et al., 1997; Keller et al., 2006) and result in FLEMs of about 6.5. These large thrust faults also characterize the western part of the Po Plain, producing FLEMs of about 7.0. FLEMs between 6.5 and 7.0 are also estimated toward the south along the Apennine chain and in the Tuscany region, where large normal faults are reported from the ITHACA dataset and confirmed by detailed studies (Brogi et al., 2003; Brogi and Fabbrini, 2009). The largest FLEMs along the northern Apennines (Fig. 7) are due to the Alto Tiberina low angle normal fault, a large regional structure that seems to accommodate part of the deformation by aseismic creep and microseismicity (Chiaraluce et al., 2007; Collettini, 2011; Anderlini et al., 2016).

Large FLEMs occur also in the Vesuvius area (Campania region), where faults longer than 30 km have been reported in the ITHACA dataset (Fig. 5). Several studies (Finetti and Morelli, 1974; Scandone and Cortini, 1982; Vilardo et al., 1996; Brozzetti, 2011) confirm the presence of these long faults in the Vesuvius area. In the southern Apennines (Campania, Basilicata, and Calabria regions), FLEMs are in the range of 6.5-7.0, being related to the largest extensional faults of the area (Figs. 2 and 5). To the south, in the northern portion of Sicily, FLEMs > 7.0 are related to long off-shore faults. Some of these FLEMs are related to the transtensional-transpressional Tindari Fault System, located in NE Sicily. Other FLEMs are connected to the extensional system of the Messina Straits (Ghisetti, 1979; Locardi and Nappi, 1979; Lanzafame and Bousquet, 1997; Billi et al., 2006b, 2007; Palano et al., 2012; Cultrera et al., 2017).

The map of FLEMs derived using class B faults (Fig. 8) shows seismic events in the magnitude interval of 5.0-6.0 in the Tyrrhenian sector and in the southern portions of Puglia. Large FLEMs (up to M $\simeq$ 7.85) occur along the north-eastern Alps, the thrust fronts beneath the Po Delta, the north Adriatic front, the Ionian off-shore of the Calabria-Peloritani Arc, and some areas of Sardinia (Fig. 8). FLEMs $\simeq$ 8.0 derive from structures constrained only by subsurface data, for which the along-strike continuity cannot be properly assessed and, hence, the total length is likely to be largely overestimated. The largest FLEMs calculated for the Sardinian territory are related to structures inferred from large-scale (1:500,000) maps. These structures are longer (Fig. 5, red segments) than those derived, in the same areas, from detailed studies (Fig. 5, pink segments)~~and, therefore~~. Therefore, the actual length is ~~also~~ likely to be overestimated in this case.

To compare the FLEMs estimated on the grounds of geological fault length (Figs. 7 and 8) with the maximum magnitudes obtained from the historical and instrumental seismicity databases (i.e., catalogued earthquake magnitudes), we used the same grid presented above (Fig. 5). In particular, for each cell of Fig. 5, we selected the maximum earthquake magnitude recorded in historical or instrumental earthquake databases, applying a lower cut-off at magnitude 4.0 (Fig. 9). Large earthquake magnitudes - i.e., M > 6.0 earthquakes - were recorded in the north-eastern Alps, in the Po Plain, and along the entire Apennine

chain. The strongest events (M $\leq$ 7.4) were recorded in the north-eastern Alps and in the southern portion of the Apennines, including the Messina Straits and southern Sicily (Fig. 9). Earthquakes with magnitudes $\simeq$ 4.0-4.5 occurred almost everywhere in the Italian territory.

To spatially compare the earthquake magnitudes obtained through the FLEM computation (Figs. 7 and 8) and those recorded in the historical-instrumental catalogues (Fig. 9), we realized a map of the earthquake magnitude differences to show for each cell the difference between FLEMs obtained from the length of class A (Fig. 10) and class B (Fig. 11) faults and the magnitudes of historically/instrumentally-recorded earthquakes. In the case of class A faults, this comparison (Fig. 10) shows, in general, a difference of M $\leq$ 1.5. By fitting the differences distribution with a Gaussian curve, we obtained a mean of 0.86 with a $2\sigma$ (double standard deviation) of 1.47 on FLEMs derived from class A faults (Fig. 12a). The $2\sigma$ value is amplified to 1.85 (Fig. 12b) when also considering FLEMs from poorly reliable fault lengths (i.e., difference for class A+B faults in Fig. 12b).

Finally, we compared FLEMs with only the strongest earthquakes (M $\geq$ 6.5) in the historical and instrumental catalogues since 1000 A.D. to 2017 (Fig. 13 and also Table 1). In this case, the difference with the spatially-corresponding FLEMs (i.e., same cell) is less than 1.0 and, in most instances, less than 0.4 (Fig. 13a). In particular, the mean difference is -0.09 with a $2\sigma$ value of 0.53 (Fig. 13c). Note also that, even when not equal to the spatially-corresponding catalogued earthquake magnitudes, yet many FLEMs fall within the uncertainty interval (i.e., in Fig. 13b, see the red circles falling along the blue vertical bars) associated with these catalogued magnitudes.

In the histograms of Figs. 12 and 13(c), the values in the negative fields can be interpreted as FLEM's underestimation (i.e., where FLEM values do not reach the catalogued earthquake magnitudes, which are in turn affected by uncertainty; e.g., Fig. 13b). In contrast, values in the positive fields of Figs. 12 and 13(c) could be interpreted either as FLEM's overestimation (i.e., where FLEM values unsuitably exceed the catalogued earthquake magnitudes) or as a sort of catalogue incompleteness (i.e., where the catalogued earthquake magnitudes do not properly represent larger seismic potentials of faults). Occurrences in the positive fields (green portions in Figs. 12 and 13c) are ~~significantly~~ more frequent than occurrences in the negative fields (red portions in Figs. 12 and 13c), particularly in Fig. 12. These positive and negative occurrences are also mapped in Figs. 10 and 11 with green and red tones, respectively, whereas white cells are where FLEM's differences with respect to catalogued earthquake magnitudes are about null. Figs. 10 and 11 show that negative (red) occurrences are very limited.

It is also interesting to note that, in the period between our dataset of catalogued earthquakes (i.e., in times younger than December 2017) and the time of writing (September 2018), only one crustal M>5 earthquake occurred in Italy. Namely, the Montecilfone Mw 5.1 earthquake occurred on August 16th, 2018, in the central Apennines at latitude 41.87N and Longitude 14.86E and depth of 20 km. In the same locality, our FLEM is 5.9, corroborating the observations and considerations made above on the difference between FLEMs and catalogued earthquake magnitudes.

Summarizing, ~~the difference~~ large differences between FLEMs and catalogued earthquakes can be either real, indicating that some large earthquakes, possibly due to extremely long recurrence intervals ~~,~~ are not contained in the seismological records, or a bias induced mainly by the following factors: (1) impossibility to resolve fault continuity and segmentation with the adopted method and (2) deformation partially accommodated through aseismic creep.

~~The limited difference~~Small differences, as discussed above, between FLEMs and the catalogued earthquake magnitudes (Figs. 12a and 13c) ~~is~~ are due to the comprehensive knowledge of the exposed faults (particularly faults of class A, Fig. 12a). A large amount of data is indeed available from detailed field surveys and subsurface investigations realized over the years. However, some portions of the eastern Alps, northern Apennines, and southern Italy, including Sicily, show a difference in the magnitude of 2.0-2.5 (i.e., faults of class B, Fig. 12b). In these areas, more detailed studies should be developed in order to better characterize fault dimensions and properly assess seismic hazards. It is, however, very encouraging that when considering only the historical and instrumental earthquakes with M ≥ 6.5, the difference between FLEM values and the catalogued earthquake magnitudes ~~significantly~~ reduces (Fig. 13).

Regarding the FLEMs evaluated using class B faults (Fig. 11), we observe a significant difference with catalogued earthquakes in several regions. Differences of up to 4.0 in magnitude (Fig. 12b) are estimated in correspondence of the north Adriatic thrust front, off-shore from the Calabria-Peloritani Arc, and the northeast part of the Alps (Fig. 11). Smaller, but still relevant, differences of M ≃ 2.0 are documented in correspondence of the Apennines front beneath the Po Plain. As mentioned above, class B fault lengths have been evaluated mostly using seismic reflection profiles. Due to the resolution of this technique and the quality of some datasets, fault segmentation cannot be properly evaluated. More evidence would be needed. In general, the significant differences between FLEMs (class B faults) and the catalogued magnitudes require that extensive 3D geological and geophysical investigation of the structures of these areas should be performed in order to better characterize the geometry and continuity of faults. To this end, it is also noteworthy that studies on the 2016 Amatrice-Norcia (central Italy) earthquakes (Mw 6.0 and 6.5) revealed that the length of the causative faults was only partially activated by the seismogenic slip (e.g., Chiaraluce et al., 2017); however, as this fault-slip behaviour seems rather frequent (Freymueller et al., 1994; Milliner et al., 2016; Chousianitis and Konca, 2018), it is most likely that this same behaviour is incorporated and implicitly expressed by the above-mentioned empirical scaling relationships between fault length and earthquake magnitude (e.g., Wells and Coppersmith, 1994; Leonard, 2010; Thingbaijam et al., 2017).

## 6 Statistical ~~Test~~ Analysis of ~~FLEM Values~~ Results

The reliability of estimated FLEM values may be quantitatively checked, in principle, using a formal statistical test on earthquake catalogues. Anyway, models of expected maximum magnitude (Mmax) are not readily testable. A statistical test helping to discriminate between competitive FLEM values is impossible in practice, even in the simplest case of discriminating between a double truncated Gutenberg-Richter (DTGR) law and an unbounded Gutenberg-Richter (UGR, Mmax=∞) law on excellent datasets (Holschneider et al., 2014). Suffice is to say that the DTGR's probability distribution differs from the UGR's one by a constant $c = 1 - 10^{-b(M_{max} - M_c)}$ (where $M_c$ is the minimum magnitude). This value is very close to 1, if not when $M_c$ is close to Mmax. This means that the upper cut-off of the Gutenberg-Richter distribution can be explored using only large, very rare earthquakes, with magnitudes close to the maximum possible value, on which statistical inference is necessarily limited (Holschneider et al., 2014). Therefore, it is essentially impossible to statistically infer, with sufficient confidence, the maximum possible earthquake magnitude, in terms of alternative testing, from an earthquake catalogue alone (Holschneider

et al., 2014). In light of the limitations of purely statistical inference, geological and tectonic information provides, therefore, important and exclusive constraints on the expected maximum magnitude. If we look at statistical testing in detail, we can check the FLEM values on a catalogue, while controlling the probability of wrongly rejecting them, but without reducing the probability of a wrong non-rejection (Holschneider et al., 2014). This makes it, indeed, impossible to discriminate between two likely values of FLEM. In other words, if we do not reject a FLEM value, we cannot say if it is true or if the data are inadequate in terms of revealing its failure. Keeping all these considerations in mind, we test the estimated FLEM values on the CSI1.1 and ISIDe databases (available online at https://csi.rm.ingv.it/ and http://iside.rm.ingv.it/, respectively). Since our analysis requests a database containing small magnitudes events, we consider ISIDe data of earthquakes occurred before ~~April~~ 16 April 2005, when the new INGV (Istituto Nazionale di Geofisica e Vulcanologia) National Seismic Network, completely reorganized and equipped with a new acquisition system, became operational (Amato and Mele, 2008). Without this separation, the hypothesis of temporal homogeneity for magnitude data would not be appropriate. Moreover, most events in the ISIDe and CSI1.1 databases have a ML magnitude, while FLEM values are in moment magnitude scale. Therefore, we repeat the same testing procedure, described below, twice, without any difference in results: firstly, we consider original FLEM values and, secondly, we convert them in local magnitude scale through the relations proposed by (Gasperini et al., 2013). We adopt the following test procedure.

a) We select cells for which a test is helpful, i.e. having a FLEM value above the related historical/instrumental observed magnitude. In this way we exclude 41 of 1100 cells for which FLEM values are estimated. A further selection consists of keeping only cells for which the Gutenberg-Richter model, with a b-value equal to 1, is not rejected. In this way, we are sure that a possible rejection of FLEM values cannot be ascribed to the low reliability of a Gutenberg-Richter relation or to wrong b-values. To this end, we apply a goodness of fit test (GFT, Wiemer and Wyss, 2000), at a 95% confidence level, that also provides a completeness magnitude value, $M_c$, for each cell. The GFT is based on difference R of the observed ($O_i$) and expected ($E_i$) numbers of events in each magnitude bin. Values $E_i$ are computed by assuming a UGR distribution, with a b-value equal to 1, above an ascending magnitude cutoff $Mt$. So that R is given by:

$$R(M_t) = 1 - \frac{\sum_i |E_i - O_i|}{\sum_i O_i} \tag{2}$$

where sums are done for magnitude bins above $M_t$. The completeness magnitude $M_c$ is defined as the first value of $M_t$ for which R($M_t$)>0.95. The cells passing this test are 30 (Table S1) and 67 (Table S2) for the CSI1.1 and ISIDe databases, respectively. We stress that this first test does not require a Mmax value.

b) A first specific check of the DTGR distribution, having FLEM as maximum magnitude, is carried out by using a log-likelihood test. Specifically, for each cell passing the previous test, the log-likelihood value of the DTGR model is computed. By considering that a DTGR distribution has a probability density function given by:

$$f_{DTGR}(M) = \frac{\beta e^{\beta(M-M_c)}}{1 - e^{\beta(Mmax-M_c)}} \tag{3}$$

where $b = \beta ln(10)$; we compute for each cell the real log-likelihood $LL_R$ on N events magnitudes $M_i, i = 1, .., N$, by equation:

$$LL_R = \sum_{i=1}^{N} ln[f_{DTGR}(M_i)] \tag{4}$$

The DTGR model is rejected if $LL_R$ is significantly lower than expected values. The probability of having smaller log-likelihood values than $LL_R$ is estimated using $10^4$ simulated datasets (having the same size of real datasets) by model DTGR. Specifically, for each cell, we compute this probability pLL as the proportion of simulated log-likelihoods smaller than $LL_R$. We select cells for which these probabilities are larger than 5%. In this way, the cells are reduced to 20 and 45 for the CSI1.1 and ISIDe databases, respectively (Figure 14 and Tables S1 and S2). We stress that we select the same cells by assuming a UGR model, suggesting that the problem of excluded cells lies in the whole Gutenberg-Richter relation and not in the FLEM values.

c) Finally, FLEM values are tested by applying the procedure proposed by (Holschneider et al., 2014), involving a comparison of the maximum observed magnitude with a suitable threshold magnitude, computed by the DTGR model. Specifically, we reject the FLEM values, if the observed magnitude (oMmax) is larger than a threshold value $M_t$, given by:

$$M_t = M_c - \frac{1}{\beta} ln\{1 - (1 - \alpha)^{\frac{1}{n}} (1 - e^{-\beta(FLEM - M_c)})\} \tag{5}$$

where $\alpha = 0.05$ (see Holschneider et al., 2014, for details). The computed threshold values ($M_t 1$) are listed in Tables S1 and S2, together with oMmax values. All previously selected cells pass this test, for both the CSI1.1 and ISIDe databases, suggesting that related FLEM values cannot be rejected (Figure 14 and Tables S1 and S2).

d) The last two steps are repeated by reducing the magnitude range covered by data. Specifically, we increase, where possible, the minimum magnitude to FLEM-2.0. The relative threshold values $M_t 2$ are listed in Tables S1 and S2 for the CSIv1.1 and ISIDe databases, respectively. Also in this case, we cannot reject the FLEM values. As stated above, this analysis does not exclude alternative values of FLEM; as . As matter of fact, it is also the case that a UGR model cannot be rejected in all the selected cells.

## 7 Conclusions

In this study, (1) we first provided an updated compilation of all a comprehensive dataset of mapped faults in Italy, (2) then, using known scaling laws, calculated the related Fault-Length Earthquake Magnitude (FLEM), and, (3) finally lastly, compared FLEM values with historically/instrumentally-catalogued earthquake magnitudes. Where faults are geologically well constrained (class A faults), either good agreements or some agreements or differences are observed between FLEMs and historically/instrumentally-catalogued earthquake magnitudes: the agreement is increased agreements increase for M $\geq$ 6.5

earthquakes. In areas where fault geometries are poorly constrained (class B faults inferred solely by subsurface 2D investigations), ~~large~~ larger differences are observed~~: these~~. These areas have to be further characterized to better estimate fault dimensions and hence properly assess the FLEM. Our results are partly encouraging and suggest the testing and validation of this experiment elsewhere. This method cannot, however, be a substitute for time-dependent (paleo)seismological methods for

seismic hazard assessments. Rather, it can ~~provide an approximate perspective time-independent seismic potential of faults and~~ highlight areas where further detailed studies on faults are required.

*Data availability.* Copyrighted catalogue data are available online at csi.rm.ingv.it, iside.rm.ingv.it and emidius.mi.ingv.it/CPTI15-DBMI15. Original data are available at http://doi.org/10.5880/fidgeo.2018.003 (Petricca et al., 2018). For copyright reasons, the related vector data are available only for personal scientific use upon request to the authors. Alternatively, some vector data are available online at sgi.isprambiente.it/geoportal.

Further data are available in the supplement associated to this paper.

*Author contributions.* CD obtained the funding; all authors conceived the experiment and the paper; PP, AB, FT, DS, MC, and CC performed the experiment and analyzed the data under the coordination of CD and GV; all authors discussed the results and drew the related conclusions; FT, AB, and PP wrote most part of the manuscript; PP realized the figures; all authors reviewed and accepted the manuscript and figures.

*Competing interests.* The authors declare that they have no conflict of interest.

*Acknowledgements.* Most figures were produced using the GMT software (http://gmt.soest.hawaii.edu/). Pierfrancesco Burrato is thanked for providing the digital version of the Structural Model of Italy. A special thanks to Federica Riguzzi for fruitful discussion during the project. Copyrighted catalogue data are available online at ~~http://~~csi.rm.ingv.it, ~~http://~~iside.rm.ingv.it ~~/iside/standard/index.jsp and https://~~and emidius.mi.ingv.it/CPTI15-DBMI15. Original data are available at ~~the following link: http://pmd.gfz-potsdam.de/panmetaworks/review/924b171fd21c78f~~ ~~(~~http://doi.org/10.5880/fidgeo.2018.003 ~~;~~ (Petricca et al., 2018). For copyright reasons, the related vector data are available only for personal

scientific use upon request to the authors. Alternatively, some vector data are available online at http://sgi.isprambiente.it/geoportal. Further data are available in the supplement associated to this paper. We warmly thank F. Rossetti, Y. van Dinther, S. Nandan, K.K. Thingbaijam, an anonymous reviewer, and other colleagues (Valensise et alii, see the Discussion Forum of Solid Earth) for constructive comments. The presented results should be considered more in a theoretical and methodological perspective for comparison with future similar studies rather than in an applicative perspective for the case of Italy. In particular, our assessed earthquake magnitudes (FLEMs) for the Italian territory

are proposed in this paper for scientific reasons and not for their use for civil protection and prevention purposes. Moreover, in this article, we do not address or estimate the probability of earthquake occurrence. Yet, we would like to acknowledge that some large magnitudes of earthquakes (FLEMs), calculated in this article are considered very unlikely in the existing literature of seismic hazard in Italy.

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

# From mapped faults to Fault-Length Earthquake Magnitude (FLEM): A test on Italy with methodological implications

Fabio Trippetta[1], Patrizio Petricca[1], Andrea Billi[2], Cristiano Collettini[1,3], Marco Cuffaro[4], Anna Maria Lombardi[3], Davide Scrocca[4], Giancarlo Ventura[5], Andrea Morgante[5], and Carlo Doglioni[1,3]

[1]Dipartimento di Scienze della Terra, Sapienza Universitá di Roma, Rome, Italy
[2]Consiglio Nazionale delle Ricerche, Rome, Italy
[3]Istituto Nazionale di Geofisica e Vulcanologia, Rome, Italy
[4]Istituto di Geologia Ambientale e Geoingegneria, CNR, Rome, Italy
[5]Sogin (Società Gestione Impianti Nucleari) S.p.a., Rome, Italy

**Correspondence:** Andrea Billi (andrea.billi@cnr.it)

**Abstract.** Empirical scaling laws between fault/slip dimensions and earthquake magnitudes are often used to assess the maximum possible earthquake magnitude of a territory. In this paper, upon the assumption of the reactivability of any fault, these seismic scaling relationships are compared at the national scale in Italy against catalogued magnitudes, considering a comprehensive fault dataset regardless fault age, stress field orientation, strain rate, or else. Italy is a suitable case for comparing the fault-size-derived seismic magnitudes with the existing accurate catalogues of historical-instrumental earthquakes. To do so: (1) a comprehensive catalogue of all known faults is compiled by merging the most complete databases available; (2) the earthquakes magnitude (FLEM) is simply derived from fault length; and (3) the resulting FLEMs are compared (i.e., the mathematical difference) with catalogued earthquake magnitudes. Results show that the largest FLEMs as well as the largest differences between FLEMs and catalogued magnitudes are observed for poorly constrained faults, mainly inferred from subsurface data. It is suggested that these areas have to be further characterized to better estimate fault dimension and segmentation and hence properly assess the FLEM. Where, in contrast, the knowledge of faults is geologically well constrained, the calculated FLEM is often consistent with the catalogued seismicity, with the $2\sigma$ value of the distribution of differences being 1.47 and reducing to 0.53 when considering only the $M \geq 6.5$ earthquakes. The main advantage of this method is its independence from temporal and (paleo)seismological information, whereas the main novelty is its use at the national scale also for faults considered inactive.

## 1 Introduction

In some seismically active regions like California, information derived from geodesy, the geology of active faults, and seismology (both historical and instrumental) have been combined to develop a comprehensive method that predicts 99% of the chances of having one or more $M \geq 6.7$ earthquakes in the following 30 years (USGS Fact Sheet, 2008). Along the North Anatolian Fault (Turkey), fault geometry and slip accumulated during strong earthquakes have been used to infer the transfer of stress during seismic sequences and to estimate the increase of earthquake probability with $M \geq 6.7$ (Stein, 1997; Parsons

et al., 2000; Bohnhoff et al., 2016). These case studies involve areas characterized by high strain rates and affected by large plate boundary faults that can be easily recognized in geological and/or geophysical records.

In contrast, other areas in the world where strain rates are low-to-intermediate and faults show smaller dimensions and unclear surface expression of recent activity, the connection between faults and earthquakes is not straightforward. This, in particular, applies to the relationship between potentially seismogenic faults and the maximum possible earthquake magnitude. Moreover, where lands were largely uninhabited during historical times (but are now densely populated and, therefore, potentially exposed at the seismic threat), the recurrence time of strong earthquakes is significantly longer with respect to the age of the seismological network. Further, in these areas information on historical and instrumental seismicity can be largely incomplete and the assessment of the maximum possible earthquake magnitude can be difficult (Camelbeeck et al., 2007; Kafka, 2007; Stein and Mazzotti, 2007; Swafford et al., 2007; Dawson et al., 2008; Braun et al., 2009; Boyd et al., 2013; Leonard et al., 2014; Talwani, 2014; Campbell et al., 2015; Calais et al., 2016; Christophersen et al., 2017; Wang et al., 2019).

Where frequent earthquakes are absent and seismic history is unknown, the seismic potential and the temporal knowledge of seismic activity of faults can be determined using geological studies such as paleoseismological trenching or radiometric dating of slip indicators (Galli, 2000; Rockwell et al., 2000; Palumbo et al., 2004; Dixon et al., 2003; Galli et al., 2008; McCalpin, 2009; Sherlock et al., 2009; Nuriel et al., 2012; Viete et al., 2018). These studies however, are usually time-consuming and expensive. Alternatively, the maximum potential magnitude of earthquakes can be assessed using empirical scaling laws of active faults, that is, using fault length and/or fault slip (Wells and Coppersmith, 1994; Pegler and Das, 1996; Mai and Beroza, 2000; Henry and Das, 2001; Liu-Zeng et al., 2005; Leonard, 2010; Thingbaijam et al., 2017). In this latter case, however, a lack of information on fault age and earthquake recurrence time may induce to neglect some faults as potential seismogenic sources. One method to overcome the problems connected with the lack of information on the age of (seismogenic) faulting may be to apply the existing seismic scaling relationships, which directly link fault/slip dimensions and earthquake magnitudes (e.g., Wells and Coppersmith, 1994; Leonard, 2010; Thingbaijam et al., 2017), to the whole set of known faults including the (presumably-)inactive and undetermined faults. This method would rely on the long-held concept of fault reactivation (Sibson, 1985) due to the weakness of the fault surface compared with the host rock (Zoback et al., 1987; Collettini et al., 2009). This concept is particularly relevant when considering the future behaviour of faults over long terms. For nuclear plants, for instance, it is important to assess the occurrence of events, even if very unlikely, also in areas historically free of damage, where normally these plants are built. In these sites (nuclear plants) indeed only a very low level of risk can be accepted. In other cases such as the geological disposals of radioactive wastes, the lifetimes to be considered are on the order of 105 years at least (up to 1 Ma according to prescriptions by the Nuclear Energy Agency and International Atomic Energy Agency, www.oecd-nea.org, www.iaea.com; NEA, 2004; IAEA, 2016). Accordingly, very long seismic return times must be considered, far exceeding the lifespan of historical seismic catalogues (and also paleoseismological records) normally used in standard seismic hazard assessment practices.

For the richness of data (both tectonic and catalogued historical/instrumental-seismology; Fig. 1), Italy is a suitable country for deriving and comparing the potential maximum magnitude from earthquake data and from scaling relationships. Italy has, indeed, a long tradition of geological-structural mapping over the entire territory (since at least Murchison, 1849; von Zittel,

1869; Viola, 1893; Pagani, 1907, to the recent national geological maps available online at http://sgi.isprambiente.it/geoportal/) as well as a dense seismological network operating since at least the 1980s (available online at http://cnt.rm.ingv.it/ Iside Working Group, 2016). Moreover, Italy has a rare, if not unique, historic record of earthquakes (Rovida et al., 2016).

This work includes: (1) the composition of a comprehensive catalogue of all mapped faults in Italy, merging materials from previous documents; (2) the calculations of earthquake magnitudes (FLEM, Fault-Length Earthquake Magnitude) from the fault length in map view; (3) a comparison (i.e., the difference) of these calculations (FLEM) with catalogued seismological data (historical and instrumental earthquake magnitudes); and (4) a discussion of the agreements and differences between the calculated (FLEM) and catalogued magnitudes.

We anticipate that, with this work, we do not intend to propose an alternative method for seismic hazard assessment or to better previous methods (e.g., Giardini, 1999; Jiménez et al., 2001; Michetti et al., 2005; Field et al., 2009, 2015; Reicherter et al., 2009). Our main aim is to test whether solely considering the known mapped faults (both active, inactive, and undetermined) and disregarding further information (e.g., historically- and instrumentally-recorded earthquakes as well as the regional stress field and strain rate) it is possible to provide, through existing seismic scaling relationships of faults and earthquakes, reasonable assessments of the maximum possible earthquake magnitude over an entire nation. The resulting (assessed) magnitudes (FLEMs) are compared (i.e., the mathematical difference) with catalogued earthquake magnitudes that are the only existing points of reference against which assessed magnitudes can be compared. Note that these results should be considered more in a theoretical and methodological perspective for comparison with future similar studies rather than in an applicative perspective for the case of Italy. In particular, our assessed earthquake magnitudes (FLEMs) for the Italian territory are proposed in this paper for scientific reasons and not for their use for civil protection and prevention purposes. Moreover, in this article, we do not consider or estimate at all the probability of earthquake occurrence. Yet, we would like to acknowledge that some large magnitudes of earthquakes (FLEMs) calculated in this article are considered very unlikely in the existing literature of seismic hazard in Italy (e.g., Cinti et al., 2004; Slejko et al., 2010).

## 2 Seismotectonic setting

The present setting of the Italian peninsula (Fig. 1) derives from the interaction (mostly convergence) of at least three main plates, namely, Eurasia to the north, Africa to the south, and Adria to the east and southeast. During Cenozoic-Quaternary times, this tectonic interaction has led to the formation of two major mountain chains - the Alps to the north and the Apennines along the peninsula - and to the opening of two major oceanic basins - the Ligurian-Provencal basin to the west of Sardinia and Corsica and the Tyrrhenian Sea basin between the Italian peninsula and the Sicily and Sardinia major islands (Dewey et al., 1989; Malinverno and Ryan, 1986; Doglioni, 1991; Doglioni et al., 1999; Faccenna et al., 2004; Rosenbaum and Lister, 2004; Carminati et al., 2010; Carminati and Doglioni, 2012).

The Alps and Apennines are characterized by very different tectonic styles. The Alps show double-verging growth (northward and southward), with the involvement of large volumes of crystalline basement and the exhumation of metamorphic rocks (Nicolas et al., 1990; Schmid et al., 1996). The Apennines form a single-verging chain characterized by a radial vergence and

a strong curvature from northwest to southeast. The Apennines are characterized by thin-skinned tectonics with rare exposures of the crystalline basement and metamorphic rocks (Barchi et al., 1998; Patacca et al., 2008; Scrocca et al., 2005). The development of the Apennines (contraction) occurred partly at the same time with the development of the Ligurian-Provencal and Tyrrhenian oceanic basins (extension). The different tectonic styles of Alps and Apennines are also well represented by

the different settings of the related foreland monoclines and basins. The foreland basin is shallow in front of the Alps with a monocline dipping by 2-4° and deep in front of the Apennines with a monocline dipping by 6-15° (Mariotti and Doglioni, 2000).

The present seismotectonic setting of Italy is still mainly ruled by the interaction between the African and Eurasian plates, presently converging at a rate of c. 10 mm/yr along a NNW-SSE direction (DeMets et al., 1990; Serpelloni et al., 2005;

Billi et al., 2011). This kinematic setting is complicated by the Adria plate (an independent microplate or a promontory of the African plate), which contributes to cause contractional deformations along the central-northern Adriatic margins (both toward the Apennines to the west and toward the Dinarides to the east) and in the Po Plain between the northern Apennines and the southern Alps. Contractional deformations are in contrast absent or poorly active in the southern Apennines where these deformations have been inactive or poorly active since about mid-Pleistocene times. In synthesis, for the aims of this

work, it is important to know that most of the Italian territory has been subject to numerous deformation phases with related activations, re-activations, and/or inversions of faults. These phases are schematically amenable to one or two main complex tectonic mechanisms: the Alpine orogenesis (Paleozoic to present times) and the Hercinian orogenesis (Paleozoic times) (Vai, 2001). At present, within the Alpine orogenesis, back-arc extensional seismic mechanisms prevail in large areas of the Italian territory (e.g., the axial portion of the Apennines), whereas compressional seismic mechanisms are less frequent and prevail

elsewhere (e.g., the eastern or Adriatic portion of the Apennines) (e.g., Basili and Meghraoui, 2001; Neri et al., 2002, 2005; Roberts and Michetti, 2004; Chiarabba et al., 2005, 2015; Palano et al., 2012; Presti et al., 2013; Ferranti et al., 2014; Cowie et al., 2017; Orecchio et al., 2017). Strike-slip tectonics is limited to a few major fault zones (e.g., Grasso and Reuther, 1988; Billi et al., 2003; Viganò et al., 2015; Polonia et al., 2016, 2017).

Italy has a widespread crustal seismicity (depth < 35 km) that concentrates along some portions of the Alps and mostly on

the Apennines, including Calabria and Sicily (Fig. 1). Deep earthquakes are mainly located beneath the Calabrian Arc and, secondarily, beneath the northern Apennines and eastern Alps, where sparse seismicity is recorded (Chiarabba et al., 2005). The eastern and western Alps show a clustered crustal seismicity (Fig. 1). In the central Alps, the number of recorded events seems to be less densely distributed. This is probably related to the lack of an appropriately-dense seismic network in this area (Amato, 2004; Chiarabba et al., 2005).

The western Alps are particularly active in their southern portion, where active N-S-striking faults accommodate most of the extensional and wrench deformations that characterize the present-day tectonics of the area (Chiarabba et al., 2005; Sanchez et al., 2010; Sue et al., 1999; Sue and Tricart, 2002). Focal mechanisms of the sparse seismicity in the central Alps show extensional kinematics on N-S-striking fault planes (Chiarabba et al., 2005); however, stress field studies in this area suggest mainly active N-S compression (Montone and Mariucci, 2016). The clustered seismicity recorded in the eastern Alps

is mainly located along E-W trending structures. This seismicity shows an overall compressive tectonics generating large thrust

earthquakes in response to a N-S trending compression, like the 1976 Friuli, Mw = 6.4 earthquake (Bressan et al., 1998; Cheloni et al., 2012; Michetti et al., 2012).

The Apennines seismicity is densely distributed along the whole chain (Fig. 1). Focal mechanisms of the chain show a rotation of the fault strike from NW-trending alignments to NNE-trending ones moving from north to south, along the arc-like shape of the Apennines. In general, in the middle portion (axis) of the Apennines, earthquakes are characterized by extensional kinematics, whereas the Adriatic front is mainly affected by compressive kinematics, which is in agreement with the regional stress field (Montone and Mariucci, 2016). In particular, the north-western portion of the Apennines shows a NW-SE trending cluster of seismicity. Moving eastward, beneath the Po Plain, seismicity involves the northern Apennines outer front (Fig. 1), where focal mechanisms are mainly compressive. These mechanisms highlight E-W-striking fault planes consistent with the attitude of the main structures of the area. Here, the largest instrumentally measured earthquakes (Mw = 6.1) occurred in 2012 during the Emilia (N-Apennines) sequence (Govoni et al., 2014). Moving southward, crustal seismicity (M 4-6.5) is densely distributed and follows the Apennine chain axis. In this zone (Umbria-Marche), the largest instrumental earthquakes with extensional focal mechanisms occurred in the 1997 Colfiorito, Mw = 6.0 (Amato et al., 1998); 2016 Norcia, Mw = 6.5 (Chiaraluce et al., 2017); 2016 Amatrice, Mw = 6.0 (Chiaraluce et al., 2017); 2009 L'Aquila (Abruzzi), Mw = 6.3 (Chiarabba et al., 2009); and 1980 Irpinia (Campania), Mw = 6.9 (Nostro et al., 1997) earthquakes. A cluster of moderate (M 4.0-5.0) seismicity is recognized close to the Tyrrhenian coast in relation to active geothermal and volcanic districts (Gasparini et al., 1985). Larger earthquakes characterize the Apennines southern portion (Calabria), with historical seismic events that reached magnitudes up to ≃7.1 (Rovida et al., 2016; Guidoboni et al., 2018).

The Calabrian Arc (Fig. 1) is mostly characterized by deep instrumental seismicity related to the NW dipping Benioff plane (Polonia et al., 2016; Selvaggi and Chiarabba, 1995). Both the Etna area and the northern on-shore portion of Sicily show clustered seismicity, where the latter is mainly characterized by extensional and oblique-extensional earthquakes (Azzaro and Barbano, 2000; Billi et al., 2006a). The strongest earthquakes (some of them with associated destructive tsunamis; Billi et al., 2010) from historical catalogues in the area occurred predominantly on extensional faults and are: the 1908 (Mw 7.2) earthquake of Messina and Reggio Calabria (Messina Straits, Sicily), the 1638 and 1905 (Mw 7.0) earthquakes of Nicastro (Calabria), and the 1693 (Mw 7.4) earthquake of eastern Sicily (Rovida et al., 2016).

The Southern Tyrrhenian (Fig. 1) is characterized by intense seismicity, with magnitudes up to 7.1, which occurred in 1938 at 290 km depth (Selvaggi and Chiarabba, 1995). The seismicity of shallow levels (< 30 km) below the southern Tyrrhenian Sea indicates the presence of two adjacent domains characterized by different tectonic environments: (1) to the northwest of the Aeolian Islands, a N-S compressive tectonics is present, whereas (2) to the east and southeast of these same islands, a NW-SE extensional tectonics occurs (Fig. 2) (Goes et al., 2004; Pondrelli et al., 2004; Billi et al., 2006a; Cuffaro et al., 2010). Intermediate and deep seismicity concentrates along a roughly uninterrupted, narrow, and steep (70°) Benioff zone, which strikes SW-NE, dips toward NW, and reaches a depth of about 500 km. Only one earthquake occurred inland at depth of 350 km (Pondrelli et al., 2004; Selvaggi and Chiarabba, 1995). Shallow compressive seismicity (< 30 km deep) is characterized by epicentres mainly aligned E-W (Pondrelli et al., 2004; Presti et al., 2013), off-shore northern Sicily. These epicentres align

roughly NNW-SSE moving eastward (Fig. 1). Sparse compressive events have also been recorded off-shore north-eastern and southern Sardinia.

## 3  Input data

### 3.1  Fault data

To build a comprehensive dataset of faults in Italy (Figs. 1-3; supplement; Petricca et al., 2018), the following databases were merged: (1) the entire fault collection of the Italian Geological Maps at the 1:100,000 scale (i.e., Carta Geologica d'Italia available online at www.isprambiente.it); (2) the fault compilation from the Structural Model of Italy at the 1:500,000 scale (Bigi et al., 1989); (3) all faults provided in the ITHACA-Italian Catalogue of Capable Faults (Michetti et al., 2000); and (4) the inventory of active faults from the GNDT (Gruppo Nazionale per la Difesa dai Terremoti; Galadini et al., 2000) database.

The strength point of our approach is the assemblage of different fault datasets heterogeneously built for different purposes and based on different primary information and methods. Although different, the common point of all used datasets is that they have faults mapped and therefore measurable over the Earth's surface. Geological studies and related mapping are ongoing in Italy and, frequently, new geological-structural maps are produced and published. We considered these further maps only for those areas that are, in our opinion, poorly covered by the main four databases mentioned above (see below for further

explanations).

Faults from the 1:100,000 Italian Geological Maps and 1:500,000 Structural Model of Italy (black and black dashed lines, respectively, in Fig. 2) are essentially based on field surveys integrated with subsurface geophysical data and, therefore, were drawn and constrained by geological and geophysical observations.

The ITHACA-Italian Catalogue of Capable Faults (Michetti et al., 2000) is a database developed by the ISPRA (Istituto

Superiore per la Protezione e la Ricerca Ambientale) containing cartographic and parametric information of active faults - i.e., faults with evidence of repeated reactivation during the last 40,000 years - capable of rupturing the ground surface in Italy. The database is available as a layer in a web GIS (http://www.isprambiente.gov.it/it/progetti/suolo-e-territorio-1/ithaca-catalogo-delle-faglie-capaci) and contains both the geographic location and a text description of each fault. The entire set of capable faults is included in this compilation (blue lines in Fig. 2).

The inventory of active faults of the GNDT database (Galadini et al., 2000) represents a collection of the Italian active faults. The activity of these faults is mainly deduced through surface geological evidence and (paleo)seismological data including historical information (red lines in Fig. 2).

To improve and implement these fault databases, we selected published complementary studies for some specific areas considered to not be exhaustively covered by the aforementioned collection of faults including Sardinia, SW Alps, Tuscany,

the Adriatic front, Puglia, and the Calabrian Arc. For these areas, we selected faults on the grounds of scientific contributions that documented the fault presence based on field, seismic, and paleoseismological data (pink lines in Fig. 2). In particular, for the Campidano area (southern Sardinia), we used the fault pattern proposed by Casula et al. (2001), who reconstructed fault geometry with recent tectonic activity based on field and seismic reflection profiles. For the SW Alps, we followed the works

of Augliera et al. (1994), Courboulex et al. (1998), Larroque et al. (2001), Christophe et al. (2012), Sue et al. (2007), Capponi et al. (2009), Turino et al. (2009) and Sanchez et al. (2010). For the Tuscany area, we consulted Brogi et al. (2003), Brogi et al. (2005), Brogi (2006), Brogi (2008), Brogi (2011) and Brogi and Fabbrini (2009). For the buried northern Apennines and Adriatic front, we used the fault datasets provided by Scrocca (2006), Cuffaro et al. (2010) and Fantoni and Franciosi (2010).

For the Puglia region, we used data from Patacca and Scandone (2004) and Del Gaudio et al. (2007), whereas, for the Calabrian Arc, we used data from Polonia et al. (2016) and Polonia et al. (2017).

Furthermore, we are aware of the DISS compilation of seismogenic sources in Italy (Basili et al., 2008; DISS Working Group, 2018) as one of the most important integrated datasets of the Italian territory in this field. However, we did not use this dataset in this work since, as described in Basili et al. (2008), the dataset aims to identify the "seismogenic sources" rather

than the actual faults. A composite seismogenic source (CSS in the DISS nomenclature) is a complex fault system showing homogeneous kinematic and geometric parameters and contains an unspecified number of aligned possible ruptures (i.e., faults) that cannot be isolated. Hence, a CSS, for its own nature, cannot be considered a real fault as declared in our scope.

The above-mentioned fault datasets form a tectonic image of the Italian territory (Fig. 2). These fault data and the spatial grid are available for download as ASCII files in the supplement (see also Petricca et al., 2018). The major thrust faults occur along:

(1) the N-verging western and north-western external front of the northern Alps; (2) the S-verging frontal ramp of the southern Alps (Po Plain and Veneto-Friuli regions); (3) the E- to N-verging external front of the central-northern Apennines from the Po Plain down to the central Adriatic off-shore; (4) the SE-verging outer front of the Calabrian Arc; (5) the outer (southern) front of the Maghrebian-Apennines chain in Sicily; and (6) the E-W-trending contractional belt located in the southern Tyrrhenian Sea, close to the northern Sicily coast.

The major normal faults occur along the median zone of the Apennines fold-thrust belt (Fig. 2), namely: (1) in northern Tuscany; (2) in central Italy including the Tuscany, Umbria, Marche, and Abruzzi regions; (3) in the southern Apennines including the Molise, Campania, Basilicata, and Calabria regions; and (4) in eastern Sicily including the Messina Straits, part of the Ionian Sea areas, and the Hyblean foreland. In particular, SW-dipping, high-angle normal faults (NW-striking) host the strongest seismicity recorded along the northern-central Apennines belt (Figs. 1 and 3). This fault pattern rotates to a NE-

trending direction toward the southern Calabria region consistently with the focal solutions of the area (Fig. 1). Moreover, extensional faults have also been mapped in the Sardinian region, specifically in the Campidano graben, mainly based on subsurface data (Casula et al., 2001).

Strike-slip faults are located in some areas of the Italian territory (e.g., Billi et al., 2003, 2007), in particular: (1) in the southern Alps (Veneto) with NNW-striking structures; (2) across the external front of the central-southern Apennines foreland-

fold-thrust belt (e.g., south of the Montemurro area, Fig. 1); (3) along the central Adriatic-Gargano-Molise belt with E-W-striking structures; (4) across the Calabrian Arc with radial structures cutting through the accretionary wedge; (5) in eastern Sicily from the Aeolian Islands (Tyrrhenian Sea) and southward into the Ionian Sea; and (6) in south-western Sicily and in the Sicily Channel with structures striking between N-S and NNE-SSW (Figs. 1 and 2).

## 3.2 Earthquake data

To obtain a complete earthquake dataset for the Italian territory (Figs. 1 and 3; supplement; Petricca et al., 2018), we integrated the existing most comprehensive catalogues of instrumental and historical seismicity: (1) the CSI1.1 instrumental database (csi.rm.ingv.it; Castello et al., 2006) for the 1981–2002 period; (2) the ISIDe instrumental database (iside.rm.ingv.it; Iside Working Group, 2016) for the 2003–2017 period; and (3) the CPTI15 historical-instrumental database (emidius.mi.ingv.it; Rovida et al., 2016) for the 1000-1981 period.

The CSI 1.1 database (Castello et al., 2006) is a catalogue of Italian relocated earthquakes for the 1997–2002 period. This collection derives from the work by Chiarabba et al. (2005), who relocated, using a homogeneous procedure, approximately 45,000 events provided by several seismological networks (both national, regional, and local) operating in the Italian territory. Most seismic events are lower than 4.0 in magnitude and are mostly located in the upper 12 km of the crust. A few earthquakes exceed magnitude 5.0, whereas the largest event is Mw 6.0. The time-span of this compilation is 1981–2002. From the CSI 1.1 database, we selected events with Mw > 4.0 (Fig. 3).

The ISIDe database (Iside Working Group, 2016) provides the parameters of earthquakes from real-time recordings and from the Italian Seismic Bulletin. The main aim of this database is to supply information on the seismicity as soon as it becomes available by integrating it with updated information on past seismicity. The time-span of this compilation begins in 1985 and lasts up to the present day. To avoid an overlap with the CSI database (1981–2002), from the ISIDe database, we considered events with Mw > 4.0 only from the 2003–2017 period (Fig. 3).

The CPTI15 historic database integrates the Italian macroseismic database version 2015 (DBMI15; Locati et al., 2016) and instrumental data from 26 different catalogues, databases, and regional studies (including the CSI and ISIDe databases) for the Italian territory, starting from 1000 A.D. until 2014. This catalogue provides moment magnitudes from macroseismic determination for more than 3200 earthquakes with values in the 4-7 range of Mw. To avoid any overlapping of data with the aforementioned instrumental datasets, we used data from the 1000-1981 period from the CPTI2015 catalogue.

We acknowledge that the earthquake magnitude used in this paper is the moment magnitude (Mw) and it should be noted that, for a few earthquakes from the aforementioned datasets, only the Ml (local magnitude) is available. However, according to Grünthal and Wahlström (2003), the difference between Mw an Ml can be ignored for magnitudes above 4, which represents the main focus of this study.

## 4 Method

### 4.1 FLEM computation

Starting from the entire dataset of faults in Italy, as a first step, we measured the length of each fault as the real fault trace length in map view, i.e., the length of the vertical projection of the fault trace as observed on the Earth's surface over a horizontal plane (Fig. 2; supplement; Petricca et al., 2018). Our complete dataset includes 12467 faults. Specifically, it includes 9169 A-class faults and 3298 B-class faults. Explanations for the classification into A- and B-class faults are given in the following sections.

As most faults have a horizontal length that is less than 25 km regardless of the selected database (Fig. 4), we divided the Italian territory into a grid with square cells of 25x25 km (Fig. 5; i.e., see also the explanation below). The length of the longest fault crossing each cell determined the parameter "fault length" (Lf) of the considered cell. In the second step, we used these lengths (Lf) as input parameters to empirically derive the earthquake magnitude (i.e., FLEM) of each cell containing at least one fault (Supplement; Petricca et al., 2018). Our only criterion to choose the fault from which the FLEM of the considered cell is computed is the greatest fault length in map view. In such a way, only one fault (the longest one) will provide the FLEM in a given cell. Therefore, faults cannot be and are not double-counted. Several studies investigated the scaling relationship between earthquake magnitudes and various geometric-kinematic parameters (i.e., fault dimensions and slip) of causative faults (e.g., Wells and Coppersmith, 1994; Leonard, 2010; Thingbaijam et al., 2017, and references therein), providing similar empirical equations. We used the equation proposed by Leonard (2010) (hereafter named as L10) that is expressed as:

$$FLEM = M = a + b * log(Lf) \tag{1}$$

where $a$ and $b$ are constants related to fault kinematics and are equal to 4.24 and 1.67, respectively (see table 6 in Leonard, 2010), and $Lf$ is the fault length as explained above. From Eq. 1, in our study, FLEM values range from 4.74 to 7.40, where the two end members are obviously imposed by the minimum and maximum fault length of the dataset, i.e., $\simeq 2$ and $\simeq 75$ km, respectively (Supplement; Petricca et al., 2018). Reasons for the choice of L10 are explained below.

## 4.2   Limits and assumptions

The method used in this study is based on some approximations and assumptions regarding the following points: (1) the arbitrary dimension of the cells (25x25 km); (2) the choice of the Leonard's equation and parameters (L10; Leonard, 2010) instead of others (e.g., Wells and Coppersmith, 1994); (3) the reliability of fault lengths; and (4) the assumption that all considered faults are active or can be potentially reactivated.

(1) Cell dimensions: as explained above, we choose a cell size of 25x25 km after considering the mean length (Figs. 4 and 5) of our fault dataset (Supplement; Petricca et al., 2018). Moreover, crustal earthquakes in Italy are mostly generated at a depth of 4-10 km. Within this depth range, the attitude of faults determines the distance between the fault trace and the epicentre over the Earth's surface. For instance, a hypocentral depth of 10 km along a causative fault dipping 45° implies a horizontal shift of about 10 km between the fault trace on the Earth's surface and the earthquake epicentre. Since the hypocentral depth is usually less than 10 km, a cell dimension of 25 km favours the occurrence of causative faults and related earthquake epicentres in the same cell. Moreover, to test the suitability of the size (25 km) of our square cells, we tested the sensitivity of the FLEM and the difference between the FLEM and the catalogued earthquake magnitudes on the cell size. We performed this sensitivity test on central Italy that is one of the most seismic areas of Italy. In this area, we changed our grid (i.e., cell size 25x25 km) both into a finer grid characterized by a cell size of 12.5x12.5 km and into a coarser grid characterized by a cell size of 50x50 km (Figs. S1 and S2). Each 50 km side square cell includes four 25 km side square cells and sixteen 12.5 km side square cells. For all cells, we computed the FLEM (Fig. S1) and the difference between the FLEM and the catalogued earthquake magnitudes

(Fig. S2). Then, cell by cell, we compared results (i.e., the FLEM) obtained for the largest and smallest grids (i.e., with square cell size of 12.5 and 50 km) against the FLEM obtained in the same geographical position using the intermediate grid (i.e., with square cell size of 25 km) (Fig. S3). This analysis shows that a change of grid size between 12.5 and 50 km does not provoke dramatic changes in the spatial distribution of both the FLEM (Fig. S1) and the difference between the FLEM and

the catalogued earthquake magnitudes (Fig. S2). In particular, using the largest grid (cell size = 50 km), the FLEM tends to be smoothed toward an upper boundary between magnitude 6 and 7 (Fig. S3a). This is due to the fact that only the longest faults of the dataset are considered for the FLEM computation. On the contrary, using the smallest grid (cell size = 12.5 km), the FLEM tends to be rather scattered and depressed around a magnitude 6 (Fig. S3a). This is due to the fact that reducing the cell size, in many cases, minor (small) faults result as being the longest ones in small cells. These results are also synthetically

expressed by the arithmetic averages of non-null FLEMs in the three different grids that are: $FLEM_{average}$ = 6.04, 6.19, and 6.62 for the grids with 12.5, 25, and 50 km sized cells, respectively. This sensitivity analysis shows that a microzonation (12.5 km sized cells) tends to locally overvalue small faults that are clearly poorly relevant when compared with adjacent (10-20 km distant) longer faults. On the contrary, a macrozonation (50 km sized cells) could overvalue long faults whose effect may be strongly reduced in a large area such as that included in a 50x50 km cell. Our choice (cell size = 25 km) appears therefore a

good compromise between significant faults and their potential areal influence.

     (2) Adopted equation parameters: The most popular scaling relationships that directly relate fault length and earthquake magnitude are provided by Wells and Coppersmith (1994), Leonard (2010), and Thingbaijam et al. (2017), among others. In this work, we used the equation (L10) by Leonard (2010) for dip-slip faults (both extensional and compressional). This equation is particularly suitable for Italy, where most earthquakes are generated by dip-slip faults (e.g., Chiarabba et al., 2005).

The main advantage of using Eq. 1 is that L10 is valid for both buried and outcropping fault as well as for normal and reverse (dip-slip) faults. To assess the difference (i.e., Fig. 6) between L10 and the equations provided by others, in Fig. 6a, we plotted the computed earthquake magnitude as a function of fault length for all scaling relationships by Wells and Coppersmith (1994), Leonard (2010), and Thingbaijam et al. (2017). Fig. 6b shows the difference between L10 and all other equations. For large fault lengths (i.e., corresponding to large earthquake magnitudes), the difference is mostly less than 5%. This difference increases

to about 15% for smaller faults (i.e., for fault length ≤ 2 km) corresponding to earthquake magnitudes less than about 4.8 (Fig. 6a). These fault lengths (≤ c. 2 km) and earthquake magnitudes (≤ 4.8) represent the lower boundary of this study. For this reason, we assessed the difference of results from L10 with results from other scaling relationships (Fig. 6) as acceptable for the aims of our study. In particular, also considering the version of L10 given for strike-slip faults, the difference with the version of L10 that we used (i.e., for dip-slip faults) is less than 3% (Fig. 6b). Note also that, due to the lower boundaries of the

magnitude range for which the equations considered in Fig. 6 are valid (i.e., Mw ≥ 4.8-5.0 forWells and Coppersmith (1994) and Mw ≥ 5.8 for Thingbaijam et al. (2017)), Fig. 6 is relevant for Mw higher than these lower boundaries.

     (3) Fault length reliability: FLEM is calculated through Eq. 1 from the longest fault falling in each cell (Fig. 5). For faults located on-shore and well exposed at the surface, detailed studies allow for a well-constrained and reliable characterization of fault length (class A faults). On the contrary, for some specific regions like the south-western Alps, the eastern front of the

Po Plain, the eastern Alps, the off-shore Adriatic front and Calabrian Arc, where fault planes are not exposed at the surface

(i.e., buried and/or off-shore faults), fault geometry has been mainly constrained from regional seismic reflection profiles or from earthquake sequences. Therefore, fault length and along strike continuity are not well constrained, producing the longest faults of the whole dataset. We assess the dimension of these latter faults as poorly constrained and reliable (class B faults). Following this analysis on the accuracy on fault dimension, we decided to divide the dataset into two quality classes. Class A (high quality) includes exposed faults where subsurface and surface data allow for a detailed and reliable characterization of fault length (light purple cells in Fig. 5), whereas class B (low quality) contains buried and off-shore faults investigated mainly by seismic surveys, for which a precise characterization of fault length cannot be achieved (dark grey cells in Fig. 5). Moreover, we acknowledge that FLEM (fault length earthquake magnitude) is indeed the maximum magnitude expectable from the actual size of the causal faults based on Eq. 1. A co-seismic lengthening of the causal faults through, for instance, the rupture of a bridge separating two adjacent faults to form one may produce earthquakes with a magnitude greater than that expected from the length of each of the two faults as measured before the fault co-seismic junction. Moreover, rupture jumps and coseismic slip distribution can happen during earthquakes thus producing earthquakes more energetic than what expected from the activation of a single fault or fault segment. An explicatory example in this sense is the Mw 7.8, 2016, Kaikoura earthquake, New Zealand, when the coseismic rupture nucleated as a weak strike-slip event along the Humps Fault. This rupture progressively moved northward onto a shallow contractional fault, where most seismic moment was delivered, before it activated slip on a further system of strike-slip faults at the northern tip of the rupture (e.g., Cesca et al., 2017). Another explicatory example of the uncertainty connected with the adopted method is the Fucino Fault in central Italy, which generated the destructive Mw 7.0 (estimated from damages and a maximum Mercalli intensity of XI) earthquake of 1915. The Fucino Fault in our database is 15.86 km long, corresponding to a FLEM (Mw) of 6.25. It is also true, however, that the 1915 coseismic rupture occurred along at least two parallel normal faults (Michetti et al., 1996). Hence, also the Fucino case shows that the method adopted in this work is poorly suitable in cases of multiple or complex coseismic ruptures.

(4) Fault orientation and activity: it is well known that each earthquake is usually associated with a precise slip on a fault plane and that the potential of a fault to undergo (seismic) slip depends on its orientation within the stress field (Morris et al., 1996; Collettini and Trippetta, 2007). It is also true, however, that the seismic history of the Earth is characterized by many examples of unexpected earthquakes occurring where plate boundaries are far away (i.e., intraplate earthquakes), and/or the stress field is apparently badly oriented to trigger earthquakes (Bouchon et al., 1998; Camelbeeck et al., 2007; Kafka, 2007; Stein and Mazzotti, 2007; Swafford et al., 2007; Braun et al., 2009; Boyd et al., 2013; Leonard et al., 2014; Talwani, 2014; Campbell et al., 2015; Walsh III and Zoback, 2016; Christophersen et al., 2017). Since our study aims to define the Fault-Length Earthquake Magnitude (FLEM) of each fault and cell, based on the aforementioned instances, we assumed that all faults are potentially reactivable. This notion becomes increasingly relevant when considering the prospective behaviour of faults over long terms. As mentioned above, indeed, for some societal challenges such as the safety of nuclear waste repositories, the recommendations are to consider the behaviour of faults in the future up to even 1 Ma (see, for instance, prescriptions by the Nuclear Energy Agency and International Atomic Energy Agency, www.oecd-nea.org, www.iaea.com; NEA, 2004; IAEA, 2016).

## 5   Results

Bearing in mind the limits mentioned above, we firstly calculated the FLEM for the Italian territory using the faults of class A (Fig. 7; supplement; Petricca et al., 2018). Most of the Italian territory, including the entire Apennine belt, the north-eastern Alps, and the central part of the Po Plain, is characterized by $6.0 \leq FLEM \leq 7.0$. The largest FLEMs have been obtained for the north-eastern Alps (FLEMs $\leq 7.4$) and are mainly related to the E-W-striking thrusts of the area responsible for the 1976 Friuli earthquake (Mw = 6.4; Bressan et al., 1998; Cheloni et al., 2012). In this area, we recall also the Carinthian great earthquake of 1348 (Villach, Mw 7.0; Rohr, 2003). In the central Alps, long and continuous thrust faults are reported in the ITHACA dataset (e.g., Maurer et al., 1997; Keller et al., 2006) and result in FLEMs of about 6.5. These large thrust faults also characterize the western part of the Po Plain, producing FLEMs of about 7.0. FLEMs between 6.5 and 7.0 are also estimated toward the south along the Apennine chain and in the Tuscany region, where large normal faults are reported from the ITHACA dataset and confirmed by detailed studies (Brogi et al., 2003; Brogi and Fabbrini, 2009). The largest FLEMs along the northern Apennines (Fig. 7) are due to the Alto Tiberina low angle normal fault, a large regional structure that seems to accommodate part of the deformation by aseismic creep and microseismicity (Chiaraluce et al., 2007; Collettini, 2011; Anderlini et al., 2016).

Large FLEMs occur also in the Vesuvius area (Campania region), where faults longer than 30 km have been reported in the ITHACA dataset (Fig. 5). Several studies (Finetti and Morelli, 1974; Scandone and Cortini, 1982; Vilardo et al., 1996; Brozzetti, 2011) confirm the presence of these long faults in the Vesuvius area. In the southern Apennines (Campania, Basilicata, and Calabria regions), FLEMs are in the range of 6.5-7.0, being related to the largest extensional faults of the area (Figs. 2 and 5). To the south, in the northern portion of Sicily, FLEMs > 7.0 are related to long off-shore faults. Some of these FLEMs are related to the transtensional-transpressional Tindari Fault System, located in NE Sicily. Other FLEMs are connected to the extensional system of the Messina Straits (Ghisetti, 1979; Locardi and Nappi, 1979; Lanzafame and Bousquet, 1997; Billi et al., 2006b, 2007; Palano et al., 2012; Cultrera et al., 2017).

The map of FLEMs derived using class B faults (Fig. 8) shows seismic events in the magnitude interval of 5.0-6.0 in the Tyrrhenian sector and in the southern portions of Puglia. Large FLEMs (up to M $\simeq$ 7.85) occur along the north-eastern Alps, the thrust fronts beneath the Po Delta, the north Adriatic front, the Ionian off-shore of the Calabria-Peloritani Arc, and some areas of Sardinia (Fig. 8). FLEMs $\simeq$ 8.0 derive from structures constrained only by subsurface data, for which the along-strike continuity cannot be properly assessed and, hence, the total length is likely to be largely overestimated. The largest FLEMs calculated for the Sardinian territory are related to structures inferred from large-scale (1:500,000) maps. These structures are longer (Fig. 5, red segments) than those derived, in the same areas, from detailed studies (Fig. 5, pink segments). Therefore, the actual length is likely to be overestimated in this case.

To compare the FLEMs estimated on the grounds of geological fault length (Figs. 7 and 8) with the maximum magnitudes obtained from the historical and instrumental seismicity databases (i.e., catalogued earthquake magnitudes), we used the same grid presented above (Fig. 5). In particular, for each cell of Fig. 5, we selected the maximum earthquake magnitude recorded in historical or instrumental earthquake databases, applying a lower cut-off at magnitude 4.0 (Fig. 9). Large earthquake magnitudes - i.e., M > 6.0 earthquakes - were recorded in the north-eastern Alps, in the Po Plain, and along the entire Apennine

chain. The strongest events (M ≤ 7.4) were recorded in the north-eastern Alps and in the southern portion of the Apennines, including the Messina Straits and southern Sicily (Fig. 9). Earthquakes with magnitudes ≃ 4.0-4.5 occurred almost everywhere in the Italian territory.

To spatially compare the earthquake magnitudes obtained through the FLEM computation (Figs. 7 and 8) and those recorded in the historical-instrumental catalogues (Fig. 9), we realized a map of the earthquake magnitude differences to show for each cell the difference between FLEMs obtained from the length of class A (Fig. 10) and class B (Fig. 11) faults and the magnitudes of historically/instrumentally-recorded earthquakes. In the case of class A faults, this comparison (Fig. 10) shows, in general, a difference of M ≤ 1.5. By fitting the differences distribution with a Gaussian curve, we obtained a mean of 0.86 with a $2\sigma$ (double standard deviation) of 1.47 on FLEMs derived from class A faults (Fig. 12a). The $2\sigma$ value is amplified to 1.85 (Fig. 12b) when also considering FLEMs from poorly reliable fault lengths (i.e., difference for class A+B faults in Fig. 12b).

Finally, we compared FLEMs with only the strongest earthquakes (M ≥ 6.5) in the historical and instrumental catalogues since 1000 A.D. to 2017 (Fig. 13 and also Table 1). In this case, the difference with the spatially-corresponding FLEMs (i.e., same cell) is less than 1.0 and, in most instances, less than 0.4 (Fig. 13a). In particular, the mean difference is -0.09 with a $2\sigma$ value of 0.53 (Fig. 13c). Note also that, even when not equal to the spatially-corresponding catalogued earthquake magnitudes, yet many FLEMs fall within the uncertainty interval (i.e., in Fig. 13b, see the red circles falling along the blue vertical bars) associated with these catalogued magnitudes.

In the histograms of Figs. 12 and 13(c), the values in the negative fields can be interpreted as FLEM's underestimation (i.e., where FLEM values do not reach the catalogued earthquake magnitudes, which are in turn affected by uncertainty; e.g., Fig. 13b). In contrast, values in the positive fields of Figs. 12 and 13(c) could be interpreted either as FLEM's overestimation (i.e., where FLEM values unsuitably exceed the catalogued earthquake magnitudes) or as a sort of catalogue incompleteness (i.e., where the catalogued earthquake magnitudes do not properly represent larger seismic potentials of faults). Occurrences in the positive fields (green portions in Figs. 12 and 13c) are more frequent than occurrences in the negative fields (red portions in Figs. 12 and 13c), particularly in Fig. 12. These positive and negative occurrences are also mapped in Figs. 10 and 11 with green and red tones, respectively, whereas white cells are where FLEM's differences with respect to catalogued earthquake magnitudes are about null. Figs. 10 and 11 show that negative (red) occurrences are very limited.

It is also interesting to note that, in the period between our dataset of catalogued earthquakes (i.e., in times younger than December 2017) and the time of writing (September 2018), only one crustal M>5 earthquake occurred in Italy. Namely, the Montecilfone Mw 5.1 earthquake occurred on August 16th, 2018, in the central Apennines at latitude 41.87N and Longitude 14.86E and depth of 20 km. In the same locality, our FLEM is 5.9, corroborating the observations and considerations made above on the difference between FLEMs and catalogued earthquake magnitudes.

Summarizing, large differences between FLEMs and catalogued earthquakes can be either real, indicating that some large earthquakes, possibly due to extremely long recurrence intervals are not contained in the seismological records, or a bias induced mainly by the following factors: (1) impossibility to resolve fault continuity and segmentation with the adopted method and (2) deformation partially accommodated through aseismic creep.

Small differences, as discussed above, between FLEMs and the catalogued earthquake magnitudes (Figs. 12a and 13c) are due to the comprehensive knowledge of the exposed faults (particularly faults of class A, Fig. 12a). A large amount of data is indeed available from detailed field surveys and subsurface investigations realized over the years. However, some portions of the eastern Alps, northern Apennines, and southern Italy, including Sicily, show a difference in the magnitude of 2.0-2.5

(i.e., faults of class B, Fig. 12b). In these areas, more detailed studies should be developed in order to better characterize fault dimensions and properly assess seismic hazards. It is, however, very encouraging that when considering only the historical and instrumental earthquakes with M ≥ 6.5, the difference between FLEM values and the catalogued earthquake magnitudes reduces (Fig. 13).

Regarding the FLEMs evaluated using class B faults (Fig. 11), we observe a significant difference with catalogued earth-

quakes in several regions. Differences of up to 4.0 in magnitude (Fig. 12b) are estimated in correspondence of the north Adriatic thrust front, off-shore from the Calabria-Peloritani Arc, and the northeast part of the Alps (Fig. 11). Smaller, but still relevant, differences of $M \simeq 2.0$ are documented in correspondence of the Apennines front beneath the Po Plain. As mentioned above, class B fault lengths have been evaluated mostly using seismic reflection profiles. Due to the resolution of this technique and the quality of some datasets, fault segmentation cannot be properly evaluated. More evidence would be needed. In general, the

significant differences between FLEMs (class B faults) and the catalogued magnitudes require that extensive 3D geological and geophysical investigation of the structures of these areas should be performed in order to better characterize the geometry and continuity of faults. To this end, it is also noteworthy that studies on the 2016 Amatrice-Norcia (central Italy) earthquakes (Mw 6.0 and 6.5) revealed that the length of the causative faults was only partially activated by the seismogenic slip (e.g., Chiaraluce et al., 2017); however, as this fault-slip behaviour seems rather frequent (Freymueller et al., 1994; Milliner et al.,

2016; Chousianitis and Konca, 2018), it is most likely that this same behaviour is incorporated and implicitly expressed by the above-mentioned empirical scaling relationships between fault length and earthquake magnitude (e.g., Wells and Coppersmith, 1994; Leonard, 2010; Thingbaijam et al., 2017).

## 6   Statistical Analysis of Results

The reliability of estimated FLEM values may be quantitatively checked, in principle, using a formal statistical test on earth-

quake catalogues. Anyway, models of expected maximum magnitude (Mmax) are not readily testable. A statistical test helping to discriminate between competitive FLEM values is impossible in practice, even in the simplest case of discriminating between a double truncated Gutenberg-Richter (DTGR) law and an unbounded Gutenberg-Richter (UGR, Mmax=∞) law on excellent datasets (Holschneider et al., 2014). Suffice is to say that the DTGR's probability distribution differs from the UGR's one by a constant $c = 1 - 10^{-b(M_{max} - M_c)}$ (where $M_c$ is the minimum magnitude). This value is very close to 1, if not when

$M_c$ is close to Mmax. This means that the upper cut-off of the Gutenberg-Richter distribution can be explored using only large, very rare earthquakes, with magnitudes close to the maximum possible value, on which statistical inference is necessarily limited (Holschneider et al., 2014). Therefore, it is essentially impossible to statistically infer, with sufficient confidence, the maximum possible earthquake magnitude, in terms of alternative testing, from an earthquake catalogue alone (Holschneider

et al., 2014). In light of the limitations of purely statistical inference, geological and tectonic information provides, therefore, important and exclusive constraints on the expected maximum magnitude. If we look at statistical testing in detail, we can check the FLEM values on a catalogue, while controlling the probability of wrongly rejecting them, but without reducing the probability of a wrong non-rejection (Holschneider et al., 2014). This makes it, indeed, impossible to discriminate between two likely values of FLEM. In other words, if we do not reject a FLEM value, we cannot say if it is true or if the data are inadequate in terms of revealing its failure. Keeping all these considerations in mind, we test the estimated FLEM values on the CSI1.1 and ISIDe databases (available online at https://csi.rm.ingv.it/ and http://iside.rm.ingv.it/, respectively). Since our analysis requests a database containing small magnitudes events, we consider ISIDe data of earthquakes occurred before 16 April 2005, when the new INGV (Istituto Nazionale di Geofisica e Vulcanologia) National Seismic Network, completely re-organized and equipped with a new acquisition system, became operational (Amato and Mele, 2008). Without this separation, the hypothesis of temporal homogeneity for magnitude data would not be appropriate. Moreover, most events in the ISIDe and CSI1.1 databases have a ML magnitude, while FLEM values are in moment magnitude scale. Therefore, we repeat the same testing procedure, described below, twice, without any difference in results: firstly, we consider original FLEM values and, secondly, we convert them in local magnitude scale through the relations proposed by (Gasperini et al., 2013). We adopt the following test procedure.

a) We select cells for which a test is helpful, i.e. having a FLEM value above the related historical/instrumental observed magnitude. In this way we exclude 41 of 1100 cells for which FLEM values are estimated. A further selection consists of keeping only cells for which the Gutenberg-Richter model, with a b-value equal to 1, is not rejected. In this way, we are sure that a possible rejection of FLEM values cannot be ascribed to the low reliability of a Gutenberg-Richter relation or to wrong b-values. To this end, we apply a goodness of fit test (GFT, Wiemer and Wyss, 2000), at a 95% confidence level, that also provides a completeness magnitude value, $M_c$, for each cell. The GFT is based on difference R of the observed ($O_i$) and expected ($E_i$) numbers of events in each magnitude bin. Values $E_i$ are computed by assuming a UGR distribution, with a b-value equal to 1, above an ascending magnitude cutoff $Mt$. So that R is given by:

$$R(M_t) = 1 - \frac{\sum_i |E_i - O_i|}{\sum_i O_i} \tag{2}$$

where sums are done for magnitude bins above $M_t$. The completeness magnitude $M_c$ is defined as the first value of $M_t$ for which R($M_t$)>0.95. The cells passing this test are 30 (Table S1) and 67 (Table S2) for the CSI1.1 and ISIDe databases, respectively. We stress that this first test does not require a Mmax value.

b) A first specific check of the DTGR distribution, having FLEM as maximum magnitude, is carried out by using a log-likelihood test. Specifically, for each cell passing the previous test, the log-likelihood value of the DTGR model is computed. By considering that a DTGR distribution has a probability density function given by:

$$f_{DTGR}(M) = \frac{\beta e^{\beta(M - M_c)}}{1 - e^{\beta(Mmax - M_c)}} \tag{3}$$

where $b = \beta ln(10)$; we compute for each cell the real log-likelihood $LL_R$ on N events magnitudes $M_i, i = 1, .., N$, by equation:

$$LL_R = \sum_{i=1}^{N} ln[f_{DTGR}(M_i)] \tag{4}$$

The DTGR model is rejected if $LL_R$ is significantly lower than expected values. The probability of having smaller log-likelihood values than $LL_R$ is estimated using $10^4$ simulated datasets (having the same size of real datasets) by model DTGR. Specifically, for each cell, we compute this probability pLL as the proportion of simulated log-likelihoods smaller than $LL_R$. We select cells for which these probabilities are larger than 5%. In this way, the cells are reduced to 20 and 45 for the CSI1.1 and ISIDe databases, respectively (Figure 14 and Tables S1 and S2). We stress that we select the same cells by assuming a UGR model, suggesting that the problem of excluded cells lies in the whole Gutenberg-Richter relation and not in the FLEM values.

c) Finally, FLEM values are tested by applying the procedure proposed by (Holschneider et al., 2014), involving a comparison of the maximum observed magnitude with a suitable threshold magnitude, computed by the DTGR model. Specifically, we reject the FLEM values, if the observed magnitude (oMmax) is larger than a threshold value $M_t$, given by:

$$M_t = M_c - \frac{1}{\beta} ln\{1 - (1-\alpha)^{\frac{1}{n}}(1 - e^{-\beta(FLEM-M_c)})\} \tag{5}$$

where $\alpha = 0.05$ (see Holschneider et al., 2014, for details). The computed threshold values ($M_t 1$) are listed in Tables S1 and S2, together with oMmax values. All previously selected cells pass this test, for both the CSI1.1 and ISIDe databases, suggesting that related FLEM values cannot be rejected (Figure 14 and Tables S1 and S2).

d) The last two steps are repeated by reducing the magnitude range covered by data. Specifically, we increase, where possible, the minimum magnitude to FLEM-2.0. The relative threshold values $M_t 2$ are listed in Tables S1 and S2 for the CSIv1.1 and ISIDe databases, respectively. Also in this case, we cannot reject the FLEM values. As stated above, this analysis does not exclude alternative values of FLEM. As matter of fact, it is also the case that a UGR model cannot be rejected in all the selected cells.

## 7  Conclusions

In this study, (1) we first provided an updated compilation of a comprehensive dataset of mapped faults in Italy, (2) then, using known scaling laws, calculated the related Fault-Length Earthquake Magnitude (FLEM), and, (3) lastly, compared FLEM values with historically/instrumentally-catalogued earthquake magnitudes. Where faults are geologically well constrained (class A faults), either agreements or differences are observed between FLEMs and historically/instrumentally-catalogued earthquake magnitudes: agreements increase for M $\geq$ 6.5 earthquakes. In areas where fault geometries are poorly constrained (class B

faults inferred solely by subsurface 2D investigations), larger differences are observed. These areas have to be further charac-terized to better estimate fault dimensions and hence properly assess the FLEM. Our results are partly encouraging and suggest the testing and validation of this experiment elsewhere. This method cannot, however, be a substitute for time-dependent (pa-leo)seismological methods for seismic hazard assessments. Rather, it can highlight areas where further detailed studies on

faults are required.

*Data availability.*   Copyrighted catalogue data are available online at csi.rm.ingv.it, iside.rm.ingv.it and emidius.mi.ingv.it/CPTI15-DBMI15. Original data are available at http://doi.org/10.5880/fidgeo.2018.003 (Petricca et al., 2018). For copyright reasons, the related vector data are available only for personal scientific use upon request to the authors. Alternatively, some vector data are available online at sgi.isprambiente.it/geoportal. Further data are available in the supplement associated to this paper.

*Author contributions.*   CD obtained the funding; all authors conceived the experiment and the paper; PP, AB, FT, DS, MC, and CC performed the experiment and analyzed the data under the coordination of CD and GV; all authors discussed the results and drew the related conclusions; FT, AB, and PP wrote most part of the manuscript; PP realized the figures; all authors reviewed and accepted the manuscript and figures.

*Competing interests.*   The authors declare that they have no conflict of interest.

*Acknowledgements.*   Most figures were produced using the GMT software (http://gmt.soest.hawaii.edu/). Pierfrancesco Burrato is thanked for
providing the digital version of the Structural Model of Italy. A special thanks to Federica Riguzzi for fruitful discussion during the project. Copyrighted catalogue data are available online at csi.rm.ingv.it, iside.rm.ingv.it and emidius.mi.ingv.it/CPTI15-DBMI15. Original data are available at http://doi.org/10.5880/fidgeo.2018.003 (Petricca et al., 2018). For copyright reasons, the related vector data are available only for personal scientific use upon request to the authors. Alternatively, some vector data are available online at http://sgi.isprambiente.it/geoportal. Further data are available in the supplement associated to this paper. We warmly thank F. Rossetti, Y. van Dinther, S. Nandan, K.K. Thingbai-
jam, an anonymous reviewer, and other colleagues (Valensise et alii, see the Discussion Forum of Solid Earth) for constructive comments. The presented results should be considered more in a theoretical and methodological perspective for comparison with future similar studies rather than in an applicative perspective for the case of Italy. In particular, our assessed earthquake magnitudes (FLEMs) for the Italian territory are proposed in this paper for scientific reasons and not for their use for civil protection and prevention purposes. Moreover, in this article, we do not address or estimate the probability of earthquake occurrence. Yet, we would like to acknowledge that some large magnitudes of
earthquakes (FLEMs), calculated in this article are considered very unlikely in the existing literature of seismic hazard in Italy.

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

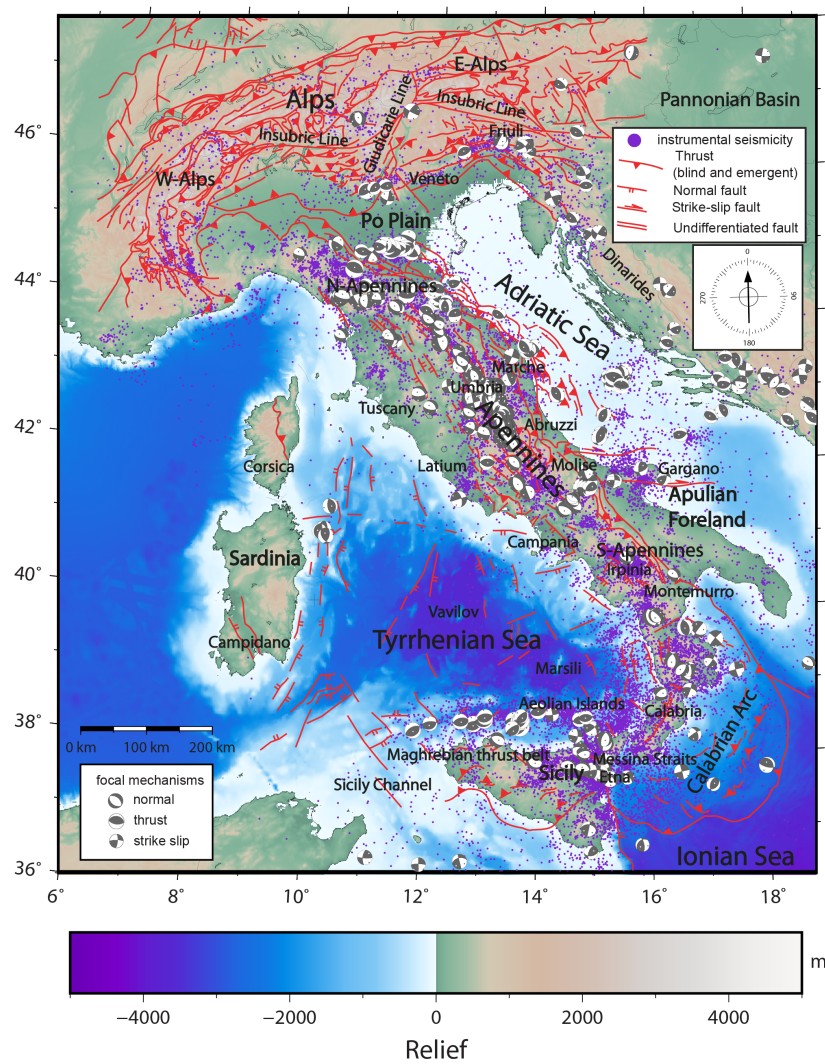

**Figure 1.** Tectonic map of Italy with instrumentally-recorded earthquakes (epicentres of M ≥ 2.0) in the 1981-2017 interval (ISIDe and CSI1.1 databases). Focal mechanisms are for M ≥ 4.0 events (Pondrelli et al., 2006), showing extensional tectonics along the Apennine chain and compressional tectonics in the NE portion of the Alps and in the N-NE portion of the Apennines.

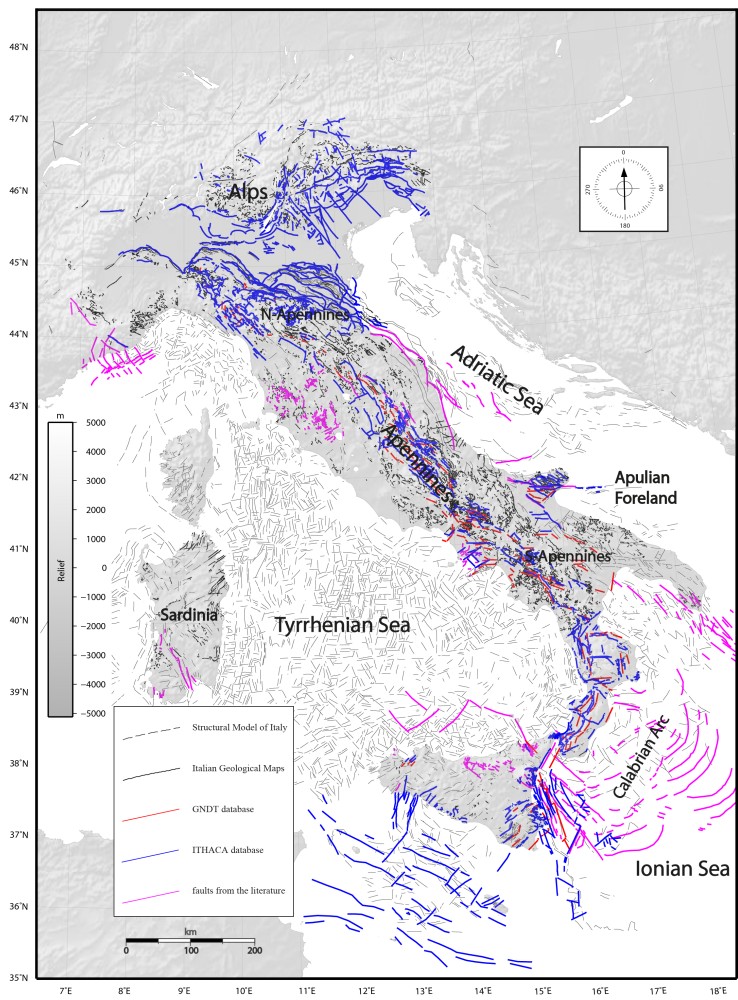

**Figure 2.** Map of faults from the five datasets used in this work: the Structural Model of Italy at the 1:500,000 scale (dashed black; Bigi et al., 1989); the Italian Geological Maps at the 1:100,000 scale (black; available online at www.isprambiente.it); the GNDT database of active faults (red; Galadini et al., 2000); the ITHACA database (blue; Michetti et al., 2000); and selected active faults from complementary studies published by different authors (pink; see text for explanations).

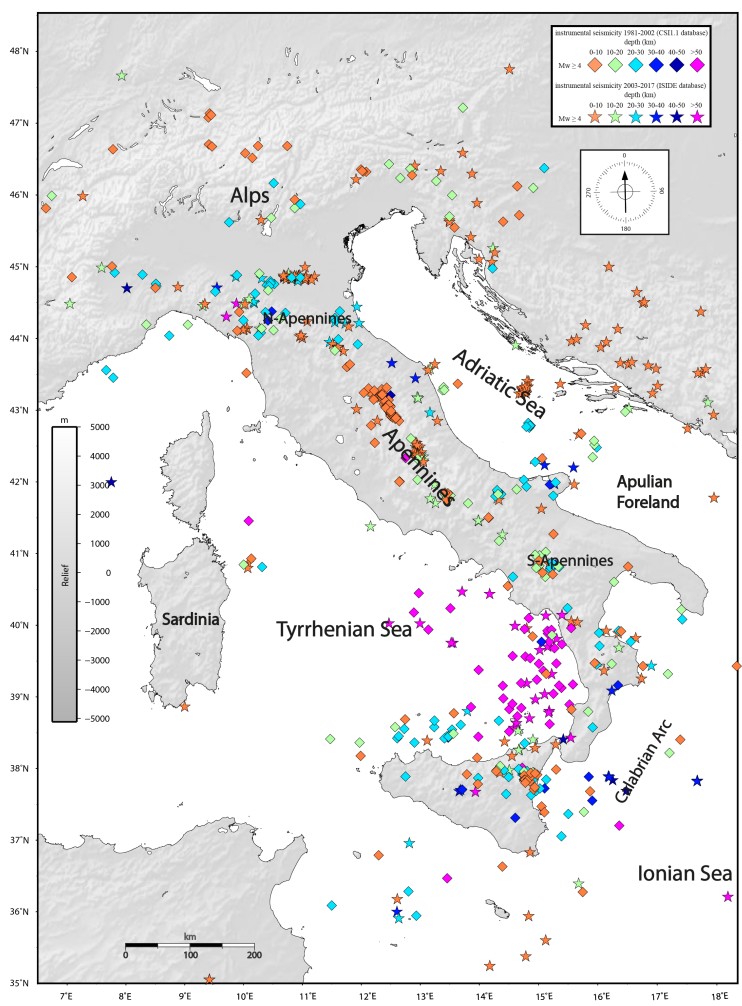

**Figure 3.** Instrumental seismicity (epicenters of M ≥ 4.0 earthquakes) distribution in Italy from ISIDe (stars) and CSI (diamonds) datasets. The CSI1.1 database (http://csi.rm.ingv.it; Castello et al., 2006) reports events in the 1981-2002 time interval, whereas the ISIDe database (http://iside.rm.ingv.it/iside/standard/index.jsp; Iside Working Group, 2016) reports events in the time interval 2003-2017. Events are differentiated by depth (change in color every 10 km).

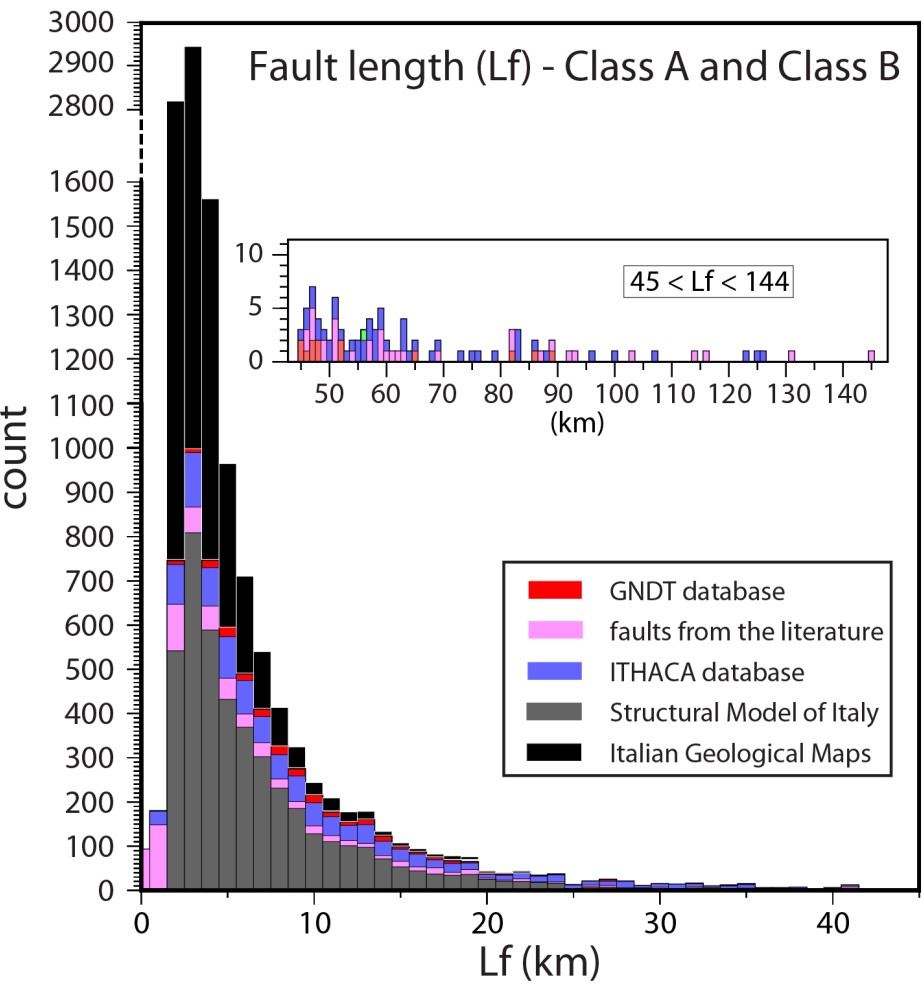

**Figure 4.** Histogram of fault lengths (Lf) derived from all analyzed datasets. Inset shows the same histogram for fault lengths ≥ 45 km.

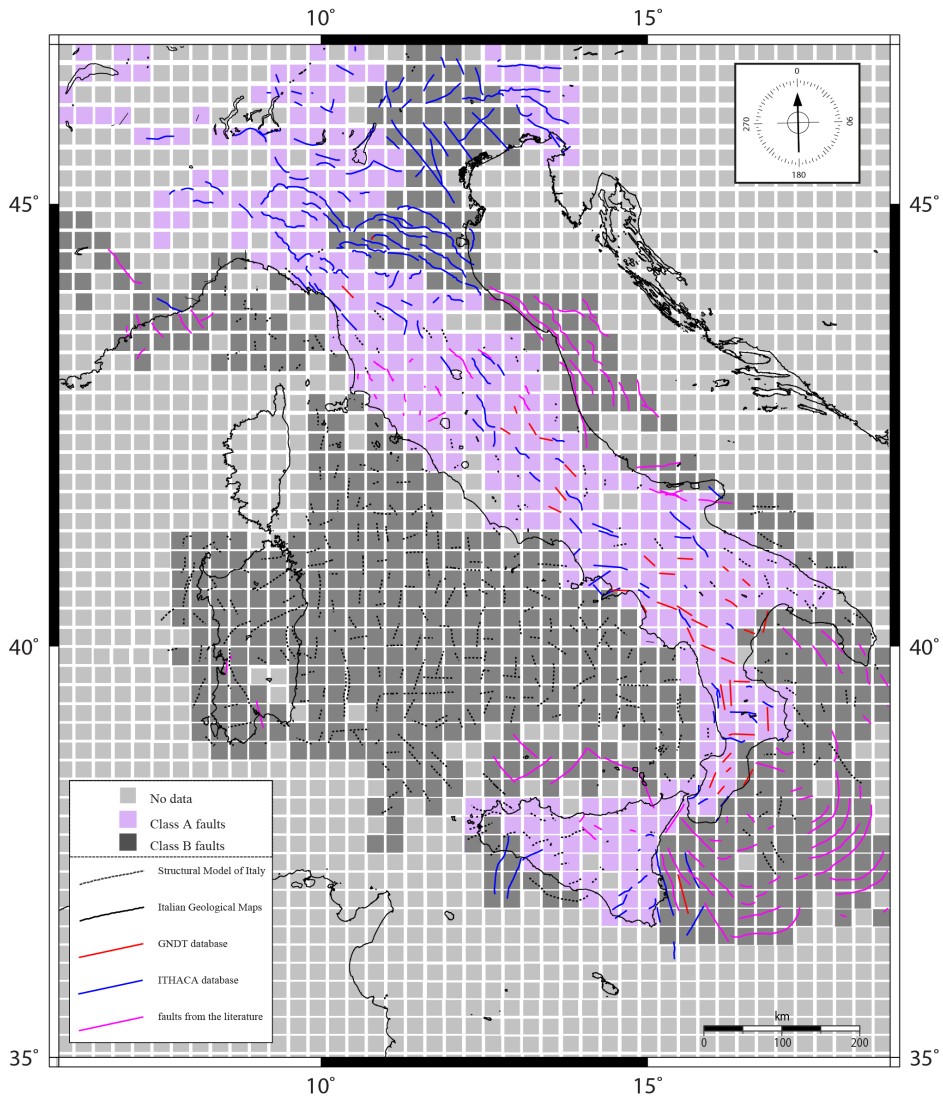

**Figure 5.** Map of the Italian territory divided into a grid with 25x25 km square cells. The map shows also the longest faults (from the analyzed datasets) falling into each cell. Light purple cells are for class A faults whereas dark grey cells are for class B faults.

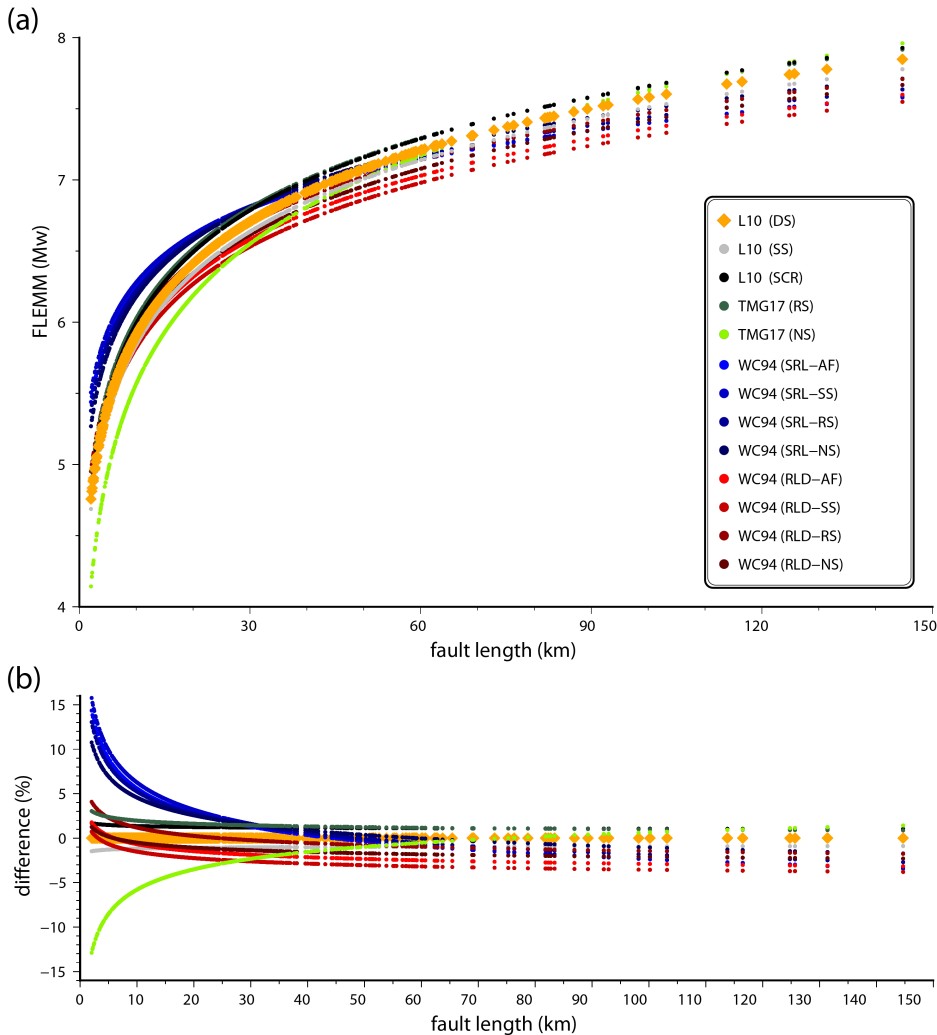

**Figure 6.** Earthquake magnitude vs. fault length, obtained using empirical scaling laws from: L10 = Leonard (2010); WC94 = Wells and Coppersmith (1994); and TMG17 = Thingbaijam et al. (2017). Keys: DS = dip slip faults; NS = normal slip faults; RS = reverse slip faults; SS = strike-slip faults; SCR = stable continental region faults; AF = all faults; SRL = surface rupture length; and RLD = sub-surface rupture length. (b) Difference vs. fault length. The difference is between earthquake magnitudes obtained from L10 vs. each of all other scaling laws (namely, WC94 and TMG17).

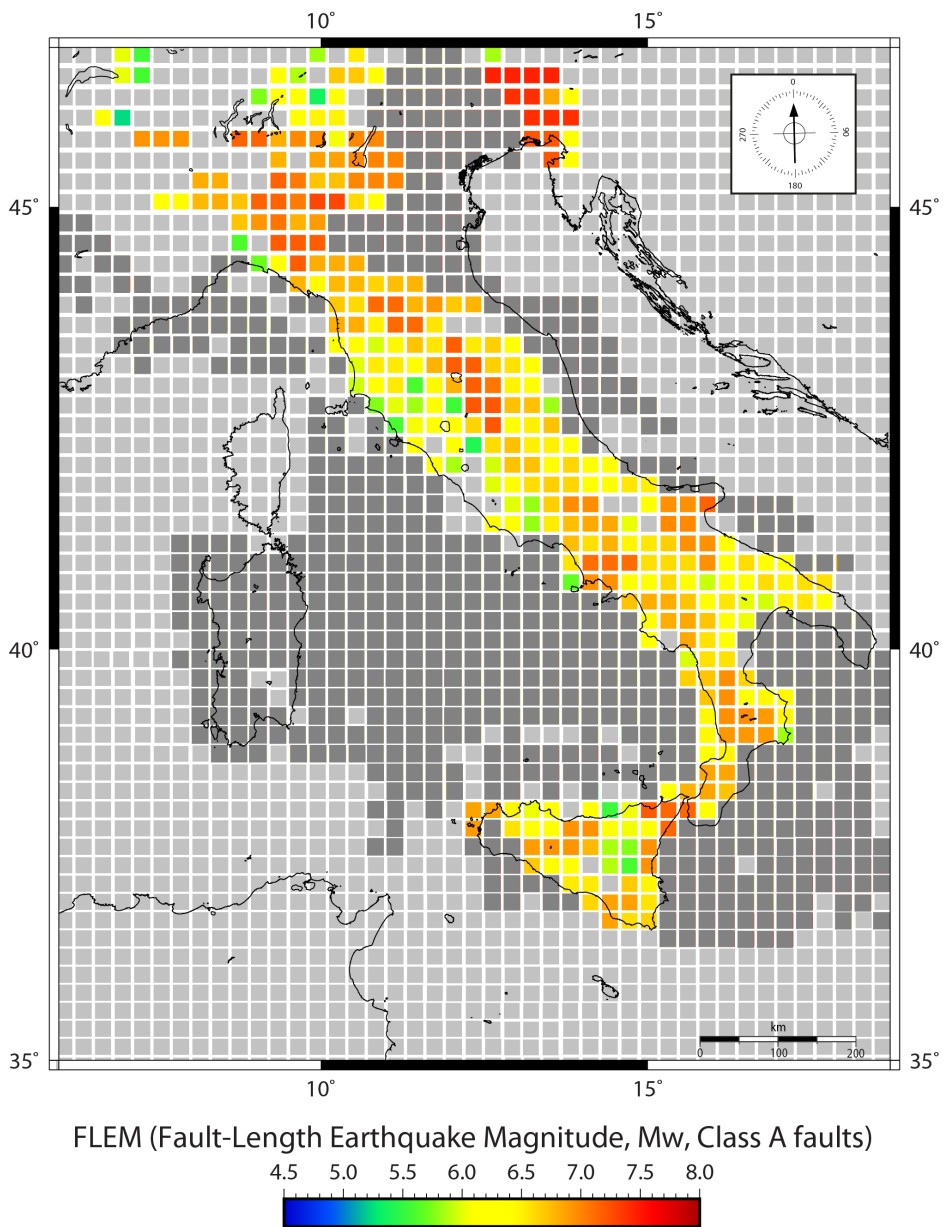

**Figure 7.** Map of the Fault-Length Earthquake Magnitude (FLEM) calculated, for each cell, from the length of class A faults. FLEM values are obtained using L10. Light grey is for cells with no data whereas dark grey is for cells where the longest faults belong to the class B set. Note that these results should be considered more in a theoretical and methodological perspective for comparison with future similar studies rather than in an applicative perspective for the case of Italy. In particular, our assessed earthquake magnitudes for the Italian territory are here proposed for scientific reasons and not for their use for civil protection and prevention purposes.

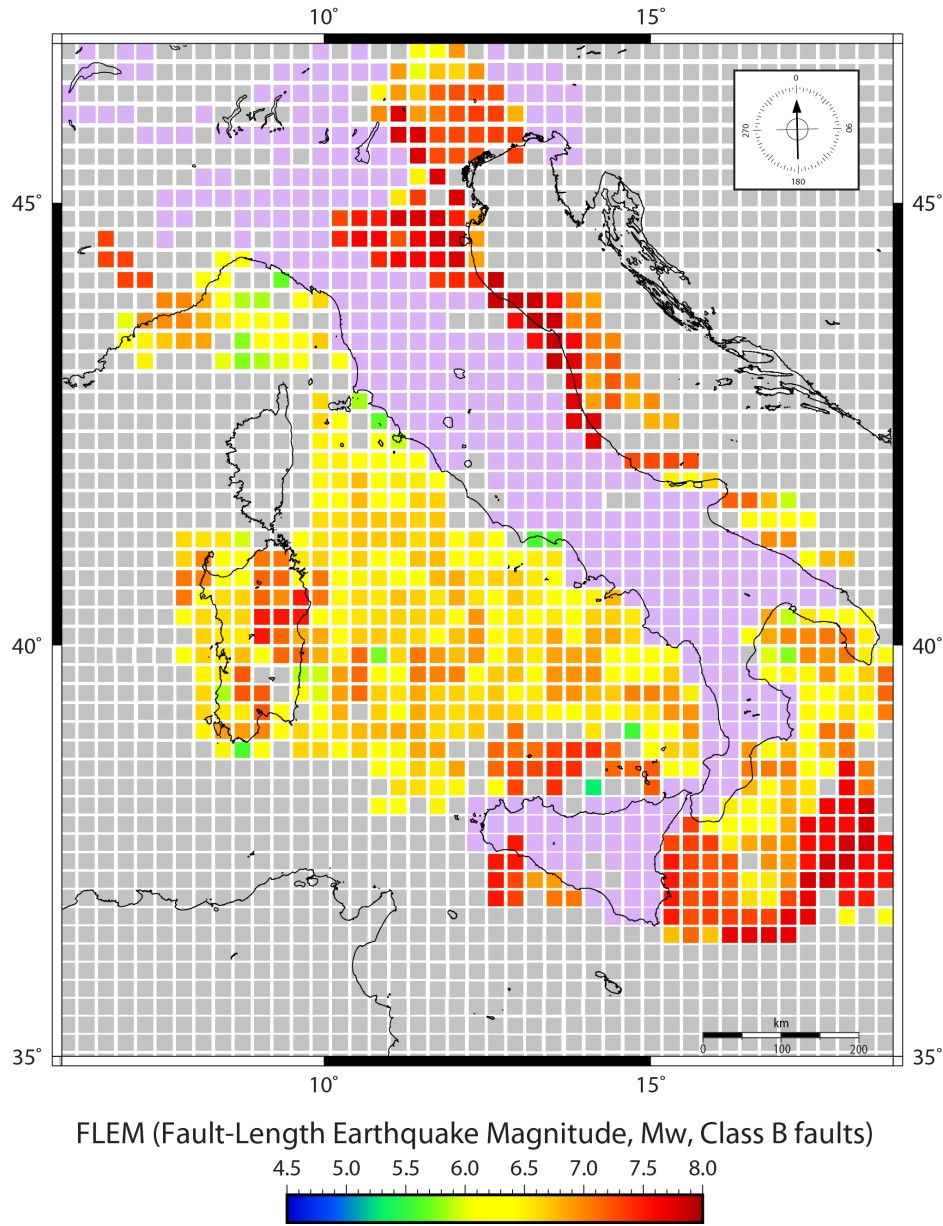

**Figure 8.** Map of the Fault-Length Earthquake Magnitude (FLEM) calculated, for each cell, from the length of class B faults. FLEM values are obtained using L10. Light grey is for cells with no data whereas light purple is for cells where the longest faults belong to the class A set. Note that these results should be considered more in a theoretical and methodological perspective for comparison with future similar studies rather than in an applicative perspective for the case of Italy. In particular, our assessed earthquake magnitudes for the Italian territory are here proposed for scientific reasons and not for their use for civil protection and prevention purposes.

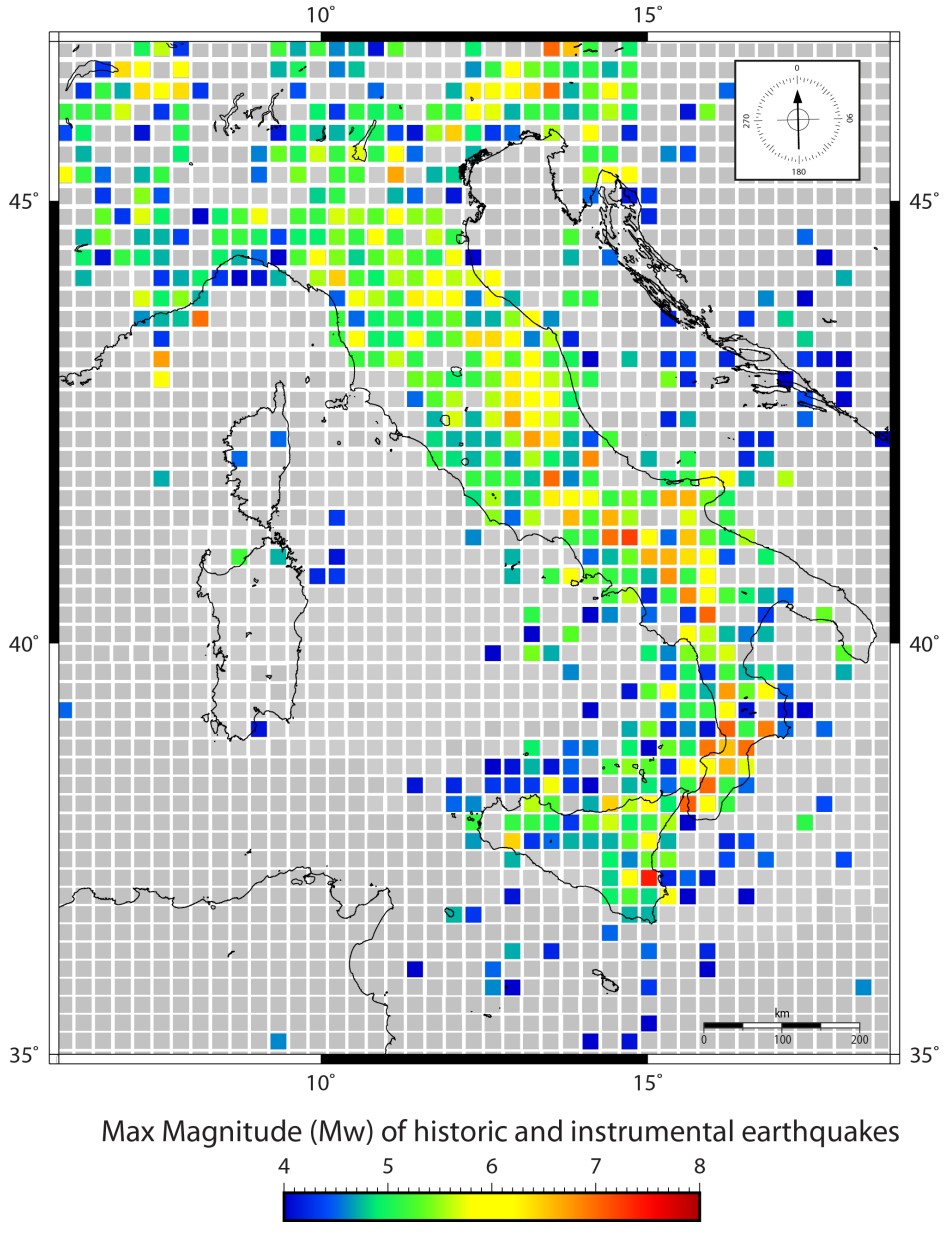

**Figure 9.** Map of maximum magnitude (for each cell) from historic or instrumental earthquakes as recorded in the CSI 1.1, ISIDe, and CPTI15 databases within the 1000-2017 period. Grey is for cells with no data.

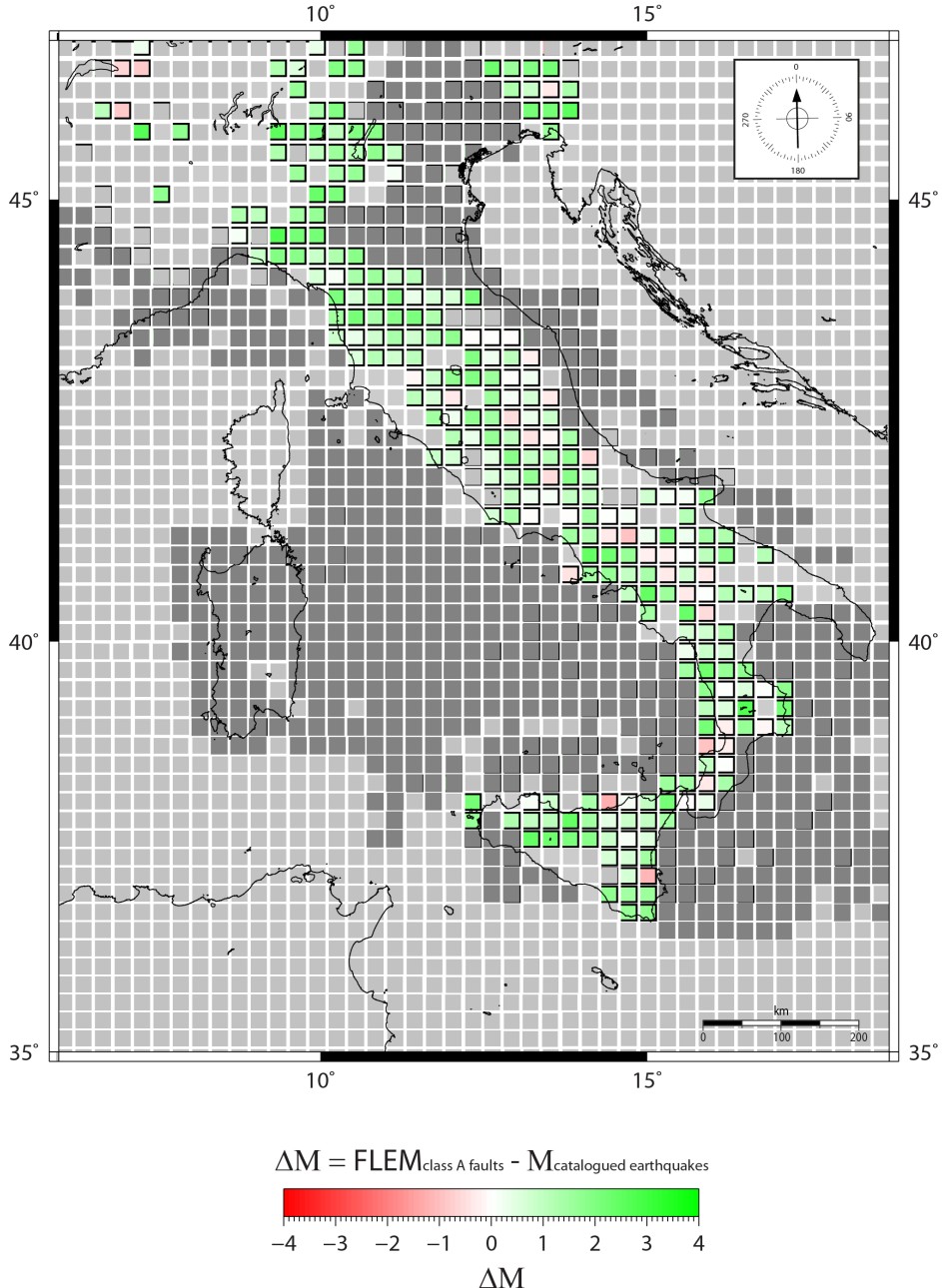

**Figure 10.** Map of the difference between FLEM values (only from class A faults; Fig. 7) and maximum magnitudes from historic or instrumental earthquakes (Fig. 9). Light grey is for cells with no data whereas dark grey is for cells where the longest faults belong to the class B set. See Fig. 12 for the interpretation of red vs. green cells.

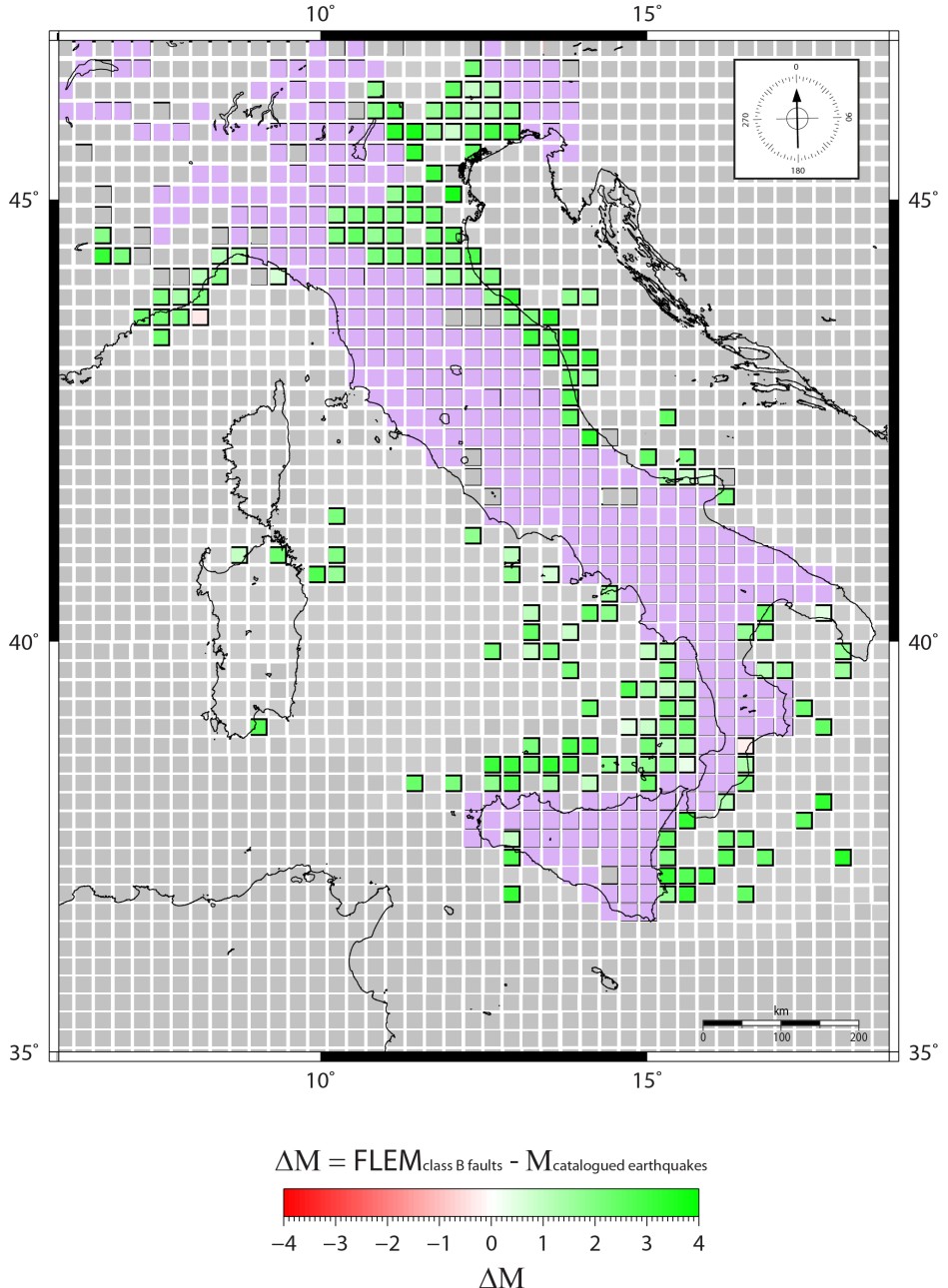

**Figure 11.** Map of the difference between FLEM values (only from class B faults; Fig. 8) and maximum magnitudes from historic or instrumental earthquakes (Fig. 9). Light grey is for cells with no data whereas light purple is for cells where the longest faults belong to the class A set. See Fig. 12 for the interpretation of red vs. green cells.

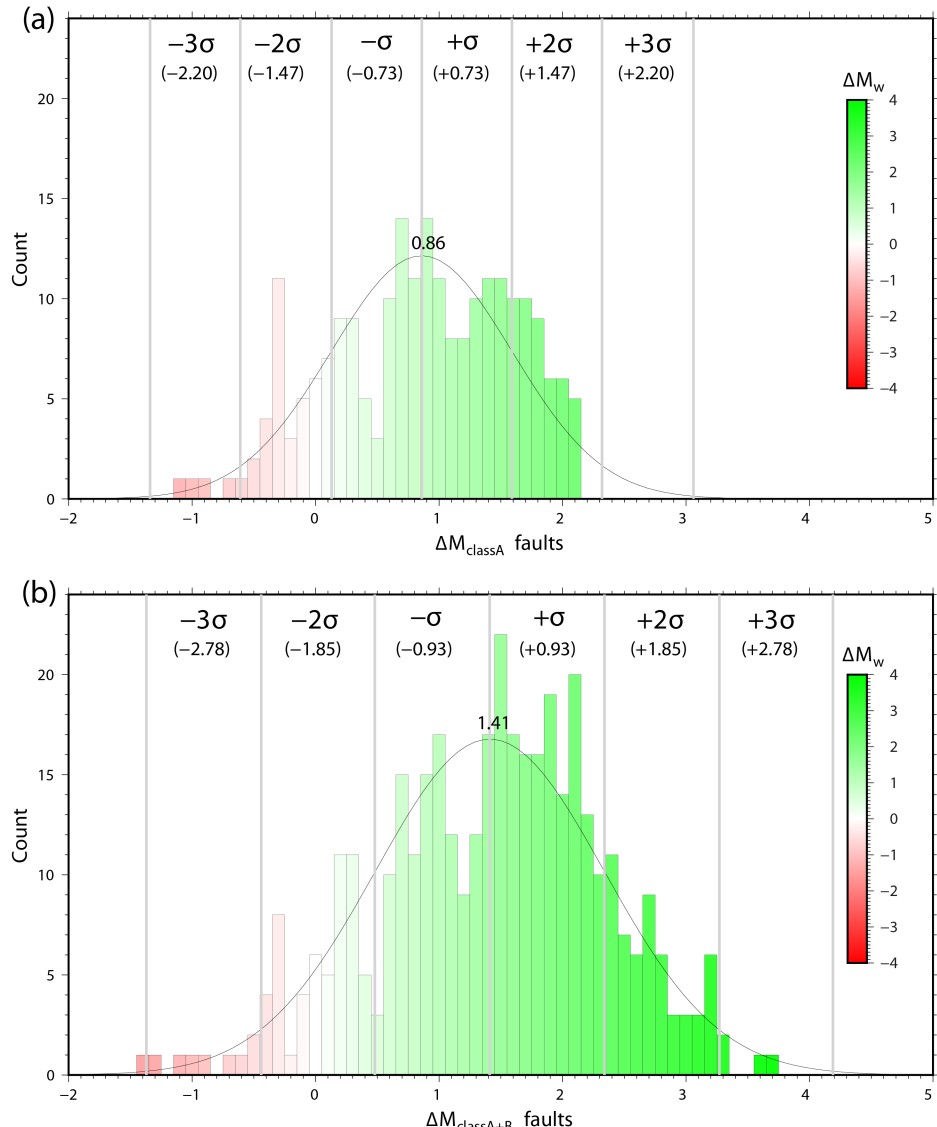

**Figure 12.** (a) Histogram of the difference between FLEMs (from class A faults) and spatially-coincident (same cell) historical/instrumental-catalogued magnitudes of earthquakes. Mean = 0.86; deviation standard $\sigma = 0.73$. (b) Histogram of the difference between FLEMs (from class A and B faults) and spatially-coincident (same cell) historical/instrumental-catalogued magnitudes of earthquakes. Mean = 1.41; deviation standard $\sigma = 0.93$. In both histograms, values in the negative fields can be interpreted as FLEM's underestimation with respect to catalogued earthquake magnitudes whereas values in the positive fields could be interpreted either as FLEM's overestimation with respect to catalogued magnitudes or as a sort of catalogue incompleteness. Occurrences in the positive fields (green portions) are significantly more frequent than occurrences in the negative fields (red portions). Green and red colors in Figs. 10, 11, and 13(c) have the same meaning as in this figure. See text for further information.

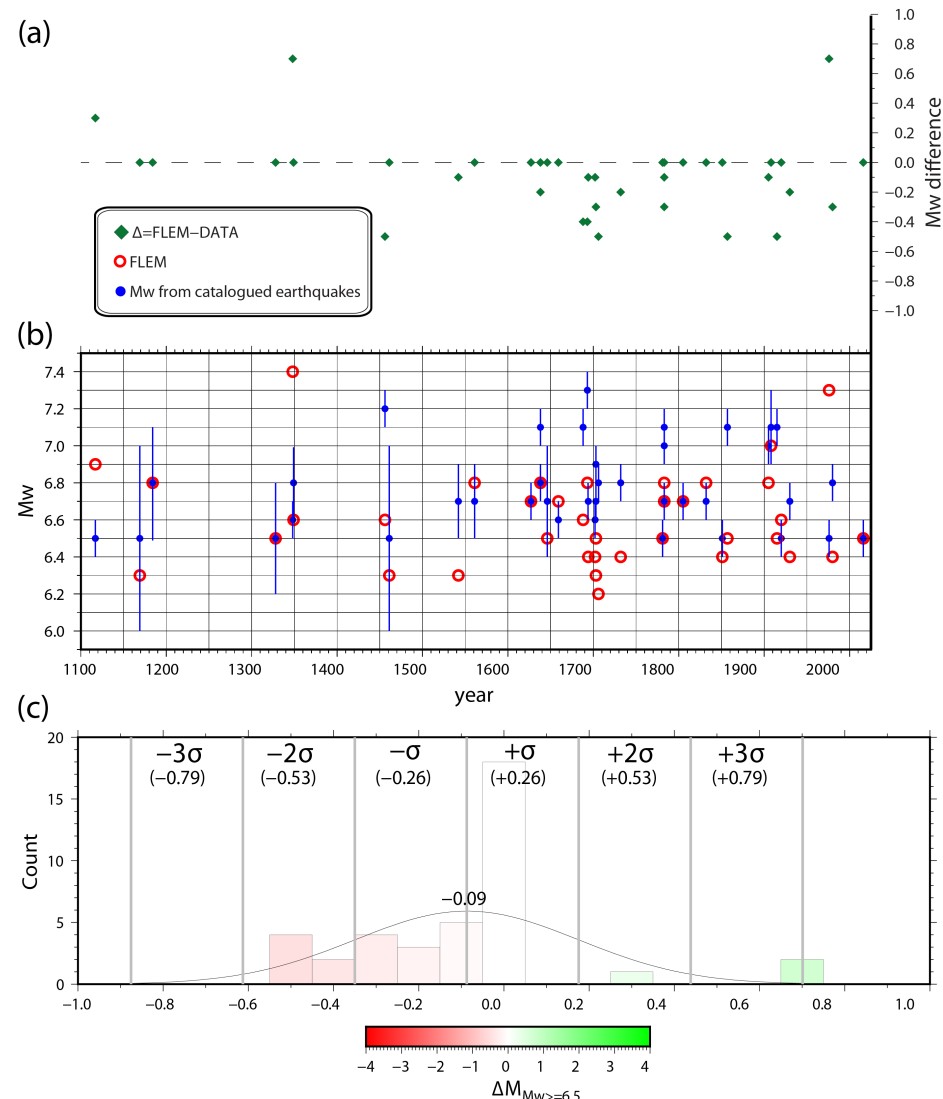

**Figure 13.** (a) Temporal series (since 1000 A.D.) of the difference between FLEMs (from class A faults) and spatially-coincident (same cell) historical/instrumental-catalogued magnitudes of earthquakes (only M ≥ 6.5). (b) In blue, temporal series of magnitudes from catalogued historical or instrumental earthquakes (M ≥ 6.5, 1000-2017 period). Blue vertical bars are for magnitude uncertainty. In red, FLEM values from the same cells hosting the catalogued historical or instrumental earthquakes drawn in blue. (c) Histogram of the difference between FLEMs (from class A faults) and spatially-coincident (same cell) historical/instrumental-catalogued magnitudes of earthquakes (M ≥ 6.5, 1000-2017 period). Mean = -0.09; standard deviation $\sigma$ = 0.26. See Fig. 12 for the interpretation of red vs. green portions of the histogram.

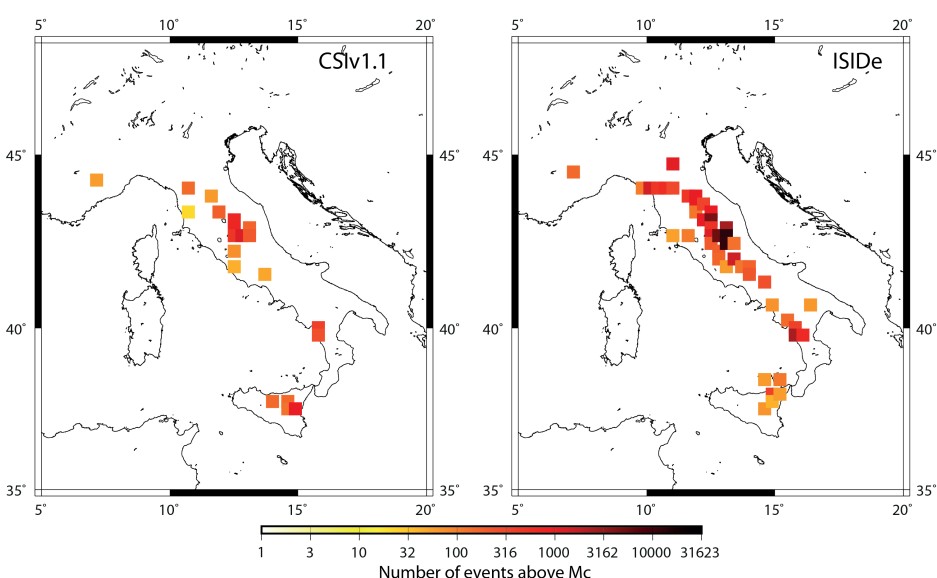

**Figure 14.** Maps of cells for which FLEM values are tested, on the CSiv1.1 (left panel, 20 cells) and ISIDe (right panel, 45 cells) databases (Tables S1 and S2). Colours are scaled with the number of events above the estimated completeness magnitude, $M_c$. In these cells the magnitude of events is distributed accordingly using a DTGR law, with a b-value equal to 1.

**Table 1.** Catalogued historical and instrumental earthquakes (M ≥ 6.5, 1000-2017 period) plotted in Fig. 13 with spatially-coincident (same 25x25 km cell) FLEM. Catalogued data are from Castello et al. (2006), Iside Working Group (2016), and Rovida et al. (2016).

| Region | Year | Lat | Lon | Mw Data | Err Mw Data | FLEM | Min DIFF |
|---|---|---|---|---|---|---|---|
| Apennine (S) | 1456 | 41.30 | 14.71 | 7.2 | 0.1 | 6.6 | -0.5 |
| Sicily (SE) | 1693 | 37.14 | 15.01 | 7.3 | 0.1 | 6.8 | -0.4 |
| Calabria | 1783 | 38.79 | 16.46 | 7.0 | 0.1 | 6.8 | -0.1 |
| Calabria | 1783 | 38.30 | 15.97 | 7.1 | 0.1 | 6.7 | -0.3 |
| Apennine (S) | 1857 | 40.35 | 15.84 | 7.1 | 0.1 | 6.5 | -0.5 |
| Calabria | 1905 | 38.81 | 16.00 | 7.0 | 0.1 | 6.8 | -0.1 |
| Messina Straits | 1908 | 38.15 | 15.69 | 7.1 | 0.2 | 7.0 | 0 |
| Abruzzi | 1915 | 42.01 | 13.53 | 7.1 | 0.1 | 6.5 | -0.5 |
| Apennine (S) | 1688 | 41.28 | 14.56 | 7.1 | 0.1 | 6.6 | -0.4 |
| Calabria | 1638 | 39.05 | 16.29 | 7.1 | 0.1 | 6.8 | -0.2 |
| Apennine (N) | 1920 | 44.19 | 10.28 | 6.5 | 0.1 | 6.6 | 0 |
| Apennine (S) | 1930 | 41.07 | 15.32 | 6.7 | 0.1 | 6.4 | -0.2 |
| Alps (E) | 1976 | 46.24 | 13.12 | 6.5 | 0.1 | 7.3 | 0.7 |
| Irpinia | 1980 | 40.84 | 15.28 | 6.8 | 0.1 | 6.4 | -0.3 |
| Umbria | 2016 | 42.83 | 13.11 | 6.5 | 0.1 | 6.5 | 0 |
| Po Plain | 1117 | 45.27 | 11.02 | 6.5 | 0.1 | 6.9 | 0.3 |
| Sicily | 1169 | 37.22 | 14.95 | 6.5 | 0.5 | 6.3 | 0 |
| Calabria | 1184 | 39.40 | 16.19 | 6.8 | 0.3 | 6.8 | 0 |
| Umbria | 1328 | 42.86 | 13.02 | 6.5 | 0.3 | 6.5 | 0 |
| Alps (E) | 1348 | 46.50 | 13.58 | 6.6 | 0.1 | 7.4 | 0.7 |
| Latium-Molise | 1349 | 41.55 | 13.94 | 6.8 | 0.2 | 6.6 | 0 |
| Abruzzi | 1461 | 42.31 | 13.54 | 6.5 | 0.5 | 6.3 | 0 |
| Sicily | 1542 | 37.22 | 14.94 | 6.7 | 0.2 | 6.3 | -0.1 |
| Apennine (S) | 1561 | 40.56 | 15.51 | 6.7 | 0.2 | 6.8 | 0 |
| Gargano | 1627 | 41.74 | 15.34 | 6.7 | 0.1 | 6.7 | 0 |
| Calabria | 1638 | 39.28 | 16.81 | 6.8 | 0.1 | 6.8 | 0 |
| Gargano | 1646 | 41.91 | 15.99 | 6.7 | 0.3 | 6.5 | 0 |
| Calabria | 1659 | 38.69 | 16.25 | 6.6 | 0.1 | 6.7 | 0 |
| Irpinia | 1694 | 40.86 | 15.41 | 6.7 | 0.1 | 6.4 | -0.1 |
| Irpinia | 1702 | 41.12 | 14.99 | 6.6 | 0.1 | 6.4 | -0.1 |
| Abruzzi | 1703 | 42.43 | 13.29 | 6.7 | 0.1 | 6.3 | -0.3 |
| Umbria | 1703 | 42.71 | 13.07 | 6.9 | 0.1 | 6.5 | -0.3 |
| Abruzzi | 1706 | 42.08 | 14.08 | 6.8 | 0.1 | 6.2 | -0.5 |
| Irpinia | 1732 | 41.06 | 15.06 | 6.8 | 0.1 | 6.4 | -0.2 |
| Marche | 1781 | 43.60 | 12.51 | 6.5 | 0.1 | 6.5 | 0 |
| Calabria | 1783 | 38.58 | 16.20 | 6.7 | 0.1 | 6.7 | 0 |
| Molise | 1805 | 41.50 | 14.47 | 6.7 | 0.1 | 6.7 | 0 |
| Calabria | 1832 | 39.08 | 16.92 | 6.7 | 0.1 | 6.8 | 0 |
| Apennine (S) | 1851 | 40.96 | 15.67 | 6.5 | 0.1 | 6.4 | 0 |