# Peer review of "From mapped faults to Fault-Length Earthquake Magnitude (FLEM): A test on Italy with methodological implications"

_Solid Earth, 2018_

## Short Comment (SC1) · 13 Dec 2018

Submitted by Gianluca Valensise, Roberto Basili, Pierfrancesco Burrato, Michele Carafa, Francesca Cinti, Riccardo Civico, Paolo Marco De Martini, Deborah Di Naccio, Umberto Fracassi, Fabrizio Galadini, Vanja Kastelic, Daniela Pantosti, Mara Monica Tiberti, and Paola Vannoli

The geology of Italy is normally considered to be extraordinarily complex, which makes it very fascinating but also quite difficult to investigate. The authors of this paper indeed make an exception, as they approach Italian geology as if major scientific questions concerning the relationships between earthquakes and their causative faults could be

addressed by following a handful of simple, universally recognised rules.

We doubt this is the case. In fact, we argue that Trippetta and co-workers did not consider at least two decades of literature on the relationships between seismogenic faulting at upper-mid crustal levels and brittle faulting at shallow crustal depth. Their paper is essentially a GIS exercise;, backed by references that are generally highly selective, often inappropriate and sometimes very old. For instance, a very general statement such as "Larger earthquakes characterize the Apennines southern portion (Calabria), with historical seismic events that reached magnitudes up to 6.9-7.5" is backed by a reference to Cello et al. (2003), a 15 years-old paper dealing with a specific earthquake in Val d'Agri, 50 km north of Calabria, and to Gasparini et al. (1985), a 33 years-old paper that belongs to a distant past of seismotectonics in Italy. A simple reference to the Catalogo Parametrico dei Terremoti Italiani (CPTI), the Italian reference parametric catalogue, would have been enough; it would also have prevented a mistake, since no M 7.5 earthquake is reported anywhere in Italy.

The conclusions of the paper are based on several misconceptions, leading to results that could have catastrophic outcomes for seismic risk mitigation in Italy, if taken seriously by any authority in charge. Its main misconception probably rests in the assumption that "...all considered faults are active or can be potentially reactivated..." (section 4.2, page 9 of the manuscript). It is unfortunate that the fault database that should support this bold statement is not accessible (see Petricca, P., et al., Revised dataset of known faults in Italy, GFZ Data Services, https://doi.org/10.5880/fidgeo.2018.003, http://pmd.gfz-potsdam.de/panmetaworks/, 2018; the first link leads to an error page, while the second leads to a generic page of the GFZ website).

Simply put, Trippetta and co-workers maintain that they can obtain the Potential Expected Maximum Magnitude of earthquakes (PEMM) from any fault that appears in their (currently inaccessible) compilation, which includes (from the caption of their Figure 2): • the Structural Model of Italy at the 1:500,000 scale; • the Italian Geological Maps at the 1:100,000 scale; • the GNDT database of active faults (Galadini et

al., 2000); • the ITHACA database (Michetti et al., 2000); • selected active faults from complementary (complementary to what?) studies published by different authors.

Notice that three of these studies refer explicitly to active - or at least capable - faults, but the other two sets, which happen to be the most numerous, do not. Also notice that Trippetta and co-workers subdivided each dataset into Class A (high quality), that includes ". . .exposed faults where subsurface and surface data allow for a detailed and reliable characterization of fault length. . .", and class B (low quality), containing ". . .buried and off-shore faults investigated mainly by seismic surveys, for which a precise characterization of fault length cannot be achieved. . .". Finally, notice that Trippetta and co-workers deliberately ignore the down-dip dimension (width) of the faults.

A basic consideration is that by assembling faults from such different and non-homogeneous sources, Petricca et al. inevitably put together a) alternative views on the same faults, possibly stemming from widely alternative conceptual models; b) faults that are mutually exclusive due to their geometry (typically, faults crossing each other in the subsurface: if one fault ends against another, its seismic potential based solely on length is largely overestimated); c) faults that cannot be simultaneously active, or reactivated, in the current stress regime; and d) blind faults whose actual length may be strongly biased by the availability and density of subsurface data.

In our opinion it is extremely unsafe to derive the maximum earthquake magnitude from the length of a fault of which we do not even know (a) if it is active or if it can be reactivated (e.g. whether or not it cuts or deforms deposits of a specified age or it is suitably oriented with respect to the current stress regime), and (b) whether and how it is connected with a deep-seated seismogenic source. For decades earthquake geologists have investigated the surface expression of presumed active faults to gain insight into their long-term behaviour and seismogenic potential. The main purpose of such studies is to distinguish between active and non-active faults (sealed, truncated, or misoriented with respect to the current stress field). From this paper we learned that this is no longer necessary, as any mapped fault has the potential of generating a large

earthquake.

Trippetta and co-workers propose a very simplistic approach to a rather complicated problem, based on criteria and assumptions that are not exposed in the text. A revealing exercise is to simply look at the faults that actually generated the most recent and well investigated earthquakes in Italy. To this end, we focused on Italian seismicity of the past 50 years (Table 1). According to the latest version of CPTI (https://emidius.mi.ingv.it/CPTI15-DBMI15/), 32 earthquakes of magnitude equal or larger than 5.5 occurred during this interval, plus the three largest shocks of the 2016 central Apennines sequence (shown in bold italics).

The question to ask is: for how many of these earthquakes could the authors declare a positive relationship between the magnitude and the surface length of the presumed earthquake causative fault? • probably for the Amatrice and Norcia shocks of the 2016 sequence; • not for the 6 April 2009, Mw 6.3, Aquilano earthquake, for which the fault dataset supplied by Trippetta and co-workers in their Supplementary Data (Dataset-S1-ItalianFaults.kmz: Figure 1) shows a fault that is less than 1 km in length, reportedly taken from the Structural model of Italy and included in Class A. This fault is at least 15 times shorter than the fault necessary to generate a Mw 6.3 earthquake, and it does not even coincide with the surface ruptures observed following the 6 April 2009 mainshock; • not for the 23 November 1980, Mw 6.8 Irpinia-Basilicata earthquake, for which the Class A subset of the dataset used by Trippetta and co-workers, derived from the Ithaca database, reports a ≈10 km-long fault; nearly four times shorter than the 38 km-long rupture reported in the classical paper by Pantosti and Valensise (1990), among those used by Wells and Coppersmith (1994) to derive their empirical relationships. At any rate, even this ≈10 km-long fault was not known prior to the earthquake; • and finally, nor for the 1968 Belice, 1976 Friuli, 1984 Abruzzo, 1990 Potentino, 1997 Colfiorito, 2002 Molise, 2012 Pianura emiliana earthquakes, whose causative faults were most likely blind.

One could argue that the earthquakes that occurred over the past 50 years were not

very large, on average, and hence are not representative of the problem at hand. But even the M 7+ earthquakes that have occurred in the early 20th century, the 1908 Messina Straits and the 1915 Avezzano (both of Mw 7.1 according to CPTI), provided contrasting evidence; • no surface breaks have ever been reported for the ≈40-km-long source of the 1908, Messina Straits earthquake, whose causative fault was most likely blind. Nevertheless, the Class A subset of the dataset used by Trippetta and co-workers, derived from GNDT, does report a ≈36-km-long offshore fault, probably based on geodetic evidence, showing that this dataset in fact reports a combination of inferred seismogenic sources and actual surface faults; • for the 1915, Avezzano earthquake the Class A subset of the dataset used by Trippetta and co-workers, derived from both the Ithaca and GNDT datasets, reports a ≈15-km-long fault. This fault accounts only partially for the 30-50-km-long fault that is needed to justify a Mw 7.1 event. The reported faults appear as the surface projection of a deeper portion of the seismogenic fault and do not follow the reported coseismic breaks.

In summary, it is likely that none of these earthquakes have been generated by a surface fault reported in Petricca et al.'s database and having the characteristics upon which the paper by Trippetta and co-workers is based. In other words, for none of these earthquakes could the magnitude be derived with confidence based solely on the reported length of their surface trace, which is systematically shorter than that observed (or inferred) after the earthquake. But if the proposed approach does not work on relatively large and recent earthquakes for which there is good control on the causative fault and on the magnitude, why should it work for older events and, most importantly for future ones?

The largest earthquakes require an additional consideration. Virtually no 20-40-km-long fault has ever been reported in the areas that were struck by the largest Italian earthquakes prior to their occurrence; but such long faults do occur in areas of more mature geology were seismicity is minimal or absent, such as in the Alps. This might indicate that currently active, seismogenic dip-slip faults (recall that most Italian upper

crustal earthquakes are generated by normal, reverse or thrust faults) are normally characterised by a discontinuous surface expression. The established basic geology of the orogens seems to show that when their activity stops, some of these faults are uplifted and rejuvenated along with their associated landscape, making them appear more continuous, more mature and downright ominous, right at a time of their evolution when they are no longer a concern for seismic hazard.

At the opposite end of the spectrum lie some catastrophic Italian earthquakes of the past few centuries that occurred in foreland areas and that reportedly were not accompanied by surface faulting. Such is the case of the 1693 Eastern Sicily earthquake, the largest in the Italian earthquake catalogue (Mw 7.3).

Under these ill-posed assumptions (assuming that any mapped fault is potentially active, or can be reactivated; mixing faults that may be mutually exclusive; disregarding the post-orogenic rejuvenation of many faults, etc.), the results obtained by Trippetta and co-workers - and plastically represented in their Figures 11 and 12 - are all but unexpected. They show very high-magnitude values of the PEMM for the less-reliable Class B faults; a circumstance that can be simply explained considering that buried fault traces obtained by interpolating sparse data over wide areas will necessarily appear longer than faults within an exposed fault system. It is true that we may have yet to see some exceptionally large crustal earthquake, i.e. larger than any earthquake already known to the historical record; but without adopting any criteria regarding the likelihood of these large ruptures, such scenario is totally at odd with the unique historical, archaeological and palaeoseismological richness of the Italian earthquake record. This information provides the basis for a much more mindful way to estimating the magnitude of future large earthquakes, also in view of the frequency-magnitude distribution of earthquake occurrence.

The bottom line is that the approach proposed in the paper by Trippetta and co-workers goes against the knowledge acquired during decades of seismic hazard studies in Italy and the rest of the world. It does not predict anything useful but is potentially dangerous

and very costly, if not put in the right perspective. The authors never admit that their results are pointless, even when they remark (Page 12) that "... the negative occurrences are very limited...", i.e. that the number of predicted magnitudes that are larger than those observed in the historical record outnumbers by far the opposite case, resulting in the very asymmetric pattern shown in Figure 12. In fact, they refer to a "...limited difference... between PEMMs and the catalogued earthquake magnitudes..." (!), neglecting the obvious consideration that magnitude is a logarithmic quantity, implying that a 0.2 increase in Mw, for example from 6.0 to 6.2, doubles the seismic moment. For a typical continental fault having an aspect ratio in the range 2-3 and standard scaling for coseismic slip, doubling the moment implies that fault length may increase by over 20%. For a magnitude increase of 0.5, for example from 6.0 to 6.5, the seismic moment becomes 5.6 times larger, which may require a fault that is 100% or more longer than that necessary to generate the smaller earthquake.

In commenting their rather disappointing results the authors invoke potential incompleteness of the earthquake catalogue, which is always possible locally, but is also very unlikely for a work that considers the whole of Italy and all the data available in the richest earthquake catalogue worldwide. This record has certainly missed specific events but is reliable enough to constrain the frequency-magnitude distribution of Italian seismicity. A piece of information that a study about the estimates of earthquake magnitude cannot neglect.

Trippetta and co-workers conclude by stating that "...more detailed studies should be developed...". This is always a good thing to do, except for the fact that detailed studies of Italian faults do exist and are often more accurate and to the point with respect to what is proposed here.

The conclusions of this paper are worrisome, in consideration of the large number of areas where the authors envision the possibility of M 7.5 and larger earthquakes, that is to say earthquakes bigger than the largest magnitude ever recorded in Italy, without any consideration as to how frequently this may occur. In a standard PSHA approach

these large magnitudes would be assigned a very low probability of occurrence, leading to a minimal statistical impact on the expected ground shaking for short average return periods. The information about the possible largest earthquakes may generate a great deal of confusion if not appropriately communicated. We cannot imagine how the residents of Bologna, Ancona, Pescara, but also Padua, Trento, Vicenza and even Venice, cities lying in areas that are currently considered mid- to low-hazard, would react to knowing that very large earthquakes may occur below their feet at any moment.

Another major flaw in the approach taken by Trippetta and co-workers lies in their discretisation of seismogenic zones into 25x25 km sub-areas. Of course, some discretisation is inevitable, but one has to be aware that a 25x25 km cell may host a 35 km-long fault, at the most. According to the equation proposed by Leonard (2010), the empirical law adopted by Trippetta and co-workers, a 35 km fault length corresponds to a Mw 6.8 earthquake. Hence, any larger earthquake will necessarily encompass two or more cells. A close inspection of Figures 7 and 8 of the paper, however, reveals that several cells filled in red or dark red, which according to the adopted colour-coding should correspond to an expected Mw in the range 7.4 to 7.8, occur isolated, i.e., surrounded by cells for which the expected PEMM is much smaller. According to same equation by Leonard (2010), this magnitude range corresponds to a fault length in the range 78 to 135 km, which should involve a minimum of 2 to 4 adjacent cells, depending on fault strike. An isolated cell capable of a Mw 7.4 earthquake is hence a seismological paradox that has no physical meaning, as the earthquake causative fault will necessarily extend to adjacent cells.

We are a group of INGV seismologists and earthquake geologists who regularly provide active faulting data and seismogenic models to Italian, European, and global SHA practitioners, and to the Italian Civil Protection authorities. As such we are especially concerned that inaccurately collected fault data, inconsistent elaborations and unjustified conclusions such as those presented by Trippetta and co-workers may be implicitly validated by appearing in a respectable journal such as Solid Earth, thus becoming embedded in the literature. In addition to that, chasing the problem of the occurrence of very large earthquakes with an overly simplistic approach is the most effective way to shift our attention from the areas where earthquakes in the Mw range 6-0-7.0 are more likely to hit. These areas include most of the eastern Alps, various portions of the Apennines, most of Calabria and several parts of Sicily. Encouraging earthquake retrofitting in these areas should be the main target of any responsible seismological community.

References

Cello, G., Tondi, E., Micarelli, L., and Mattioni, L.: Active tectonics and earthquake sources in the epicentral area of the 1857 Basilicata earthquake (southern Italy), Journal of Geodynamics, 36, 37–50, doi: 10.1016/S0264-3707(03)00037-1, 2003.

Galadini, F., Meletti, C., and Vittori, E.: Stato delle conoscenze sulle faglie attive in Italia: elementi geologici di superficie, Le ricerche del GNDT nel campo della pericolosità sismica (1996–1999), pp. 107–136, 2000.

Gasparini, C., Iannaccone, G., and Scarpa, R.: Fault-plane solutions and seismicity of the Italian peninsula, Tectonophysics, 117, 59–78, doi: 10.1016/0040-1951(85)90236-7, 1985.

Leonard, M.: Earthquake fault scaling: self-consistent relating of rupture length, width, average displacement, and moment release earthquake fault scaling, Bull. Seism. Soc. Am., 100, 1971, doi: 10.1785/0120090189, 2010.

Michetti, A. M., Serva, L., and Vittori, E.: ITHACA Italy Hazard from Capable Faults: a database of active faults of the Italian onshore territory, with CD-ROM. Explanatory notes Published by ANPA, Rome, 150 pp., 2000.

Pantosti, D., and Valensise G.: Faulting mechanism and complexity of the 23 November 1980, Campania-Lucania earthquake, inferred from surface observations, J. Geophys. Res., 95, 15319-15341, 1990.
[Figure]

Rovida, A., M. Locati, R. Camassi, B. Lolli, and Gasperini, P. (eds.): CPTI15, the 2015 Version of the Parametric Catalogue of Italian Earthquakes. Published by Istituto Nazionale di Geofisica e Vulcanologia. doi: 10.6092/INGV.IT-CPTI15, 2016.

Wells, D.L., and Coppersmith K. J.: New empirical relationships among magnitude, rupture length, rupture width, rupture area, and surface displacement, Bull. Seismol. Soc. Am., 84, 974–1002, 1994.

Please also note the supplement to this comment:
https://www.solid-earth-discuss.net/se-2018-98/se-2018-98-SC1-supplement.pdf

[Figure]

[Figure]

**Figure 1** - The Paganica Fault (shown by the red arrow) as reported in the database provided with the paper.

**Supplement:**

**Table 1**

List of the 32 earthquakes of magnitude equal or larger than 5.5 that occurred in Italy over the past 50 years (i.e. since 1968), taken from the latest version of CPTI (https://emidius.mi.ingv.it/CPTI15-DBMI15/). The last three lines (shown in bold italics) contain the three largest shocks of the 2016 central Apennines sequence.

| Year | Month | Day | Epicentral Area | Mw |
|------|-------|-----|-----------------|-----|
| 1968 | 1 | 15 | Valle del Belice | 6.41 |
| 1968 | 1 | 15 | Valle del Belice | 5.53 |
| 1971 | 7 | 15 | Parmense | 5.51 |
| 1976 | 5 | 6 | Friuli | 6.45 |
| 1976 | 9 | 11 | Friuli | 5.60 |
| 1976 | 9 | 15 | Friuli | 5.93 |
| 1976 | 9 | 15 | Friuli | 5.95 |
| 1978 | 4 | 15 | Golfo di Patti | 6.03 |
| 1979 | 9 | 19 | Valnerina | 5.83 |
| 1980 | 5 | 28 | Tirreno meridionale | 5.66 |
| 1980 | 11 | 23 | Irpinia-Basilicata | 6.81 |
| 1984 | 4 | 29 | Umbria settentrionale | 5.62 |
| 1984 | 5 | 7 | Monti della Meta | 5.86 |
| 1990 | 5 | 5 | Potentino | 5.77 |
| 1990 | 12 | 13 | Sicilia sud-orientale | 5.61 |
| 1997 | 9 | 26 | Appennino umbro-marchigiano | 5.66 |
| 1997 | 9 | 26 | Appennino umbro-marchigiano | 5.97 |
| 1997 | 10 | 14 | Valnerina | 5.62 |
| 1998 | 4 | 12 | Slovenia nord-occidentale | 5.64 |
| 1998 | 9 | 9 | Appennino lucano | 5.53 |
| 2002 | 9 | 6 | Tirreno meridionale | 5.92 |
| 2002 | 10 | 31 | Molise | 5.74 |
| 2002 | 11 | 1 | Molise | 5.72 |
| 2009 | 4 | 6 | Aquilano | 6.29 |
| 2009 | 4 | 7 | Aquilano | 5.54 |
| 2012 | 5 | 20 | Pianura emiliana | 6.09 |
| 2012 | 5 | 29 | Pianura emiliana | 5.90 |
| 2012 | 5 | 29 | Pianura emiliana | 5.50 |
| *2016* | *8* | *24* | *Amatrice* | *6.00* |
| *2016* | *10* | *26* | *Visso* | *5.90* |
| *2016* | *10* | *30* | *Norcia* | *6.50* |

---

## Referee Comment (RC1) · Anonymous Referee #1 · 14 Dec 2018

Review of manuscript SE-2018-98 "From mapped faults to earthquake magnitude: a test on Italy with methodological implications" by F. Trippetta, P. Petricca, A. Billi, C. Collettini, M. Cuffaro, D. Scrocca, G. Ventura, A. Morgante, F. Chiaravalli & C. Doglioni

**Main comments**

This manuscript describes an approach to evaluate the maximum possible earthquake magnitude from the geometry of active faults in Italy. The topic has a broad interest, in particular for fault-based seismic hazard modelling, because the correct evaluation of the seismic potential of a seismogenic source is one of the main required parameters for seismic hazard studies. The use of active faults in seismic hazard assessment has become extensive in the last decades due to efforts of data compilation and analysis. Active faults provide the information to extend the observational time of large magnitude earthquakes, which often is not captured by the existing catalogues of observed seismicity.

The authors apply one empirical scaling relationship (not correctly defined by the authors as scaling law!) between fault length and expected magnitude, to a clearly incomplete and inhomogeneous fault database, to evaluate the maximum possible earthquake magnitude in Italy. The manuscript is mostly well written and the figures are clear but, from my point of view, the approach has several misconceptions and incompleteness and the conclusions are not supported by the results.

For these reasons, the manuscript does not represent a substantial contribution to scientific progress in seismic hazard assessment and seismic risk reduction, as required by a high-level Journal as Solid Earth. The applied methods are valid but too simplistic and applied to incomplete and inhomogeneous data with consequently a poor scientific significance of the manuscript. Moreover, the references used for the fault database compilation is largely incomplete.

**Recommendation**

I recommend that this manuscript is not suitable in its present form for a publication on a regular issue of the Solid Earth. In the following detailed comments I describe more in details the main improvements needed, from my point of view, to re-submit the manuscript.

**Detailed comments**

1. Fault database: the database is largely incomplete and inhomogeneous, with an incompleteness variable in space. The suggestions to improve the database are:
   a. Consider the abundant literature in the compilation of active fault database for Italy, mostly more recent than the used one (in the following references a partial and not yet complete list of papers not considered in the manuscript);
   b. Separate in the database the recognized active faults to the not clearly active ones. In literature are available several definitions of fault activity, taking into account the age of the involved deposits, the associated earthquakes, the continuity and kinematics compatibility, and many others, and the authors need to consider it. In this way the authors could define more classes of faults (more than the two defined in the manuscript) based on the goodness of data and

recent activity and need to treat separately the classes in the approach to evaluate the seismogenic potential;

   c. Evaluate and handle with the spatial variable incompleteness of the database;

2. Consider the fault segmentation variability in the correct evaluation of the seismogenic potential, essential in fault-based seismic hazard approaches, as confirmed by recent complex coseismic ruptures (e.g., 2010 M 7.1 Canterbury, 2012 Mw 8.6 Sumatra, 2016 Mw 7.8 Kaikōura, 2016 Mw 6.5 central Italy);

3. Organize a table with earthquake-fault associations, in order to avoid the double counting or the source missing. There are several examples of 'problems' in the *kmz of the authors, as the missing of the Paganica fault, responsible of the M6.3 2009 L'Aquila earthquake, the double counting of some faults in the Fucino area where the M7 1915 earthquake occurred, and the not correct definition of the total length of the fault responsible in the Irpinia region of the M6.8 1980 earthquake;

4. Consider also the seismogenic depth and the length of the faults along dip, to better define the total potential rupture area, better linked to the seismogenic potential of the sources;

5. Deal with greater accuracy the empirical scaling relationships, by:

   a. Compare the results of the approach using the different available relationships;

   b. Handle with the uncertainties, both inter- the different relationships and intra- the single relationships (sometimes the authors define very large standard deviation in the empirical relationship not treated in the manuscript);

   c. Compare the results using different geometrical parameters, e.g. the surface rupture length, the subsurface rupture length, the rupture area (and so considering the seismogenic thickness) together with the different kinematics, treated separately in the different scaling relationships;

6. Handle with the uncertainties in the results, comparing the differences between the seismogenic potential of the faults estimated by the empirical relationships and the earthquakes in the historical catalogue, in terms of seismic moment and not only magnitude. Magnitude is a logarithmic quantity and so a simple comparison as done by the authors in the conclusions has a clear bias;

7. Treat the probability of occurrence in the conclusions. In seismic hazard models is necessary to define the seismic rates for the different magnitude classes and so the probability of occurrence of a defined magnitude depends on the average recurrence time of that value in a specific area. The conclusions of the authors show the largest expected magnitudes in areas with very low seismicity, like Sardinia, suggesting there high seismic hazard values. Such conclusions have to be more strongly supported by considerations in terms of probability of occurrence.

**References for the fault database**

Barchi, M., F. Galadini, G. Lavecchia, P. Messina, A.M. Michetti, L. Peruzza, A. Pizzi, E. Tondi, and E. Vittori. 2000. Sintesi delle conoscenze sulle faglie attive in Italia centrale. CNR - Gruppo Nazionale per la Difesa dai Terremoti, Rome, Italy, 62 pp

Benedetti, L., Manighetti, I., Gaudemer, Y., Finkel, R., Malavieille, J., Pou, K., Arnold, M., Aumaitre, G., Bourles, D., and Keddadouche, K. 2013. 'Earthquake synchrony and clustering on Fucino faults (Central Italy) as revealed from in situ Cl-36 exposure dating', J. Geophys. Res.-

Sol. Ea., 118, 4948–4974

Boncio, P., G. Lavecchia, and B. Pace. 2004. 'Defining a model of 3D seismogenic sources for Seismic Hazard Assessment applications: The case of central Apennines (Italy)', Journal of Seismology, 8: 407-25

Brozzetti, F. 2011. 'The Campania-Lucania Extensional Fault System, southern Italy: A suggestion for a uniform model of active extension in the Italian Apennines', Tectonics, 30

Burrato, P., M. E. Poli, P. Vannoli, A. Zanferrari, R. Basili, and F. Galadini. 2008. 'Sources of M-w 5+ earthquakes in northeastern Italy and western Slovenia: An updated view based on geological and seismological evidence', Tectonophysics, 453: 157-76

Calamita, F., M. Coltorti, D. Piccinini, P. P. Pierantoni, A. Pizzi, M. Ripepe, V. Scisciani, and E. Turco. 2000. 'Quaternary faults and seismicity in the Umbro-Marchean Apennines (Central Italy): evidence from the 1997 Colfiorito earthquake', Journal of Geodynamics, 29: 245-64

Catalano, S., G. De Guidi, C. Monaco, G. Tortorici, and L. Tortorici. 2008. 'Active faulting and seismicity along the Siculo-Calabrian Rift Zone (Southern Italy)', Tectonophysics, 453: 177-92

Cello, G., S. Mazzoli, E. Tondi, and E. Turco. 1997. 'Active tectonics in the central Apennines and possible implications for seismic hazard analysis in peninsular Italy', Tectonophysics, 272: 43-68

Cinque, A., A. Ascione, and Caiazzo. 2000. Distribuzione spazio-temporale e caratterizzazione della fagliazione quaternaria in Appennino meridionale. 2000. in: Galadini F., Meletti C., and Rebez A. (Eds), Le ricerche del GNDT nel campo della pericolosita sismica (1996- 1999), CNR-Gruppo Nazionale per la Difesa dai Terremoti - Roma, 203-218

Faure-Walker J. 2014. Mechanics of continental extension from Quaternary strain fields in the Italian Apennines. PhD Thesis, pp 405

Galadini, F. and P. Galli. 2000. Active Tectonics in the Central Apennines (Italy) - Input Data for Seismic Hazard Assessment. Natural Hazards, 22: 225-270

Galli, P., F. Galadini, and D. Pantosti. 2008. 'Twenty years of paleoseismology in Italy', Earth-Science Reviews, 88: 89-117

Kastelic, V., P. Vannoli, P. Burrato, U. Fracassi, M. M. Tiberti, and G. Valensise. 2013. 'Seismogenic sources in the Adriatic Domain', Marine and Petroleum Geology, 42: 191-213

Monaco, C., and L. Tortorici. 2000. 'Active faulting in the Calabrian arc and eastern Sicily', Journal of Geodynamics, 29: 407-24

Morewood, N. C., and G. P. Roberts. 2000. 'The geometry, kinematics and rates of deformation within an en echelon normal fault segment boundary, central Italy', Journal of Structural Geology, 22: 1027-47

Pace, B., L. Peruzza, G. Lavecchia, and P. Boncio. 2006. 'Layered seismogenic source model and probabilistic seismic-hazard analyses in central Italy', Bulletin of the Seismological Society of America, 96: 107-32

Papanikolaou, I. D., and G. P. Roberts. 2007. 'Geometry, kinematics and deformation rates along the active normal fault system in the southern Apennines: Implications for fault growth', Journal of Structural Geology, 29: 166-88

Papanikolaou, I. D., G. P. Roberts, and A. M. Michetti. 2005. 'Fault scarps and deformation rates in Lazio-Abruzzo, Central Italy: Comparison between geological fault slip-rate and GPS data', Tectonophysics, 408: 147-76

Peruzza, L., B. Pace, and F. Visini. 2011. 'Fault-Based Earthquake Rupture Forecast in Central Italy: Remarks after the L'Aquila M-w 6.3 Event', Bulletin of the Seismological Society of America, 101: 404-12

Pizzi, A., F. Calamita, M. Coltorti and P. Pieruccini. 2002. Quaternary normal faults, intramontane basins and seismicity in the Umbria-Marche Abruzzi Appenine ridge (Italy): contribution of neotectonic analysis to seismic hazard assessment. Bollettino della società geologica italiana, 1: 923-929

Schlagenhauf, A., I. Manighetti, L. Benedetti, Y. Gaudemer, R. Finkel, J. Malavieille, and K. Pou. 2011. 'Earthquake supercycles in Central Italy, inferred from Cl-36 exposure dating', Earth and Planetary Science Letters, 307: 487-500

Tortorici, L., C. Monaco, C. Tansi, and O. Cocina. 1995. 'Recent and Active Tectonics in the

Calabrian Arc (Southern Italy)', Tectonophysics, 243: 37-55

Valensise, G., and D. Pantosti. 2001. 'The investigation of potential earthquake sources in peninsular Italy: A review', Journal of Seismology, 5: 287-306

Valentini, A. F. Visini and B. Pace. 2017. 'Integrating faults and past earthquakes into a probabilistic seismic hazard model for peninsular Italy' Natural Hazards and Earth System Sciences, doi.org/10.5194/nhess-17-2017-2017

---

## Short Comment (SC2) · 15 Dec 2018

I welcome the comments by Referee 1 and thank him/her for the time spent on this manuscript. Together with my co-authors, I will soon respond point-by-point to each comments. We will surely consider these comments to improve the manuscript.

A first reply is however necessary. The Referee's comments seem strongly influenced by an initial misunderstanding. The Referee states: "the manuscript does not represent a substantial contribution to scientific progress in seismic hazard assessment and seismic risk reduction, as required by a high-level Journal as Solid Earth ... with consequently a poor scientific significance of the manuscript."

I would like to acknowledge that this is not at all a contribution to seismic hazard assessment and seismic risk reduction.

At the end of the Introduction we state: "We anticipate that, with this work, we do not intend to propose an alternative method for seismic hazard assessment or to better previous methods (e.g., Giardini, 1999; Jiménez et al., 2001; Michetti et al., 2005; Field et al., 2009, 2015; Reicherter et al., 2009). Our main aim is to test whether solely considering the known mapped faults (both active, inactive, and undetermined) and disregarding further information (e.g., historically- and instrumentally-recorded earthquakes as well as the regional stress field and strain rate) it is possible to provide, through existing seismic scaling laws of faults and earthquakes, reasonable assessments of the maximum possible earthquake magnitude over an entire nation. The resulting (assessed) magnitudes (PEMM) are compared (i.e., the mathematical difference) with catalogued earthquake magnitudes that are the only existing points of reference against which assessed magnitudes can be compared."

In other words, our work is a test (novel for the scale and mode of application) on the empirical scaling relationships between fault size and earthquake magnitude at the national scale (Wells and Coppersmith and subsequent modifications/improvements by Leonard). The scientific relevance of this test is quantitatively provided by the number of present citations received by the paper by Wells and Coppersmith (1994): at present, on ISI Thomson Web of Science, this number reaches almost 3,500 (i.e. about 140 citations per year). I deem therefore that testing these empirical scaling relationships at the national scale on a very seismic country like Italy is scientifically significant.

Seismic hazard is defined as the probability that an earthquake will occur in a given geographic area, within a given window of time, and with ground motion intensity exceeding a given threshold. Simply, this is not the aim of our work.

Seismic risk refers to the risk of damage from earthquake to a building, system, or other entity. Seismic risk has been defined, for most management purposes, as the

potential economic, social and environmental consequences of hazardous events that may occur in a specified period of time. Also in this case, seismic risk and its reduction are simply not the subject of our paper.

I reaffirm that "Our main aim is to test whether solely considering the known mapped faults (both active, inactive, and undetermined) and disregarding further information (e.g., historically- and instrumentally-recorded earthquakes as well as the regional stress field and strain rate) it is possible to provide, through existing seismic scaling laws of faults and earthquakes, reasonable assessments of the maximum possible earthquake magnitude over an entire nation."

Concerning the used database of faults, I fear that also in this case there may be a misunderstanding. The used database is not the one of Fig. 5 (which contains only the longest fault for each square cell). The fault database is the one of Fig. 2 and the database is publicly available at this link (external repository): http://pmd.gfz-potsdam.de/panmetaworks/review/924b171fd21c78f295d58a7e9e321e8ad07667ab6201634b23d3cb5a3f170d10/

I thank again the Referee for his/her constructive comments. We will soon respond in more detail and try to do our best to follow his/her indications to improve our manuscript.

Sincerely

Andrea Billi

---

## Author Comment (AC1) · 18 Dec 2018

By Fabio Trippetta, Patrizio Petricca, Andrea Billi, Cristiano Collettini, Marco Cuffaro, Davide Scrocca, Giancarlo Ventura, Andrea Morgante, Fabio Chiaravalli, and Carlo Doglioni

Concerning the comments posted by Valensise et alii, below, we respond point-by-point to all their arguments. We apologize for our repetitive responses that are induced by reiterated comments by Valensise et alii.

Introduction Comments by Valensise et alii, as happens also for most comments by

[Figure]

Referee 1, seem influenced by an initial and radical misunderstanding. Our manuscript IS NOT titled "A new map of expected earthquake magnitudes for seismic hazard and risk mitigation in Italy Simply, this is not the goal of the article. Our maps (Figs. 7 and 8) ARE NOT AT ALL predictive maps of seismic hazard.

Our manuscript is titled "From mapped faults to earthquake magnitude: A test on Italy with methodological implications" and our aim is stated clearly from the introduction where we say: "We anticipate that, with this work, we do not intend to propose an alternative method for seismic hazard assessment or to better previous methods (e.g., Giardini, 1999; Jiménez et al., 2001; Michetti et al., 2005; Field et al., 2009, 2015; Reicherter et al., 2009). Our main aim is to test whether solely considering the known mapped faults (both active, inactive, and undetermined) and disregarding further information (e.g., historically- and instrumentally-recorded earthquakes as well as the regional stress field and strain rate) it is possible to provide, through existing seismic scaling laws of faults and earthquakes, reasonable assessments of the maximum possible earthquake magnitude over an entire area. The resulting (assessed) magnitudes (PEMM) are compared (i.e., the mathematical difference) with catalogued earthquake magnitudes that are the only existing points of reference against which assessed magnitudes can be presently compared."

The empirical relationships (used by many geoscientists including Valensise et alii) between fault size and earthquake magnitude are notoriously problematic for issues such as fault segmentation, fault continuity, and seismic partial activation of long faults. These problems have been ascertained ex-post on most single faults after earthquakes. We work on these same problems at the national scale with a different approach (ex-ante). That is, given a set of "official" faults of a nation (from geological maps and other official datasets), is it possible to test, quantify, and ascertain the above-mentioned problems connected with the empirical relationships? One way to do this, is to apply the relationship to the fault dataset and then compare the results with a seismic catalogue. This is what we do in Figs. 10, 11, 12, and 13 + Table 1. The results of this work cannot be used for Seismic Hazard Assessment but rather, they can highlight areas where further studies are required to better asses expected earthquake magnitudes

In the revision we will better emphasize the above-mentioned concepts, starting from the abstract that in the previous version of the manuscript could have left some space for misinterpretation. We will also add disclaimers in Figures 7 and 8 saying that these figures are not at all for use for Civil Protection, officials, and, in general, for seismic hazard.

Comment 1: The geology of Italy is normally considered to be extraordinarily complex, which makes it very fascinating but also quite difficult to investigate. The authors of this paper indeed make an exception, as they approach Italian geology as if major scientific questions concerning the relationships between earthquakes and their causative faults could be addressed by following a handful of simple, universally recognised rules.

Response 1: As geologists, we all know that tectonics can be complex and not only in Italy. On the other hand, all scaling laws and in particular well-recognized laws, like those proposed by Wells and Coppersmith, 1994 and Leonard, 2010, must overcome the complexity of regional settings and focus on the fault size/displacement to infer the potential magnitude.

Moreover the same authors to whom we are answering wrote several papers where the magnitude of paleoearthquakes was inferred from fault size/displacement measured at or near the Earth's surface (Pantosti et al. 1993 doi: 10.1029/92JB02277; Valensise and Pantosti, 2001 doi: 10.1023/A:1011463223440; Galli et al., 2008 doi: 10.1016/j.earscirev.2008.01.001; Pantosti et al., 1993 doi: 10.1029/95JB03213; Collier et al., 1998 doi: 10.1029/98JB02643; Galadini et al., 2005 doi: 10.1111/j.1365-246X.2005.02571.x; Galadini and Galli 2003 doi: 10.4401/ag-3457 and several others by Valensise et alii).

Also the DISS database (Database of Individual Seismogenic Sources in Italy,

http://diss.rm.ingv.it/diss/), which is run by Valensise et alii, makes large use of the scaling relationships between fault size/displacement and earthquake magnitude to infer this latter parameter in Italy.

In our study, we used the same relationships (by Wells and Coppersmith, 1994, later revised by Leonard, 2010) that many authors including Valensise et alii have used for years to infer the earthquake magnitude from fault attributes. Simul stabunt vel simul cadent

Comment 2: We doubt this is the case. In fact, we argue that Trippetta and co-workers did not consider at least two decades of literature on the relationships between seismogenic faulting at upper-mid crustal levels and brittle faulting at shallow crustal depth. Their paper is essentially a GIS exercise;, backed by references that are generally highly selective, often inappropriate and sometimes very old.

Response 2: In this paper, we tried to use the most updated literature; however, since we know very well that Valensise et alii are expert in this field, we look forward to follow their suggestions on updated papers to be added and on how and why each of these missed papers affected/biased our analyses and results. However, it must be noted that it is not the presence of one or more unmapped faults that falsifies the proposed simple technique..

Comment 3: For instance, a very general statement such as "Larger earthquakes characterize the Apennines southern portion (Calabria), with historical seismic events that reached magnitudes up to 6.9-7.5" is backed by a reference to Cello et al. (2003), a 15 years-old paper dealing with a specific earthquake in Val d'Agri, 50 km north of Calabria, and to Gasparini et al. (1985), a 33 years-old paper that belongs to a distant past of seismotectonics in Italy. A simple reference to the Catalogo Parametrico dei Terremoti Italiani (CPTI), the Italian reference parametric catalogue, would have been enough; it would also have prevented a mistake, since no M 7.5 earthquake is reported anywhere in Italy

[Figure]

Response 3: This criticism refers to a sentence extracted from the seismotectonic setting (Page 5 Line 13) and not from the method and results sections. We will accept with pleasure the suggestion by Valensise et alii and will enrich the Seismotectonic Setting with some recent references in the revised version of our manuscript. However, we have to point out that the core and science of our paper is not at all affected by a supposed lack of consideration of previous works, as argued by the comment.

More importantly, in section 3.2 earthquake data, we do mention CPTI together with the most comprehensive catalogues of instrumental and historical seismicity like CSI1.1 and ISIDe.

Comment 4: The conclusions of the paper are based on several misconceptions, leading to results that could have catastrophic outcomes for seismic risk mitigation in Italy, if taken seriously by any authority in charge. Its main misconception probably rests in the assumption that "...all considered faults are active or can be potentially reactivated..." (section 4.2, page 9 of the manuscript).

Response 4: The statement "The conclusions of the paper are based on several misconceptions ..." is too vague e generic. Please be specific. We are available for improvements.

If Valensise et alii mean that our misconceptions are due to the assumption that "...all considered faults are active or can be potentially reactivated...", then the long list of previous experiences on apparently inactive faults abruptly reactivated by earthquakes support our assumption. (Bouchon et al., 1998; Camelbeeck et al., 2007; Kafka, 2007; Stein and Mazzotti, 2007; Swafford et al., 2007; Braun et al., 2009; Boyd et al., 2013; Leonard et al., 2014; Talwani, 2014; Campbell et al., 2015; Walsh III and Zoback, 2016; Christophersen et al., 2017). This is clearly explained at P. 10 L.20-28. The same map of seismic hazard in Italy has changed with time once some new earthquake strokes an "unexpected areas". Faults may be apparently inactive because of the short time frame in which we observe them, or they may be active at depth without surface expression,

possibly due to a thinner brittle crust and to low magnitude. It is evident that very long faults may be activated only along specific segments, as it happens for example along the plates interface of subduction zones. However, in the testb site of Italy, there are very few areas that are seismically silent, suggesting active faults being present almost everywhere, particularly along the Apennines rifting area.

We acknowledge that this study developed within the framework of nuclear waste repositories planning. In societal challenges such as the construction of nuclear waste repositories, the future time span to consider is 1 Ma or more. In such a time span, any fault, including those that are now sutured and apparently inactive, could slowly accumulate stress to be released in future earthquakes. For this reason, it is important to attempt knowing what type (magnitude) of earthquake could these (presumably-inactive) faults produce, regardless their recurrence time and probability of occurrence. For this reason, we have considered all know faults independently of their presumed age.

Finally, dealing with catastrophic outcomes for seismic risk mitigation. In the last two sentences of the conclusions we clearly state that: "This method cannot, however, be a substitute for time-dependent (paleo)seismological methods for seismic hazard assessments. Rather, it can rapidly provide an approximate perspective time-independent seismic potential of faults and highlight areas where further detailed studies are required".

Comment 5: It is unfortunate that the fault database that should support this bold statement is not accessible (see Petricca, P., et al., Revised dataset of known faults in Italy, GFZ Data Services, https://doi.org/10.5880/fidgeo.2018.003, http://pmd.gfz-potsdam.de/panmetaworks/, 2018; the first link leads to an error page, while the second leads to a generic page of the GFZ website).

Response 5: The database is fully and publicly available at the indicated database under this link that we will report also on the revised manuscript: http://pmd.gfz-potsdam.de/panmetaworks/review/924b171fd21c78f295d58a7e9e321e8ad07667ab6201634b23d3cb5a3f170d10/

[Figure]

Comment 6: Simply put, Trippetta and co-workers maintain that they can obtain the Potential Expected Maximum Magnitude of earthquakes (PEMM) from any fault that appears in their (currently inaccessible) compilation, which includes (from the caption of their Figure 2): âËŸA 'c the Structural Model of Italy at the 1:500,000 scale; âËŸA 'c the Italian Geological Maps at the 1:100,000 scale; âËŸA 'c the GNDT database of active faults (Galadini et al., 2000); âËŸA 'c the ITHACA database (Michetti et al., 2000); âËŸA 'c selected active faults from complementary (complementary to what?) studies published by different authors.

Response 6: This is the core of our work and its strength point. As explained in the text (P. 6 L. 8-10), "the strength point of our approach is the assemblage of different fault datasets heterogeneously built for different purposes and based on different primary information and methods. In this approach, we consider all known faults (see above) to form a dataset as comprehensive as possible." In other words, all known faults are considered as potential seismogenic sources. As mentioned previously, many studies have demonstrated that faults considered inactive can be abruptly reactivated and be the source of damaging earthquakes (see examples and a comprehensive explanation at Response 5).

Comment 7: Notice that three of these studies refer explicitly to active - or at least capable - faults, but the other two sets, which happen to be the most numerous, do not. Also notice that Trippetta and co-workers subdivided each dataset into Class A (high quality), that includes ": : :exposed faults where subsurface and surface data allow for a detailed and reliable characterization of fault length: : :", and class B (low quality), containing ": : :buried and off-shore faults investigated mainly by seismic surveys, for which a precise characterization of fault length cannot be achieved: : :". Finally, notice that Trippetta and co-workers deliberately ignore the down-dip dimension (width) of the faults.

Response 7: Concerning the used heterogeneous dataset, please see our previous responses (Responses 4 and 6). We support our subdivision into Class A (high quality)
and Class B (low quality) faults as correct. Concerning the horizontal length/continuity of faults, indeed, the quality of exposed (accessible) faults (Class A faults in our work) is surely larger than that of buried/submerged faults (Class B faults in our work) that are detected solely through indirect techniques. It is therefore necessary to highlight this difference in the quality of fault datasets (Class A and Class B). All this is clearly explained at P. 10 L. 5-19 and 30-31, P. 12 L. 29-31, and P. 15 L. 6-10.

Concerning the down-dip dimension, the empirical relationship (between fault length and earthquake magnitude, Leonard, 2010) we used does not include the down-dip dimension of the fault. The reason is clearly explained in Leonard (2010). In particular, Leonard (2010) states: "A related problem is that published empirical relations are not self-consistent. By self-consistent, I mean relations that enable seismic moment, fault length, width, area, and displacement to be estimated from each other, with all these relations being consistent with the definition of seismic moment. That is, if you start with one parameter (e.g., fault length) and determine all of the others (area, width, displacement, moment), you retrieve the same set of parameters, no matter which parameter you start with." Leonard (2010) provides instead a self-consistent relationship. For this reason, we used the relationship by Leonard (2010), and retrieved the earthquake magnitude from the fault length.

Comment 8: A basic consideration is that by assembling faults from such different and nonhomogeneous sources, Petricca et al. inevitably put together a) alternative views on the same faults, possibly stemming from widely alternative conceptual models; b) faults that are mutually exclusive due to their geometry (typically, faults crossing each other in the subsurface: if one fault ends against another, its seismic potential based solely on length is largely overestimated); c) faults that cannot be simultaneously active, or reactivated, in the current stress regime; and d) blind faults whose actual length may be strongly biased by the availability and density of subsurface data.

Response 8: We think that this is the strength of our work. For the reasons explained above (Response 4) we propose that most of the faults within the Italian territory can

host an earthquake. Therefore a comprehensive fault dataset, as the one used in the present work, can help in: 1) reducing bias induced by the availability and density of subsurface data; 2) highlighting areas where detailed future studies are required to improve seismic hazard, that is not the target of this work.

All this is clearly stated in the Introduction and in the Conclusions sections. However, we will better stress the above-mentioned concepts in the Abstract, Introduction, and Conclusions of the revised manuscript. This is a product for scientific use only (i.e. testing the empirical relationships).

Comment 9: In our opinion it is extremely unsafe to derive the maximum earthquake magnitude from the length of a fault of which we do not even know (a) if it is active or if it can be reactivated (e.g. whether or not it cuts or deforms deposits of a specified age or it is suitably oriented with respect to the current stress regime), and (b) whether and how it is connected with a deep-seated seismogenic source. For decades earthquake geologists have investigated the surface expression of presumed active faults to gain insight into their long-term behaviour and seismogenic potential. The main purpose of such studies is to distinguish between active and non-active faults (sealed, truncated, or misoriented with respect to the current stress field). From this paper we learned that this is no longer necessary, as any mapped fault has the potential of generating a large earthquake.

Response 9: Valensise et alii may be right in stating that "it is extremely unsafe to derive the maximum earthquake magnitude from the length of a fault of which we do not even know …" But this statement has to be tested, quantitatively measured, and experimented. This is exactly what we try to do in our paper. After the experiment, we do not assert that it is extremely safe to derive the maximum earthquake magnitude from the length of a fault. We simply discuss our quantitative results and reach the following conclusions: "These results are partly encouraging and suggest the testing and validation of this experiment elsewhere. In case it is successfully validated elsewhere, this method would be useful as well as fast where historical and instrumental earthquake catalogues are limited, the returning time of (strong) earthquakes is possibly very long, and information on the age of faults is poor. This method cannot, however, be a substitute for time-dependent (paleo)seismological methods for seismic hazard assessments. Rather, it can rapidly provide an approximate perspective of time-independent seismic potential of faults and highlight areas where further detailed studies are required."

The statement by Valensise et alii asserting that from our paper "it is no longer necessary to distinguish between active and non-active faults" does not match our conclusion section where we state: "This method cannot, however, be a substitute for time-dependent (paleo)seismological methods for seismic hazard assessments." Notoriously, the main objective of paleoseismology is to distinguish between active and non-active faults.

Comment 10: Trippetta and co-workers propose a very simplistic approach to a rather complicated problem, based on criteria and assumptions that are not exposed in the text.

Response 10: All our criteria and assumptions are clearly exposed in Sections 3 and 4 (Pages 6-10).

Comment 11: The question to ask is: for how many of these earthquakes could the authors declare a positive relationship between the magnitude and the surface length of the presumed earthquake causative fault? âËŸA 'c probably for the Amatrice and Norcia shocks of the 2016 sequence; âËŸA 'c not for the 6 April 2009, Mw 6.3, Aquilano earthquake, for which the fault dataset supplied by Trippetta and co-workers in their Supplementary Data (Dataset-S1-ItalianFaults.kmz: Figure 1) shows a fault that is less than 1 km in length, reportedly taken from the Structural model of Italy and included in Class A. This fault is at least 15 times shorter than the fault necessary to generate a Mw 6.3 earthquake, and it does not even coincide with the surface ruptures observed following the 6 April 2009 mainshock; âËŸA 'c not for the 23 November 1980, Mw

6.8 Irpinia-Basilicata earthquake, for which the Class A subset of the dataset used by Trippetta and co-workers, derived from the Ithaca database, reports a 10 km-long fault; nearly four times shorter than the 38 km-long rupture reported in the classical paper by Pantosti and Valensise (1990), among those used by Wells and Coppersmith (1994) to derive their empirical relationships. At any rate, even this 10 km-long fault was not known prior to the earthquake; âËŸA ′c and finally, nor for the 1968 Belice, 1976 Friuli, 1984 Abruzzo, 1990 Potentino, 1997 Colfiorito, 2002 Molise, 2012 Pianura emiliana earthquakes, whose causative faults were most likely blind.

One could argue that the earthquakes that occurred over the past 50 years were not very large, on average, and hence are not representative of the problem at hand. But even the M 7+ earthquakes that have occurred in the early 20th century, the 1908 Messina Straits and the 1915 Avezzano (both of Mw 7.1 according to CPTI), provided contrasting evidence; âËŸA ′c no surface breaks have ever been reported for the 40-kmlong source of the 1908, Messina Straits earthquake, whose causative fault was most likely blind. Nevertheless, the Class A subset of the dataset used by Trippetta and coworkers, derived from GNDT, does report a 36-km-long offshore fault, probably based on geodetic evidence, showing that this dataset in fact reports a combination of inferred seismogenic sources and actual surface faults; âËŸA ′c for the 1915, Avezzano earthquake the Class A subset of the dataset used by Trippetta and co-workers, derived from both the Ithaca and GNDT datasets, reports a 15-km-long fault. This fault accounts only partially for the 30-50-km-long fault that is needed to justify a Mw 7.1 event. The reported faults appear as the surface projection of a deeper portion of the seismogenic fault and do not follow the reported coseismic breaks. In summary, it is likely that none of these earthquakes have been generated by a surface fault reported in Petricca et al.'s database and having the characteristics upon which the paper by Trippetta and co-workers is based. In other words, for none of these earthquakes could the magnitude be derived with confidence based solely on the reported length of their surface trace, which is systematically shorter than that observed (or inferred) after the earthquake. But if the proposed approach does not work on relatively large and

recent earthquakes for which there is good control on the causative fault and on the magnitude, why should it work for older events and, most importantly for future ones?

The largest earthquakes require an additional consideration. Virtually no 20-40-kmlong fault has ever been reported in the areas that were struck by the largest Italian earthquakes prior to their occurrence; but such long faults do occur in areas of more mature geology were seismicity is minimal or absent, such as in the Alps. This might indicate that currently active, seismogenic dip-slip faults (recall that most Italian upper crustal earthquakes are generated by normal, reverse or thrust faults) are normally characterised by a discontinuous surface expression. The established basic geology of the orogens seems to show that when their activity stops, some of these faults are uplifted and rejuvenated along with their associated landscape, making them appear more continuous, more mature and downright ominous, right at a time of their evolution when they are no longer a concern for seismic hazard.

At the opposite end of the spectrum lie some catastrophic Italian earthquakes of the past few centuries that occurred in foreland areas and that reportedly were not accompanied by surface faulting. Such is the case of the 1693 Eastern Sicily earthquake, the largest in the Italian earthquake catalogue (Mw 7.3).

Under these ill-posed assumptions (assuming that any mapped fault is potentially active, or can be reactivated; mixing faults that may be mutually exclusive; disregarding the post-orogenic rejuvenation of many faults, etc.), the results obtained by Trippetta and co-workers - and plastically represented in their Figures 11 and 12 - are all but unexpected. They show very high-magnitude values of the PEMM for the less-reliable Class B faults; a circumstance that can be simply explained considering that buried fault traces obtained by interpolating sparse data over wide areas will necessarily appear longer than faults within an exposed fault system. It is true that we may have yet to see some exceptionally large crustal earthquake, i.e. larger than any earthquake already known to the historical record; but without adopting any criteria regarding the likelihood of these large ruptures, such scenario is totally at odd with the unique historical, archaeological and palaeoseismological richness of the Italian earthquake record. This information provides the basis for a much more mindful way to estimating the magnitude of future large earthquakes, also in view of the frequency-magnitude distribution of earthquake occurrence.

Response 11: The discrepancy reported by Valensise et alii between the known faults and the historically/instrumentally-recorded earthquakes is precisely our aim and the subject of our experiment. That is, (1) we consider a large dataset of known/mapped easily-accessible faults in Italy, (2) we calculate the potential magnitude (PEMM) from the fault length, (3) compare the PEMMs with the historically/instrumentally-recorded earthquakes, (4) discuss this comparison and the pros and cons of the used method, and (5) conclude that "These results are partly encouraging and suggest the testing and validation of this experiment elsewhere."

In other words, the discrepancy mentioned by Valensise et alii is included in our comparison between PEMM and the historically/instrumentally-recorded earthquakes and represented in Figs. 11, 12, 13, and 14 and Table 1 (see also the related discussion at pages 11, 12, and 13).

Our aim is not at all to compute the expected magnitudes of earthquakes in Italy. Rather, it is to test the validity of those scaling relationships (between fault attributes and earthquake magnitude) when using available fault datasets at the national scale. This is clearly and unmistakably stated in our Introduction: "Our main aim is to test whether solely considering the known mapped faults (both active, inactive, and undetermined) and disregarding further information (e.g., historically- and instrumentally-recorded earthquakes as well as the regional stress field and strain rate) it is possible to provide, through existing seismic scaling laws of faults and earthquakes, reasonable assessments of the maximum possible earthquake magnitude over an entire nation. The resulting (assessed) magnitudes (PEMM) are compared (i.e., the mathematical difference) with catalogued earthquake magnitudes that are the only existing points of reference against which assessed magnitudes can be compared."

Yes, obviously in our dataset of faults there are problems connected with poorly known faults, or buried or segmented ones, as stated by Valensise et alii. Well, how do these problems affect the computation of the expected magnitude of earthquakes? We contribute to answer this question with Figures 11, 12, 13, and 14 and Table 1 (see also the related discussion at pages 11, 12, and 13). This is our job in this paper.

Concerning the area of Paganica-L'Aquila and Messina Straits mentioned by Valensise et alii, we remark that, regardless of the faults invoked by Valensise et alii, the magnitude obtained in our experiment (PEMM) is between 6.0 and 6.5 for the Paganica-L'Aquila area and larger than 7.0 for some cells of the Messina Straits. These results are consistent with the earthquakes mentioned by Valensise et alii, that is the 2009 Mw 6.3 L'Aquila earthquake and the 1908 Mw 7.1 Messina Straits earthquake.

Comment 12 The bottom line is that the approach proposed in the paper by Trippetta and co-workers goes against the knowledge acquired during decades of seismic hazard studies in Italy and the rest of the world. It does not predict anything useful but is potentially dangerous and very costly, if not put in the right perspective.

Response 12: Our aim is not at all prediction or forecasting. We simply discuss the results of our experiment. The seismic hazard is not included in this work (see our previous responses) and not recurrence time and probabilistic calculations are included.

Comment 13: The authors never admit that their results are pointless, even when they remark (Page 12) that "... the negative occurrences are very limited...", i.e. that the number of predicted magnitudes that are larger than those observed in the historical record outnumbers by far the opposite case, resulting in the very asymmetric pattern shown in Figure 12. In fact, they refer to a "...limited difference... between PEMMs and the catalogued earthquake magnitudes..." (!), neglecting the obvious consideration that magnitude is a logarithmic quantity, implying that a 0.2 increase in Mw, for example from 6.0 to 6.2, doubles the seismic moment. For a typical continental fault having an aspect ratio in the range 2-3 and standard scaling for coseismic slip, doubling the

moment implies that fault length may increase by over 20%. For a magnitude increase of 0.5, for example from 6.0 to 6.5, the seismic moment becomes 5.6 times larger, which may require a fault that is 100% or more longer than that necessary to generate the smaller earthquake.

Response 13: Thanks. We accept this comment. We will rephrase this statement (the negative occurrences are very limited...). We agree that it is useless commenting this result as very limited. We will susbstitute "very limited" with the exact numbers that we obtain from the experiment. The term limited is too vague and subjective. However, once again, it is worth to underline that we are not proposing a forecast method, we are testing very well-known scaling laws at the national scale.

Comment 14: In commenting their rather disappointing results the authors invoke potential incompleteness of the earthquake catalogue, which is always possible locally, but is also very unlikely for a work that considers the whole of Italy and all the data available in the richest earthquake catalogue worldwide. This record has certainly missed specific events but is reliable enough to constrain the frequency-magnitude distribution of Italian seismicity. A piece of information that a study about the estimates of earthquake magnitude cannot neglect.

Response 14: By stating that "this record has certainly missed specific events", Valensise et alii confirm that invoking a potential incompleteness of the earthquake catalogue (as we did) may be correct.

Comment 15: Trippetta and co-workers conclude by stating that "...more detailed studies should be developed...". This is always a good thing to do, except for the fact that detailed studies of Italian faults do exist and are often more accurate and to the point with respect to what is proposed here.

Response 15: This statement by Valensise et alii confirms that our experiment was really necessary. Valensise et alii believe indeed that detailed studies of geological structures are good enough and sufficient. Results from our experiment (see in particular Figs 10, 11, 12, and 13 and Table 1), in contrast, show that "more detailed studies should be developed". We refer, in particular, to the discrepancy between the PEMMs and the historically/instrumentally-recorded magnitudes of earthquakes.

The experiment that we realized is a test also for other countries. One of the pillars of our experiment is its RAPIDITY. We consider indeed fault databases easily available and accessible, testing how much they are reliable to infer the earthquake magnitude. This is a completely different job from the DISS database (run by Valensise et alii) that has being implemented for almost 20 years. Our aim is to test whether we can infer possible earthquake magnitudes from available and easily-accessible datasets of faults.

Comment 16: The conclusions of this paper are worrisome, in consideration of the large number of areas where the authors envision the possibility of M 7.5 and larger earthquakes, that is to say earthquakes bigger than the largest magnitude ever recorded in Italy, without any consideration as to how frequently this may occur. In a standard PSHA approach these large magnitudes would be assigned a very low probability of occurrence, leading to a minimal statistical impact on the expected ground shaking for short average return periods. The information about the possible largest earthquakes may generate a great deal of confusion if not appropriately communicated. We cannot imagine how the residents of Bologna, Ancona, Pescara, but also Padua, Trento, Vicenza and even Venice, cities lying in areas that are currently considered mid- to low-hazard, would react to knowing that very large earthquakes may occur below their feet at any moment.

Response 16: We completely agree with the sentence "The information about the possible largest earthquakes may generate a great deal of confusion if not appropriately communicated". In fact, we are NOT proposing a new seismic hazard map, and this is really clear by reading the paper (not only looking at the figures). We simply compute potential earthquake magnitude from fault size and compare these results with seismic catalogs (Figs. 10-13 and Table1) to reason upon the validity of the scaling relation-

[Figure]

ships between fault attributes and earthquake magnitude at the national scale with available fault datasets. The comment by Valensise et alii is therefore inappropriate.

However, in the revised manuscript we will better stress the above-mentioned concepts (i.e., our aim is not a new and reliable map of expected earthquakes in Italy). We will also add disclaimers in Figures 7 and 8 saying that these figures are not at all for use for Civil Protection, officials, and, in general, for seismic hazard. They are a product for scientific use only (i.e. testing the empirical relationships).

Comment 17: Another major flaw in the approach taken by Trippetta and co-workers lies in their discretisation of seismogenic zones into 25x25 km sub-areas. Of course, some discretisation is inevitable, but one has to be aware that a 25x25 km cell may host a 35 km-long fault, at the most. According to the equation proposed by Leonard (2010), the empirical law adopted by Trippetta and co-workers, a 35 km fault length corresponds to a Mw 6.8 earthquake. Hence, any larger earthquake will necessarily encompass two or more cells. A close inspection of Figures 7 and 8 of the paper, however, reveals that several cells filled in red or dark red, which according to the adopted colour-coding should correspond to an expected Mw in the range 7.4 to 7.8, occur isolated, i.e., surrounded by cells for which the expected PEMM is much smaller. According to same equation by Leonard (2010), this magnitude range corresponds to a fault length in the range 78 to 135 km, which should involve a minimum of 2 to 4 adjacent cells, depending on fault strike. An isolated cell capable of a Mw 7.4 earthquake is hence a seismological paradox that has no physical meaning, as the earthquake causative fault will necessarily extend to adjacent cells.

Response 17: At which cell are Valensise et alii referring to? As we can see from figure 7 and 8 there are no red (M.7.5) isolated cells. In each cell, we consider the longest fault that touches/crosses the cell. This means that the faults can be longer than 35 km as indicated by Valensise et alii. This is clearly stated in our method section at Lines 27-28 Page 8: "The length of the longest fault crossing each cell determined the parameter "fault length" (Lf) of the considered cell." Our computation

is selfsustained and the complete database is available at the following link for reproducibility of our results (see also the Data availability section): http://pmd.gfz-potsdam.de/panmetaworks/review/924b171fd21c78f295d58a7e9e321e8ad07667ab6201634b23d3cb5a3f170d10/

Comment 18: We are a group of INGV seismologists and earthquake geologists who regularly provid active faulting data and seismogenic models to Italian, European, and global SHA practitioners, and to the Italian Civil Protection authorities. As such we are especially concerned that inaccurately collected fault data, inconsistent elaborations and unjustified conclusions such as those presented by Trippetta and co-workers may be implicitly validated by appearing in a respectable journal such as Solid Earth, thus becoming embedded in the literature. In addition to that, chasing the problem of the occurrence of very large earthquakes with an overly simplistic approach is the most effective way to shift our attention from the areas where earthquakes in the Mw range 6-0-7.0 are more likely to hit. These areas include most of the eastern Alps, various portions of the Apennines, most of Calabria and several parts of Sicily. Encouraging earthquake retrofitting in these areas should be the main target of any responsible seismological community.

Response 18: We do not see the scientific question posed by this comment and at the same time we are aware that Valensise et al., are INGV seismologists and earthquake geologists working on active faulting for SHA. We acknowledge the expertise of the Authors on this topic, but at the same time we emphasise the conflict of interest they have maintaining the monopoly of fundings on the seismic hazard in Italy since decades.

---

## Short Comment (SC3) · 23 Dec 2018

Submitted by Roberto Basili, Pierfrancesco Burrato, Michele Carafa, Francesca Cinti, Riccardo Civico, Paolo Marco De Martini, Deborah Di Naccio, Umberto Fracassi, Fabrizio Galadini, Vanja Kastelic, Daniela Pantosti, Mara Monica Tiberti, Gianluca Valensise, and Paola Vannoli

In an attempt to defend the manuscript by Trippetta et al., Andrea Billi (the corresponding author) and his coworkers only succeeded in worsening the fallacies we pointed out in our comment. Importantly, they contradict themselves by repeating that their work is not aimed to analyse the seismic hazard but it is rather a "product for scientific use

only", and then admit that "this study developed within the framework of nuclear waste repositories planning".

Regrettably, Billi and coworkers concluded their defense with the defamatory statement reported below:

"We acknowledge the expertise of the Authors on this topic, but at the same time we emphasise the conflict of interest they have maintaining the monopoly of fundings on the seismic hazard in Italy since decades."

This statement is unacceptable and demands that we suspend here our scientific interaction with Billi and coworkers up until they retract their allegations, or specify which aspect of the Copernicus Publications' Competing Interest Policy (https://www.solid-earth.net/about/competing_interests_policy.html) they presume we have violated, providing the necessary evidence. We otherwise reserve the right to take any action to defend our good reputation on this matter.

———————————————————

---

## Short Comment (SC4) · 23 Dec 2018

As executive editor for the SE Discussion manuscript Se-2018-98, I would first remark that the open-access platform of SE is intended to ensure all the different opinions are adequately represented during the editorial workflow, providing a fair and unbiased review process.

The open access policy of comments and replies in SE is also conceived to foster scientific debate on relevant scientific issues such as the ones presented in this SED manuscript. In such a way, SE Discussion provides the possibility for anyone, not just the reviewers, to make constructive comments suitable to help the relevant Topical

[Figure]

Editor come to decide on the manuscript. Given the societal relevance of the topics presented in the manuscript, the ongoing debate can also potentially help to make further progress in the state of knowledge on such a sensitive issue.

Nonetheless, following the SE editorial policy, as executive editor of the manuscript I have also the duty to censor "comments that are not of substantial nature or of direct relevance to the issues raised in the discussion paper or which contain personal insults, especially if Authors. . .notify the editor in case of abusive comments."

Therefore, although I am aware that there is a vigorous debate on these topics in Italy that is not only confined to the scientific community, I do believe that comments on scientific articles should be instead confined only to scientific issues. I would also emphasize that comments that regards interpersonal relationships and aspects of the professional reputation of the Author(s) (or anyone else) involved in the debate are to be avoided, because not in discussion here.

To conclude, I would thus recommend that anyone that adds further interactive comments to strictly comment the science of the manuscript and nothing else.

Sincerely,

Federico Rossetti

---

## Author Comment (AC2) · 27 Dec 2018

By Fabio Trippetta, Patrizio Petricca, Andrea Billi, Cristiano Collettini, Marco Cuffaro, Davide Scrocca, Giancarlo Ventura, Andrea Morgante, Fabio Chiaravalli, and Carlo Doglioni

As scientists, we would like to maintain this discussion on a scientific basis. Below, we will attempt to do so briefly.

Firstly, the history of science is disseminated by many occurrences, in which purely scientific results and discoveries derived or developed from applicative activities and

[Figure]

vice-versa. Our previous reply to Valensise et alii is not therefore contradictory as they stated.

Secondly, as we did not intend to be offensive towards Valensise et alii, we retract with pleasure our statement indicated by Valensise et alii as potentially offensive ("We acknowledge the expertise of the Authors on this topic, but . . . the seismic hazard in Italy since decades.").
* * *

---

## Referee Comment (RC2) · Nandan (Referee) · 8 Jan 2019

In general, I have found the work presented in the manuscript to be very clearly described. I also like the style of writing of the authors as they have stated the caveats of the study very clearly. However, I fail to appreciate the novelty of the work presented in this paper. I am wondering if it is the compilation of the comprehensive fault catalog using the existing databases or the estimation of the PEMMs. If it is the former, I would suggest that the authors stress it more in the manuscript and show what were the hurdles that they had to overcome when compiling the comprehensive fault database. If it is the latter then, I feel that the authors have oversimplified the task of estimation

of PEMMs. I the following I outline some of my main comments that have led me to the concerns raised in the previous paragraph. 1. What are the main challenges in compiling the comprehensive fault database from existing fault databases? How is this task difficult? 2. How do the authors identify and remove the duplicate faults in the regions where the two databases overlap? 3. How sensitive are the results of the authors to the assumptions described in section 4.2? For instance, would the results dramatically change if one considers a different grid resolution? Same applies for the other assumptions. I think authors should do more effort than just outlining their assumptions. A sensitivity analysis is a minimum they should strive for. 4. The authors claim that the calculated PEMMs are consistent with the largest observed earthquakes at least for the geologically well-constrained fault. First of all, it is obvious that this consistency is strongly dependent on the grid resolution that the authors will choose. Secondly, what is the reference level for consistency? What I mean to say is any prediction and observation can be deemed consistent if allow for enough uncertainty. To account for this, one needs to come up with a reasonable null hypothesis and compare the new predictions to the predictions of the null hypothesis. In this case, a reasonable null hypothesis could be an untruncated Gutenberg Richter law, with a given b-value. The authors could pose their model as the GR law with the same b-value but with the truncation at the PEMMs estimated using their approach. They can then estimate the likelihood of the largest earthquakes (M>M_threshold) and compare the two likelihoods using standard statistical tests. In this manner, the authors would have reference level that would allow them to objectively assess the quality of their prediction.

---

## Author Response (AR1)

**Our Responses to Referees' and Colleagues' Comments**

Dear Editors,

Please, find below our responses to all Referees' comments and related explanations for changes in the text, figures, and tables.

We thank very much the two Referees, the Editors, and all colleagues who provided constructive comments in the Discussion Forum.

Main changes in the submitted material are the following ones: following the Referee2 suggestions, we provided additional analyses on the sensitivity of the used cell size (Lines 16 onward p. 9, Figs. S1, S2, and S3) and on the null hypothesis analysis (Lines 9 onward p. 14, Fig. 14, Tables S1 and S2). Please, see below for explanations about these additional analyses.

Concerning the comments provided by Valensise et alii, we have already replied to all these comments in the Discussion Forum. Below, we report only those comments (with our responses and explanations) by Valensise et alii that involved changes in our submitted material.

We acknowledge also changes in the panel of authors. Dr. Fabio Chiaravalli stepped out as author for reasons connected with his Company (Sogin). Dr. Anna Maria Lombardi (annamaria.lombardi@ingv.it), who is a mathematician from INGV-Rome expert in statistical analysis of seismological data, joined this panel of author performing, in particular, some new statistical analyses requested by Referee2 (see Figure 14, and Tables S1 and S2).

Looking forward to hearing from you

Sincerely

Andrea Billi and co-authors

Rome, March 2019

**Referee #1**

**Comment 1**

*Main comments*

*This manuscript describes an approach to evaluate the maximum possible earthquake*

*magnitude from the geometry of active faults in Italy. The topic has a broad interest, in*

*particular for fault-based seismic hazard modelling, because the correct evaluation of the*

*seismic potential of a seismogenic source is one of the main required parameters for seismic*

*hazard studies. The use of active faults in seismic hazard assessment has become extensive in*

*the last decades due to efforts of data compilation and analysis. Active faults provide the*

*information to extend the observational time of large magnitude earthquakes, which often is*

*not captured by the existing catalogues of observed seismicity.*

*The authors apply one empirical scaling relationship (not correctly defined by the authors as*

*scaling law!) between fault length and expected magnitude, to a clearly incomplete and*

*inhomogeneous fault database, to evaluate the maximum possible earthquake magnitude in*

*Italy. The manuscript is mostly well written and the figures are clear but, from my point of*

*view, the approach has several misconceptions and incompleteness and the conclusions are*

*not supported by the results.*

*For these reasons, the manuscript does not represent a substantial contribution to scientific*

*progress in seismic hazard assessment and seismic risk reduction, as required by a high-level*

*Journal as Solid Earth. The applied methods are valid but too simplistic and applied to*

*incomplete and inhomogeneous data with consequently a poor scientific significance of the*

*manuscript. Moreover, the references used for the fault database compilation is largely*

*incomplete.*

**Response 1**

We acknowledge that the main goal of this paper is not the seismic hazard that is defined as:

"the probability that an earthquake will occur in a given geographic area, within a given window of time, and with ground motion intensity exceeding a given threshold. With a hazard thus estimated, risk can be assessed and included in such areas as building codes for standard buildings, designing larger buildings and infrastructure projects, land use planning and determining insurance rates.";

nor is the seismic risk that is defined as:

"the risk of damage from earthquake to a building, system, or other entity. It particular, it can be also defined, for most management purposes, as the potential economic, social and environmental consequences of hazardous events that may occur in a specified period of time."

Our main goal is different and is clearly stated (now improved in the new version) several times in the manuscript:

- In the title: "From mapped faults to Fault-Length Earthquake Magnitude (FLEM): A test on Italy with methodological implications"

- At Lines 1-4, p. 1: "Empirical scaling relationships between fault/slip dimensions and earthquake magnitudes are often used to assess the maximum possible earthquake magnitude of a territory. In this paper, upon the assumption of the reactivability of any fault, these seismic scaling relationships are benchmarked at the national scale in Italy against catalogued earthquake

magnitudes, considering all known faults regardless of their age, stress field orientation, strain rate, or else."

- At Lines 15-19, p. 1: "The main advantages of this method is its independence from temporal and (paleo)seismological information, whereas the main novelty is its use at the national scale also for faults considered inactive. Our work can provide a perspective time-independent seismic potential of faults; however, it cannot be a substitute for time-dependent (paleo)seismological methods for seismic hazard assessments."

- At Lines 13-22, p. 3: "We anticipate that, with this work, we do not intend to propose an alternative method for seismic hazard assessment or to better previous methods (e.g., Giardini et al., 1999; Jimenez et al., 2001; Michetti et al., 2005  Field et al., 2009, 2015; Reicherter et al., 2009). Our main aim is to test whether solely considering the known mapped faults (both active, inactive, and undetermined) and disregarding further information (e.g., historically- and instrumentally-recorded earthquakes as well as the regional stress field and strain rate) it is possible to provide, through existing seismic scaling relationships of faults and earthquakes, reasonable assessments of the maximum possible earthquake magnitude over an entire nation. The resulting (assessed) magnitudes (FLEMs) are compared (i.e., the mathematical difference) with catalogued earthquake magnitudes that are the only existing points of reference against which assessed magnitudes can be compared. Note that these results should be considered more in a theoretical and methodological perspective for comparison with future similar studies rather than in an applicative perspective for the case of Italy. In particular, our assessed earthquake magnitudes (FLEMs) for the Italian territory are proposed in this paper for scientific reasons and not for their use for civil protection and prevention purposes."

Therefore, our manuscript is not at all intended to be "a substantial contribution to scientific progress in seismic hazard assessment and seismic risk reduction" as stated by Referee 1. The possible perspective of this manuscript is now better stated at 13-22, p. 3.

Note also that we have modified the term FLEM (Potential Earthquake Maximum Magnitude) – that seemed a term for the seismic hazard assessment – into the term FLEM (Fault-Length Earthquake Magnitude) – that should merely represent what we have done in this work, i.e. computing the earthquake magnitude from the length of mapped faults through known empirical relationships.

We have now used the term scaling relationship rather than scaling law over the entire manuscript as suggested by Referee 1.

Following suggestions by Referee 2, the applied method has been significantly improved in the new version (e.g.,see the new Figures 14, S1, S2, and S3, and Tables S1 and S2), whereas the datasets and references are not incomplete as explained below.
* * *
**Comment 2**

*Detailed comments*

*1. Fault database: the database is largely incomplete and inhomogeneous, with an*

*incompleteness variable in space. The suggestions to improve the database are:*

*a. Consider the abundant literature in the compilation of active fault database for*

*Italy, mostly more recent than the used one (in the following references a*

*partial and not yet complete list of papers not considered in the manuscript);*

**Response 2**

As it is now specified in the manuscript (Lines 27-30, p. 8), our fault database is a big compilation of faults (total number of faults = 12467; specifically, 9169 A-type faults and 3298 B-type faults) from the main available datasets (from Line 5 onward, p. 6).

Obviously, any national fault database can be improved, but, in our case, considering 12467 faults in Italy as an incomplete database seems to be an underestimate. Note that, as stated in the Introduction, our focus is not on active faults (from Line 22 onward, p. 2).

Therefore, although our database can be improved with any single active fault discovered and published in the new literature, such literature-hunting is beyond the scope of our work, which becomes significant when done rather quickly with the available fault datasets; otherwise, other existing datasets (accurately compiled over long times) of active faults and/or seismogenic sources are already available for Italy (e.g., DISS, http://diss.rm.ingv.it/diss/).

Concerning the inhomogeneity, we specify what follows (from Lines 10 onward, p. 6): "The strength point of our approach is the assemblage of different fault datasets heterogeneously built for different purposes and based on different primary information and methods. In this approach, we consider all known faults (see above) to form a dataset as comprehensive as possible. Moreover, although different, the common point of all used datasets is that they have faults mapped and therefore measurable over the Earth's surface."
* * *
**Comment 3**

*b. Separate in the database the recognized active faults to the not clearly active*

*ones. In literature are available several definitions of fault activity, taking into*

*account the age of the involved deposits, the associated earthquakes, the*

*continuity and kinematics compatibility, and many others, and the authors need*

*to consider it. In this way the authors could define more classes of faults (more*

*than the two defined in the manuscript) based on the goodness of data and*

*recent activity and need to treat separately the classes in the approach to*

*evaluate the seismogenic potential;*

**Response 3**

Please, see our previous response.

Our scope is not working only on (presumably-)active faults, but working on all faults and on their potential over future long terms (from Line 26 onward, p. 2, and from Line 12 onward, p. 3).
* * *
**Comment 4**

*c. Evaluate and handle with the spatial variable incompleteness of the database;*

**Response 4**

Our dataset (12467 faults) cannot be considered incomplete (please, see Response 2). However, following the suggestions by Reviewer 2, we have now added a sensitivity analysis on the used cell size (Line 16 onward, p. 9 + Figures S1, S2, and S3). This analysis should help in better understanding the spatial relevance of our test and results as suggested by the Reviewers.
* * *
**Comment 5**

*Consider the fault segmentation variability in the correct evaluation of the seismogenic*

*potential, essential in fault-based seismic hazard approaches, as confirmed by recent*

*complex coseismic ruptures (e.g., 2010 M 7.1 Canterbury, 2012 Mw 8.6 Sumatra, 2016*

*Mw 7.8 Kaikōura, 2016 Mw 6.5 central Italy);*

**Response 5**

On one hand, our input is simply the fault length in map view (Line 25 onward, p. 8: "Starting from the entire dataset of faults in Italy, as a first step, we measured the length of each fault as the real fault trace length in map view, i.e., the length of the vertical projection of the fault trace as observed on the Earth's surface over a horizontal plane (Fig. 2; supplement; Petricca et al., 2018)."). Therefore, fault segmentation is already properly considered when properly mapped in the original datasets used for this work.

On the other hand, the lack of a proper segmentation in the process of fault mapping is right one of the targets of our analysis. In other words, where the computed FLEM (largely) exceeds the corresponding catalogued earthquake magnitude, the most probable cause for such excess is the lack of high-resolution datasets that allow characterizing fault geometry and in particular segmentation. The most straightforward example in our study are the class B faults. Therefore our study is useful to detect areas where faults have not been properly characterized (i.e., segmented), these areas require further detailed studies for a better comprehension of the seismic potential.

To this end (fault segmentation), we have also added this relevant statement at Lines 2 onward, p. 14:

"To this end, it is also noteworthy that studies on the 2016 Amatrice-Norcia (central Italy) earthquakes (Mw 6.0 and 6.5) revealed that the length of the causative faults was only partially activated by the seismogenic slip (e.g., Cirella et al., 2018); however, as this co-seismic behaviour of faults seems rather frequent (Freymueller et al., 1994; Milliner et al., 2016; Chousianitis and Konca, 2018), it is most likely that this same behaviour is incorporated and implicitly expressed by the above-mentioned empirical scaling relationships between fault length and earthquake magnitude (e.g., Wells and Coppersmith, 1994; Leonard, 2010; Thingbaijan et al., 2017)."
* * *
**Comment 6**

*Organize a table with earthquake-fault associations, in order to avoid the double*

*counting or the source missing. There are several examples of 'problems' in the *kmz*

*of the authors, as the missing of the Paganica fault, responsible of the M6.3 2009*

*L'Aquila earthquake, the double counting of some faults in the Fucino area where the*

*M7 1915 earthquake occurred, and the not correct definition of the total length of the*

*fault responsible in the Irpinia region of the M6.8 1980 earthquake;*

**Response 6**

As previously explained, we consider all faults (from a number of existing datasets) disregarding their age and/or their association with historical/instrumental earthquakes (Lines 2-5 p. 1, 14-16 p. 3, and 10-12 p. 6).

We do not double-count faults. Our only criterion to choose the fault from which the FLEM of the considered cell will be computed is the greatest fault length in map view. In such a way, only one fault (the longest one) will provide the FLEM in a given cell. Therefore, faults cannot be and are not double-counted (Line 30 onward p. 8).
* * *
**Comment 7**

*Consider also the seismogenic depth and the length of the faults along dip, to better*

*define the total potential rupture area, better linked to the seismogenic potential of the*

*sources;*

**Response 7**

These parameters are available only for a limited number of (active) faults in Italy, whereas we consider all faults over the national territory (Lines 2-5 p. 1, 14-16 p. 3, and 10-12 p. 6). Moreover, the input of the used empirical relationship by Leonard (2010) is simply the fault length in map view (Line 5 onward p. 9).
* * *
**Comment 8**

*Deal with greater accuracy the empirical scaling relationships, by:*

- *a. Compare the results of the approach using the different available relationships;*
- *b. Handle with the uncertainties, both inter- the different relationships and intrathe single relationships (sometimes the authors define very large standard deviation in the empirical relationship not treated in the manuscript);*

**Response 8**

This comparison is done in Fig. 6 and at Lines 16 onward p. 10. Moreover, the new analyses concerning the cell size sensitivity (Figs. S1-S3 and Lines 16 onward p. 9) and the null hypothesis (Fig. 14, Tables S1 and S2 and Lines 9 onward p. 14) provide new robustness and soundness to our results.
* * *
**Comment 9**

*Compare the results using different geometrical parameters, e.g. the surface*

*rupture length, the subsurface rupture length, the rupture area (and so*

*considering the seismogenic thickness) together with the different kinematics,*

*treated separately in the different scaling relationships;*

**Response 9**

The one proposed by the Reviewer is a totally different approach. We only consider the length of all known faults. As stated in the Conclusions section, this approach has its pros and cons: "Our results are partly encouraging and suggest the testing and validation of this experiment elsewhere. This method cannot, however, be a substitute for time-dependent (paleo)seismological methods for seismic hazard assessments. Rather, it can provide an approximate perspective time-independent seismic potential of faults and highlight areas where further detailed studies are required."

For what concerns different kinematics, we refer the reader to Lines 11 onward p. 10.

Note also that, for the Italian territory, the influence of the seismogenic thickness on the potential earthquake magnitude has already been treated in recent articles by some of us (Petricca et al., 2015, Tectonophysics 656, 202-214; Chiarabba and De Gori, Terra Nova, 2016; Petricca et al., 2018, Physics of the Earth and Planetary Interiors 284, 72-81).
* * *
**Comment 10**

*Handle with the uncertainties in the results, comparing the differences between the*

*seismogenic potential of the faults estimated by the empirical relationships and the*

*earthquakes in the historical catalogue, in terms of seismic moment and not only*

*magnitude. Magnitude is a logarithmic quantity and so a simple comparison as done by*

*the authors in the conclusions has a clear bias;*

**Response 10**

Although the suggestion is surely right from a theoretical point of view, practically, the problem of using seismic moments is that for many earthquakes (particularly in historical catalogs but also in old instrumental catalogs), independent assessments of the seismic moments do not exist. This makes impossible the use of seismic moments in our approach, which includes historical and instrumental earthquake catalogs for an entire nation. In other words, the only parameter available for the entire nation over the historical and instrumental periods is the earthquake magnitude and not the seismic moment.
* * *
**Comment 11**

*Treat the probability of occurrence in the conclusions. In seismic hazard models is*

*necessary to define the seismic rates for the different magnitude classes and so the*

*probability of occurrence of a defined magnitude depends on the average recurrence*

*time of that value in a specific area. The conclusions of the authors show the largest*

*expected magnitudes in areas with very low seismicity, like Sardinia, suggesting there*

*high seismic hazard values. Such conclusions have to be more strongly supported by*

*considerations in terms of probability of occurrence.*

**Response 11**

As previously explained (Response 1), the seismic hazard and the probability of earthquake occurrence are not the scope of our work. Consider that, in a long term perspective (e.g. IAEA, www.iaea.org; Lines 29 onward p. 2), any fault could be reactivated. Our scope is stated at Lines 14 onward p. 3:

"Our main aim is to test whether solely considering the known mapped faults (both active, inactive, and undetermined) and disregarding further information (e.g., historically- and instrumentally-recorded earthquakes as well as the regional stress field and strain rate) it is possible to provide, through existing seismic scaling relationships of faults and earthquakes, reasonable assessments of the maximum possible earthquake magnitude over an entire nation."

**Referee #2 (Nandan)**

**Comment 12**

*In general, I have found the work presented in the manuscript to be very clearly described.*

*I also like the style of writing of the authors as they have stated the caveats of*

*the study very clearly. However, I fail to appreciate the novelty of the work presented*

*in this paper. I am wondering if it is the compilation of the comprehensive fault catalog*

*using the existing databases or the estimation of the FLEMs. If it is the former, I would*

*suggest that the authors stress it more in the manuscript and show what were the hurdles*

*that they had to overcome when compiling the comprehensive fault database. If*

*it is the latter then, I feel that the authors have oversimplified the task of estimation of FLEMs.*

**Response 12**

We thank Dr Nandan for its appreciation.

The scope and novelty of our work are not in the compilation of the fault database, rather they are in the estimation of FLEMs using all known faults over a national territory and in the comparison between FLEMs and catalogued earthquake magnitudes. These concepts are clearly stated and now better emphasized in the manuscript:

- In the title: "From mapped faults to Fault-Length Earthquake Magnitude (FLEM): A test on Italy with methodological implications"

- At Lines 1-4, p. 1: "Empirical scaling relationships between fault/slip dimensions and earthquake magnitudes are often used to assess the maximum possible earthquake magnitude of a territory. In this paper, upon the assumption of the reactivability of any fault, these seismic scaling relationships are benchmarked at the national scale in Italy against catalogued earthquake magnitudes, considering all known faults regardless of their age, stress field orientation, strain rate, or else."

- At Lines 15-19, p. 1: "The main advantages of this method is its independence from temporal and (paleo)seismological information, whereas the main novelty is its use at the national scale also for faults considered inactive. Our work can provide a perspective time-independent seismic potential of faults; however, it cannot be a substitute for time-dependent (paleo)seismological methods for seismic hazard assessments."

- At Lines 13-22, p. 3: "We anticipate that, with this work, we do not intend to propose an alternative method for seismic hazard assessment or to better previous methods (e.g., Giardini et al., 1999; Jimenez et al., 2001; Michetti et al., 2005  Field et al., 2009, 2015; Reicherter et al., 2009). Our main aim is to test whether solely considering the known mapped faults (both active, inactive, and undetermined) and disregarding further information (e.g., historically- and instrumentally-recorded earthquakes as well as the regional stress field and strain rate) it is possible to provide, through existing seismic scaling relationships of faults and earthquakes, reasonable assessments of the maximum possible earthquake magnitude over an entire nation. The resulting (assessed) magnitudes (FLEMs) are compared (i.e., the mathematical difference) with catalogued earthquake magnitudes that are the only existing points of reference against which

assessed magnitudes can be compared. Note that these results should be considered more in a theoretical and methodological perspective for comparison with future similar studies rather than in an applicative perspective for the case of Italy. In particular, our assessed earthquake magnitudes (FLEMs) for the Italian territory are proposed in this paper for scientific reasons and not for their use for civil protection and prevention purposes."

Since the focus of our work is the FLEM estimation and the comparison between FLEMs and catalogued earthquake magnitudes, to avoid an oversimplification of this task (as stated by Referee 2), we have followed the Referee's suggestions and provided additional analyses on the sensitivity of the used cell size (Lines 16 onward p. 9, Figs. S1, S2, and S3) and on the null hypothesis analysis (Lines 9 onward p. 14, Fig. 14, Tables S1 and S2). Please, see below for explanations about these additional analyses.
* * *
**Comment 13**

*1. What are the main challenges in*

*compiling the comprehensive fault database from existing fault databases? How is this*

*task difficult?*

**Response 13**

As stated in our previous response (Response 12), our scope and novelty is not the fault database that is a compilation from existing databases as thoroughly explained at Lines 5 onward p. 6.
* * *
**Comment 14**

*2. How do the authors identify and remove the duplicate faults in the*

*regions where the two databases overlap?*

**Response 14**

Please, see above our Response 6. We do not double-count faults. Our only criterion to choose the fault from which the FLEM of the considered cell will be computed is the greatest fault length in map view. In such a way, only one fault (the longest one) will provide the FLEM in a given cell. Therefore, faults cannot be and are not double-counted (Line 30 onward p. 8).
* * *
**Comment 15**

*3. How sensitive are the results of the*

*authors to the assumptions described in section 4.2? For instance, would the results*

*dramatically change if one considers a different grid resolution? Same applies for the*

*other assumptions. I think authors should do more effort than just outlining their assumptions.*

*A sensitivity analysis is a minimum they should strive for.*

**Response 15**

Following the Referee's suggestion, we have performed a sensitivity analysis on the grid size (Lines 16 onward p. 9, Figs. S1, S2, and S3).
* * *
**Comment 16**

*4. The authors*

*claim that the calculated FLEMs are consistent with the largest observed earthquakes*

*at least for the geologically well-constrained fault. First of all, it is obvious that this*

*consistency is strongly dependent on the grid resolution that the authors will choose.*

*Secondly, what is the reference level for consistency? What I mean to say is any prediction*

*and observation can be deemed consistent if allow for enough uncertainty. To*

*account for this, one needs to come up with a reasonable null hypothesis and compare*

*the new predictions to the predictions of the null hypothesis. In this case, a reasonable*

*null hypothesis could be an untruncated Gutenberg Richter law, with a given b-value.*

*The authors could pose their model as the GR law with the same b-value but with the*

*truncation at the FLEMs estimated using their approach. They can then estimate the*

*likelihood of the largest earthquakes (M>M_threshold) and compare the two likelihoods*

*using standard statistical tests. In this manner, the authors would have reference level*

*that would allow them to objectively assess the quality of their prediction.*

**Response 16**

Following the Referee's suggestion, we have performed a statistical test to check, as far as possible, the reliability of our estimated FLEM values (Lines 9 onward p. 14, Fig. 14, Tables S1 and S2).

It is known that, even for excellent data and weak hypotheses (as the untruncated Gutenberg Richter), from an earthquake catalog alone it is substantially impossible, through a statistical test, to discriminate, with sufficient confidence, among competitive maximum magnitude values (Holschneider et al., 2014). This is mainly due to the upper cutoff of the Gutenberg-Richter distribution, where only rare earthquakes with magnitudes close to the maximum possible value

occur. In light of these limitations, we are able to check the reliability of the estimated FLEM values, but we cannot compare different competitive FLEMs.

Therefore, our analysis consisted in the following steps. Firstly, we selected cells for which the doubly truncated Gutenberg-Richter law, with a b-value equal to 1, could not be rejected. In this way, we excluded that a possible rejection of the estimated FLEM was actually due to the unreliability of the Gutenberg-Richter law or to the uncertainty about the b-value. Then, in the reliable cells, we tested the FLEM values, assuming them as null hypothesis. We found that in all the analyzed cells they cannot be rejected, both in the case of the CSIv1.1 catalog and in the case of the ISIDe catalog. However, we cannot exclude that the used data may be inadequate to reveal failure of the FLEM values, nor, for the above-mentioned reasons, we can compare alternative reliable FLEM values. This new analysis is now fully reported and explained at Lines 9 onward p. 14, Fig. 14, Tables S1 and S2.

**Valensise et alii**

**Introduction**

Comments by Valensise et alii, as happens also for most comments by Referee 1, seem influenced by an initial and radical misunderstanding. Our manuscript IS NOT titled "A new map of expected earthquake magnitudes for seismic hazard and risk mitigation in Italy Simply, this is not the goal of the article. Our maps (Figs. 7 and 8) ARE NOT AT ALL predictive maps of seismic hazard.

Our manuscript is titled "From mapped faults to earthquake magnitude: A test on Italy with

methodological implications" and our aim is stated clearly from the introduction where we say:

"We anticipate that, with this work, we do not intend to propose an alternative method for seismic hazard assessment or to better previous methods (e.g., Giardini, 1999; Jiménez et al., 2001; Michetti et al., 2005; Field et al., 2009, 2015; Reicherter et al., 2009). Our main aim is to test whether solely considering the known mapped faults (both active, inactive, and undetermined) and disregarding further information (e.g., historically- and instrumentally-recorded earthquakes as well as the regional stress field and strain rate) it is possible to provide, through existing seismic scaling laws of faults and earthquakes, reasonable assessments of the maximum possible earthquake magnitude over an entire area. The resulting (assessed) magnitudes (FLEM) are compared (i.e., the mathematical difference) with catalogued earthquake magnitudes that are the only existing points of reference against which assessed magnitudes can be presently compared."

The empirical relationships (used by many geoscientists including Valensise et alii) between fault size and earthquake magnitude are notoriously problematic for issues such as fault segmentation, fault continuity, and seismic partial activation of long faults. These problems have been ascertained ex-post on most single faults after earthquakes. We work on these same problems at the national scale with a different approach (ex-ante). That is, given a set of "official" faults of a nation (from geological maps and other official datasets), is it possible to test, quantify, and ascertain the above-mentioned problems connected with the empirical relationships? One way to do this, is to apply the relationship to the fault dataset and then compare the results with a seismic catalogue. This is what we do in Figs. 10, 11, 12, and 13 + Table 1.

The results of this work cannot be used for Seismic Hazard Assessment but rather, they can highlight areas where further studies are required to better asses expected earthquake magnitudes

In the revision we will better emphasize the above-mentioned concepts, starting from the abstract that in the previous version of the manuscript could have left some space for misinterpretation. We will also add disclaimers in Figures 7 and 8 saying that these figures are not at all for use for Civil Protection, officials, and, in general, for seismic hazard. (Lines 15-18 p. 1, 12-23 p. 3, + captions to Figs. 7 and 8).

**Comment 3:**

For instance, a very general statement such as "Larger earthquakes characterize the Apennines southern portion (Calabria), with historical seismic events that reached magnitudes up to 6.9-7.5" is backed by a reference to Cello et al. (2003), a 15 years-old paper dealing with a specific earthquake in Val d'Agri, 50 km north of Calabria, and to Gasparini et al. (1985), a 33 years-old paper that belongs to a distant past of seismotectonics in Italy. A simple reference to the Catalogo Parametrico dei Terremoti Italiani (CPTI), the Italian reference parametric catalogue, would have been enough; it would also have prevented a mistake, since no M 7.5 earthquake is reported anywhere in Italy

**Response 3:**

This criticism refers to a sentence extracted from the seismotectonic setting and not from the method and results sections. We will accept with pleasure the suggestion by Valensise et alii and will enrich the Seismotectonic Setting with some recent references in the revised version of our manuscript (Line 18 p. 5). However, we have to point out that the core and science of our paper is not at all affected by a supposed lack of consideration of previous works, as argued by the comment.

More importantly, in section 3.2 earthquake data, we do mention CPTI together with the most comprehensive catalogues of instrumental and historical seismicity like CSI1.1 and ISIDe.

**Comment 5:**

It is unfortunate that the fault database that should support this bold statement is not accessible (see Petricca, P., et al., Revised dataset of known faults in Italy, GFZ Data Services, https://doi.org/10.5880/fidgeo.2018.003, http://pmd.gfz-potsdam.de/panmetaworks/, 2018; the first link leads to an error page, while the second leads to a generic page of the GFZ website).

**Response 5:**

The database is fully and publicly available at the indicated database under this link that we will be report also on the revised manuscript (Lines 5 onward p. 17):

http://pmd.gfz-potsdam.de/panmetaworks/review/924b171fd21c78f295d58a7e9e321e8ad07667ab6201634b23d3cb5a3f170d10/

**Comment 8:**

A basic consideration is that by assembling faults from such different and nonhomogeneous sources, Petricca et al. inevitably put together a) alternative views on the same faults, possibly stemming from widely alternative conceptual models; b) faults that are mutually exclusive due to their geometry (typically, faults crossing each other in the subsurface: if one fault ends against another, its seismic potential based solely on length is largely overestimated); c) faults that cannot

be simultaneously active, or reactivated, in the current stress regime; and d) blind faults whose actual length may be strongly biased by the availability and density of subsurface data.

**Response 8:**

We think that this is the strength of our work. For the reasons explained above (Response 4) we propose that most of the faults within the Italian territory can host an earthquake. Therefore a comprehensive fault dataset, as the one used in the present work, can help in: 1) reducing bias induced by the availability and density of subsurface data; 2) highlighting areas where detailed future studies are required to improve seismic hazard, that is not the target of this work.

All this is clearly stated in the Introduction and in the Conclusions sections. However, we will better stress the above-mentioned concepts in the Abstract, Introduction, and Conclusions of the revised manuscript (Lines 15-18 p. 1, 12-23 p. 3, + captions to Figs. 7 and 8). This is a product for scientific use only (i.e. testing the empirical relationships).

**Comment 13:**

The authors never admit that their results are pointless, even when they remark (Page 12) that "... the negative occurrences are very limited...", i.e. that the number of predicted magnitudes that are larger than those observed in the historical record outnumbers by far the opposite case, resulting in the very asymmetric pattern shown in Figure 12. In fact, they refer to a "...limited difference... between FLEMs and the catalogued earthquake magnitudes..." (!), neglecting the obvious consideration that magnitude is a logarithmic quantity, implying that a 0.2 increase in Mw, for example from 6.0 to 6.2, doubles the seismic moment. For a typical continental fault having an aspect ratio in the range 2-3 and standard scaling for coseismic slip, doubling the moment implies that fault length may increase by over 20%. For a magnitude increase of 0.5, for example from 6.0 to 6.5, the seismic moment becomes 5.6 times larger, which may require a fault that is 100% or more longer than that necessary to generate the smaller earthquake.

**Response 13:**

Thanks. We accept this comment. We will rephrase this statement (the negative occurrences are very limited...). We agree that it is useless commenting this result as very limited. We will susbstitute "very limited" with the exact numbers that we obtain from the experiment. The term limited is too vague and subjective. However, once again, it is worth to underline that we are not proposing a forecast method, we are testing very well-known scaling laws at the national scale (Line 11 p. 13).

**Comment 16:**

The conclusions of this paper are worrisome, in consideration of the large number of areas where the authors envision the possibility of M 7.5 and larger earthquakes, that is to say earthquakes bigger than the largest magnitude ever recorded in Italy, without any consideration as to how frequently this may occur. In a standard PSHA approach these large magnitudes would be

assigned a very low probability of occurrence, leading to a minimal statistical impact on the expected ground shaking for short average return periods. The information about the possible largest earthquakes may generate a great deal of confusion if not appropriately communicated. We cannot imagine how the residents of Bologna, Ancona, Pescara, but also Padua, Trento, Vicenza and even Venice, cities lying in areas that are currently considered mid- to low-hazard, would react to knowing that very large earthquakes may occur below their feet at any moment.

**Response 16:**

We completely agree with the sentence "The information about the possible largest earthquakes may generate a great deal of confusion if not appropriately communicated". In fact, we are NOT proposing a new seismic hazard map, and this is really clear by reading the paper (not only looking at the figures). We simply compute potential earthquake magnitude from fault size and compare these results with seismic catalogs (Figs. 10-13 and Table1) to reason upon the validity of the scaling relationships between fault attributes and earthquake magnitude at the national scale with available fault datasets. The comment by Valensise et alii is therefore inappropriate.

However, in the revised manuscript we will better stress the above-mentioned concepts (i.e., our aim is not a new and reliable map of expected earthquakes in Italy). We will also add disclaimers in Figures 7 and 8 saying that these figures are not at all for use for Civil Protection, officials, and, in general, for seismic hazard. They are a product for scientific use only (i.e. testing the empirical relationships) (Lines 15-18 p. 1, 12-23 p. 3, + captions to Figs. 7 and 8).

**Comment 17:**

Another major flaw in the approach taken by Trippetta and co-workers lies in their discretisation of seismogenic zones into 25x25 km sub-areas. Of course, some discretisation is inevitable, but one has to be aware that a 25x25 km cell may host a 35 km-long fault, at the most. According to the equation proposed by Leonard (2010), the empirical law adopted by Trippetta and co-workers, a 35 km fault length corresponds to a Mw 6.8 earthquake. Hence, any larger earthquake will necessarily encompass two or more cells. A close inspection of Figures 7 and 8 of the paper, however, reveals that several cells filled in red or dark red, which according to the adopted colour-coding should correspond to an expected Mw in the range 7.4 to 7.8, occur isolated, i.e., surrounded by cells for which the expected FLEM is much smaller. According to same equation by Leonard (2010), this magnitude range corresponds to a fault length in the range 78 to 135 km, which should involve a minimum of 2 to 4 adjacent cells, depending on fault strike. An isolated cell capable of a Mw 7.4 earthquake is hence a seismological paradox that has no physical meaning, as the earthquake causative fault will necessarily extend to adjacent cells.

**Response 17:**

At which cell are Valensise et alii referring to? As we can see from figure 7 and 8 there are no red (M.7.5) isolated cells. In each cell, we consider the longest fault that touches/crosses the cell. This means that the faults can be longer than 35 km as indicated by Valensise et alii. This is clearly stated in our method section at Lines 27-28 Page 8: "The length of the longest fault crossing each cell determined the parameter "fault length" (Lf) of the considered cell."

Our computation is selfsustained and the complete database is available at the following link for reproducibility of our results (see also the Data availability section) (Lines 5 onward p. 17):

[revised manuscript text omitted]

**6**

~~Several national and international seismic hazard projects have been recently carried out for New Zealand (Stirling et al., 2002), Europe (Giardini et al., 2013; Woessner et al., 2015), USA (Petersen et al., 2008, 2014), the Middle East (Danciu et al., 2016), California (Field et al., 2009, 2014), Central Asia (Ullah et al., 2015), Italy (Cinti et al., 2004; Basili et al., 2008; DISS Working Group, 20 as well as several other studies on other countries. We here briefly compare some of these projects with our study to understand the main differences and similarities as well as the advantages and disadvantages.~~

 To this end, it is also noteworthy that studies on the

 2016 Amatrice-Norcia (central Italy) earthquakes (Mw 6.0 and 6.5)

revealed that the length of

 causative faults was only partially activated by the seismogenic slip (e.g., Chiaraluce et al., 2017); however, as this fault-slip behaviour seems rather frequent (Freymueller et al., 1994; Milliner et al., 2016; Chousianitis and Konca, 2018), it is most likely that this same behaviour is incorporated and implicitly expressed by the above-mentioned empirical scaling relationships between fault length and earthquake magnitude (e.g., Wells and Coppersmith, 1994; Leonard, 2010; Thingbaijam et al., 2017)

~~The method used by Danciu et al. (2016) for the Middle East derived from similar studies previously realized for Europe by Giardini et al. (2013), Woessner et al. (2015) (see also Giardini, 1999; Jiménez et al., 2001), aiming to assess the probabilistic seismic hazard of this continent on the basis of large tectonic and seismological datasets and advanced source models in a way similar to that previously explained for the study realized by Danciu et al. (2016).~~

**6  Statistical Test of FLEM Values**

 The reliability of estimated FLEM values may be quantitatively checked, in principle, using a formal statistical test on earthquake catalogues. Anyway, models of expected maximum magnitude (Mmax) are not readily testable. A statistical test helping to discriminate between competitive FLEM values is impossible in practice, even in the simplest case of discriminating between a double truncated Gutenberg-Richter (DTGR) law and an unbounded Gutenberg-Richter (UGR, Mmax=$\infty$) law on excellent datasets (Holschneider et al., 2014). Suffice is to say that the DTGR's probability distribution differs from the UGR's one by a constant $c = 1 - 10^{-b(M_{max} - M_c)}$ (where $M_c$ is the minimum magnitude). This value is very close to 1, if not when $M_c$ is close to Mmax. This means that the upper cut-off of the Gutenberg-Richter distribution can be explored using only large, very rare earthquakes, with magnitudes close to the maximum possible value, on which statistical inference is necessarily limited (Holschneider et al., 2014). Therefore, it is essentially impossible to statistically infer, with sufficient confidence, the

 maximum possible earthquake magnitude, in terms of alternative testing, from an earthquake catalogue alone (Holschneider et al., 20 In light of the limitations of purely statistical inference, geological and tectonic information provides, therefore, important and

5   exclusive constraints on the expected maximum magnitude. If we look at statistical testing in detail, we can check the FLEM values on a catalogue, while controlling the probability of wrongly rejecting them, but without reducing the probability of a wrong non-rejection (Holschneider et al., 2014). This makes it, indeed, impossible to discriminate between two likely values of FLEM. In other words, if we do not reject a FLEM value, we cannot say if it is true or if the data are inadequate in terms of

10    revealing its failure. Keeping all these considerations in mind, we test the estimated FLEM values on the CSI1.1 and ISIDe databases (available online at https:/

15    /csi.rm.ingv.it/ and http://iside.rm.ingv.it/, respectively). Since our analysis requests a database containing small magnitudes events, we consider ISIDe data of earthquakes occurred before April 16 2005, when the new INGV (Istituto Nazionale di

20   Geofisica e Vulcanologia) National Seismic Network, completely reorganized and equipped with a new acquisition system, became operational (Amato and Mele, 2008). Without this separation, the hypothesis of temporal homogeneity for magnitude data would not be appropriate. Moreover, most events in the ISIDe and CSI1.1 databases have a ML magnitude, while FLEM values are in moment magnitude scale. Therefore, we repeat the same testing procedure, described below, twice, without any difference in results: firstly, we consider original FLEM values and, secondly, we convert them in local magnitude scale through

25   the relations proposed by (Gasperini et al., 2013). We adopt the following test procedure.

   We select cells for which a test is helpful, i.e.

30   ~~exposed and buried faults, respectively). For the computation of the expected earthquake magnitude (PEMM), we simply relied on Leonard's scaling law (i.e. , L10; Leonard, 2010). Moreover, our aim, consisting of computing PEMM from fault length, is different from and considerably simpler than the aims of most of the aforementioned studies that point toward the calculation of seismic hazard, i.e. , the probability that an earthquake will occur in a given geographic area, within a given window of time, and with ground motion intensity exceeding a given threshold. However, thanks to its simplicity and rapidity, the method that we~~

35

information on the age of faults is poor. Bearing in mind the simplicity of our method, the observed correspondences (at least in many places) between PEMMs and the magnitudes from historically- and instrumentally-recorded earthquakes in Italy (Figs. 12a and 13c)are encouraging and suggests further validation for instance in New Zealand, USA, and other countries of Europe where the abundance of data and results (Petersen et al., 2008, 2014; Stirling et al., 2012; Giardini et al., 2013; Woessner et al., 2015) could allow a proper comparison and validation. In other words, our proposed method and results cannot replace existing estimates of time-dependent seismic hazard (e.g., Cinti et al., 2004; Basili et al., 2008; Petersen et al., 2008, 2014; Field et al., 2009, 2014; Stirling et a but rather can define the areas, i. e. those characterized by a large difference between PEMMs and catalogued earthquakes, where detailed studies are required. The main advantages of our method are the rapidity - once the faults are knownhaving a FLEM value above the related historical/mapped, vectorized, 
[revised manuscript text omitted]

Petersen, M., Moschetti, M., Powers, P., Mueller, C., Haller, K., Frankel, A., Zeng, Y., Rezaeian, S., Harmsen, S., Boyd, O., et al.: Documentation for the 2014 update of the United States national seismic hazard maps: U.S. Geological Survey Open-File Report 2014-1091, Tech.
30 rep., Geological Survey (US), https://doi.org/10.3 33/ofr20141091, 2014.

Petersen, M. D., Frankel, A. D., Harmsen, S. C., Mueller, C. S., Haller, K. M., Wheeler, R. L., Wesson, R. L., Zeng, Y., Boyd, O. S., Perkins, D. M., et al.: Documentation for the 2008 update of the United States national seismic hazard maps, Tech. rep., Geological Survey (US), 2008.

[revised manuscript text omitted]

Viola, C.: Appunti geologici ed idrologici sui dintorni di Teramo, Boll. del R. Com. Geol. d'Italia, 1, 221–228, 1893.

von Zittel, K. A.: Geologische Beobachtungen aus den Central-Apenninen, vol. 2, Oldenbourg, 1869.

Walsh III, F. R. and Zoback, M. D.: Probabilistic assessment of potential fault slip related to injection-induced earthquakes: Application to north-central Oklahoma, USA, Geology, 44, 991–994, https://doi.org/10.1130/G38275.1, 2016.

5  Wang, S., Xu, W., Xu, C., Yin, Z., Bürgmann, R., Liu, L., and Jiang, G.: Changes in groundwater level possibly encourage shallow earthquakes in central Australia: The 2016 Petermann Ranges earthquake, Geophysical Research Letters, https://doi.org/10.1029/2018GL080510, 2019.

Wells, D. L. and Coppersmith, K. J.: New empirical relationships among magnitude, rupture length, rupture width, rupture area, and surface displacement, Bulletin of the seismological Society of America, 84, 974–1002, 1994.

10  Wiemer, S. and Wyss, M.: Minimum magnitude of complete reporting in earthquake catalogs: examples from Alaska, the Western United States, and Japan, Bulletin of the Seismological Society of America, 90, 859–869, https://doi.org/10.1785/0119990114, 2000.

Woessner, J., Laurentiu, D., Giardini, D., Crowley, H., Cotton, F., Grünthal, G., Valensise, G., Arvidsson, R., Basili, R., Demircioglu, M. B., et al.: The 2013 European seismic hazard model: key components and results, Bulletin of Earthquake Engineering, 13, 3553–3596, https://doi.org/10.1007/s10518-015-9795-1, 2015.

[revised manuscript text omitted]

---

## Editor Decision (ED1)

Dear Andrea Billi,

Thank you for in most cases appropriately addressing the last comments. I found three occasions where your response was missing or not accurate, which I write below. I hence suggest technical corrections that need to address three points. Upon including these I do not need to check your manuscript again and it can proceed to the publication process.

I hope you will accept my apologies for the exceptionally long time needed to properly review and guide your paper. In part this was due to an exceptionally busy period upon return from my maternity leave and move to a new permanent position.

Best regards,

Ylona

*From that same perspective I think the worries expressed by Valensise as summarized in the centre paragraph on C5 (SC1) should be better articulated in the paper. For example, for assumptions 3 and 4 in section 4.2.*

*>> Not addressed (see start p. 2)*

*As a follow-up on comment 17 by Valensise:*
*I agree lengths in Fig. 7 and 8 seem to represent the required fault length and there are no isolated cells with too large a magnitude*
*However, in Fig. 9 for the observations I do see quite many of such isolated cells, as I guess these values have ended up only in the cell of the hypocenter. This means that when assessing the difference in Fig. 10,11, and 12 you are also including this missing information and are thus underestimating Mcatalogue and overestimating deltaM.*
Dear Editor, we have re-checked Figs. 7 and 8 and we do not see isolated cells characterized by high FLEM values (and therefore by long faults). In particular, we refer to red cells that are never isolated in Figs. 7 and 8. Please, note that cells with lower FLEM values (orange-yellow to blue cells) are characterized by shorter faults and hence can occur also as isolated. In other words, isolated orange-yellow to blue cells are compatible with the used method whereas isolated red cells would be incompatible (they are not isolated indeed) with our method as pointed out by Valensise et alii. We are obviously available for further improvements and clarifications.

*>> You have not addressed the key point, which is the second paragraph that refers to Fig. 9, not 7 or 8. I already wrote that I agree with your assessment of Fig. 7 and 8.*

*- Occasionally (p. 16, l. 15; p.1, l. 14) you still use scaling law instead of scaling relation.*
There are no "scaling laws" in our manuscript. We have found three "scaling equations" that have been changed into "scaling relationships" (Lines 16 P. 10; 22 P. 10; 28 P. 10).

*>> A simple search does show them at key locations in the manuscript. For example, see p.1, l.1 qand p. 16, l. 25.*

---

## Author Response (AR2)

**Comments and Responses**

*Comments:*

*You respond that your goal is not seismic hazard (response 1, reviewer 1), yet your abstract concludes with "Our work can provide a perspective time-independent seismic potential of faults" (l.16-18, p.1), which is a key ingredient of seismic hazard assessment. You claim - and your simplified method attest - that major implications for seismic hazard for Italy are not justified. I agree with that. So why are your last claims in abstract and conclusions about seismic potential relevant for seismic hazard assessment rather than about testing the usage of easily accessible fault databases on a national scale. I think such key pieces of formulation should be updated to reflect your goal better. I would suggest to not try to suggest implications beyond what the quality of your tests allows and remember the societal and political sensitivity such statements carry. Such worries are clearly stated by Valensise et al and reviewer 1, so please ensure that you remain concise and accurate about what your very simplified method can do and what it can not, particularly in the conclusions and implications you draw.*

Done. We have reworded the final part of Abstract (Lines 14-19 on Page 1) and Conclusions (Lines 5-6 P. 17) omitting concepts that may be somehow connected with the seismic hazard.
* * *
*You write that you "benchmark" scaling relationships against catalogued magnitudes (l. 3, p.1), but I do not think you reach such a high level related to a scientific interpretation of that word. I rather think wording like "compare" is more appropriate.*

Done. We have used "compare" instead of "benchmark". Please, see Line 3 P. 1.
* * *
*p. 6, l. 11: You write you consider "all known faults", while it is very clear from the long list of references from reviewer 1 and SC1-Valensise (p. C4-C5) that this is not the case. I think all descriptions of your fault database should be updated to reflect the actual situation accurately, while acknowledging other approaches and literature. For clarity, I do not think your database needs to be updated as you need to make assumptions to generate a database on some grounds. However, I think your text should be distinctly updated to acknowledge that you miss or miss-represent a distinct number of otherwise known faults. The worries of these experts should not be dismissed, but rather acknowledged appropriately.*

Done. We have better explained what we have excluded from our database. Please, see Lines 14-17 P. 6.
* * *
*From that same perspective I think the worries expressed by Valensise as summarized in the centre paragraph on C5 (SC1) should be better articulated in the paper. For example, for assumptions 3 and 4 in section 4.2.*

*Reviewer 1, Comment 6*

*I think it is fair to answer to the critique of missing faults with careful examples given by the reviewer. I understand your simplified method resolves the double counting aspect, but still there are two examples well total length may be underestimated. Addressing this limitation in the paper also gives the reader a better perception of the completeness of the databases you use. Here just saying they are as complete as possible is not really meaningful. Rather write exactly what you do (as you do), but than acknowledge potential though understandable flaws properly.*

Dear Editor, the limits connected with our methods are largely addressed (Pages 9-11) and have now been improved following your comments (please, see Lines 12-23 P. 11). Valensise et alii mention two earthquakes and related faults.

The Messina 1908 earthquake possibly occurred offshore (Messina Straits) in a region of high erosion and deposition due to steep sides and related slides and turbidites. The causative fault has never been found with certainty (Billi et al., 2008, GRL, doi: 10.1029/2008GL033251) and hence I do really think that this is not a good case to mention to emphasize/explain our limits. Moreover, the magnitude of this earthquake is an assessment.

Concerning the 1915 Avezzano earthquake, we would like to stress (1) that the related magnitude 7.0 is an estimate based on damages and not an instrumental measure, and (2) that the causative fault in our database (the Fucino Fault) is 15.85 km long, corresponding to a FLEM (Mw) of 6.25 (according to Leonard, 2010). This second case has now been properly mentioned in the manuscript as you suggest (Lines 17-23 P. 11).

The double counting problem is addressed at Lines 5-10 P. 9 and the completeness of our fault database is now addressed/improved (following your comments) at Lines 14-17 P. 6.
* * *
*Your analysis does not account for rupture jumps that could happen from one fault to the next, as suggested by the "segmentation" examples of e.g. 2016 M7.8 Kaikoura earthquake. That would mean that earthquake with larger magnitudes than you estimated could occur. This is quite a likely scenario with major implications if that is too happen. However, you describe this limitation in only line (p. 11, l. 4) without any scientific references or appreciation of the process or consequences. I think that should be included.*

Done. We have better explained this problem and mentioned the related paper (Cesca et al., 2017). Please, see Lines 12-17 P. 11 and the added reference to Cesca et al. 2017 work on EPSL. Moreover, also the Fucino case (see our previous response) is a case of multiple coseismic rupture. Also this concept has been now integrated in the new version of our manuscript (Lines 20-22 P. 11).
* * *
*I also very much appreciate you openly sharing the updated Italian fault database. However, I am also not able to find and download the fault database from the link given in the manuscript and rebuttal letter. Could you please ensure a long lasting and secured access?*

*http://pmd.gfzpotsdam.*

*de/panmetaworks/review/924b171fd21c78f295d58a7e9e321e8ad07667ab6201634b23d3*

*cb5a3f170d10/*

We understand the concern. The database is now open at this link:

http://pmd.gfz-potsdam.de/panmetaworks/review/924b171fd21c78f295d58a7e9e321e8ad07667ab6201634b23d3cb5a3f170d10/

Moreover, consistently with the policy of the Potsdam (GFZ) Repository, this database will be freely accessible through the DOI and related reference (Petricca et al., 2018) as soon as the present paper will be officially published on Solid Earth. This is the policy of the Potsdam (GFZ) Repository. Please, note that we have used the same repository and policy for a recent paper published on Solid Earth (https://www.solid-earth.net/10/741/2019/).
* * *
*I agree with Reviewer 2 (comment 10) and Valensise et al (comment 13) that using log-scale quantities such as magnitude introduces a bias and potential for misinterpretation of the size difference (see values in comment 13).*

*You agree with this comment, yet I do not see the suggested changes in the updated manuscript (l. 11, p. 13). I do not see exact numbers.*

*However, I find it more important that major differences often of 2 magnitude values (Fig. 10) are described as "having a good agreement ..." (p. l. ), such that "this gives credibility to the used scaling relations" (p.1, l.16) or that these "can be benchmarked" (p. 1, l. 5). Based on the information in your abstract a 2 standard deviation range includes a magnitude range of 2\*1.47 is almost 3 magnitude values, which require very large fault length differences. I think this rather shows large discrepancies, which could also be used to argue that something opposed. I have the impression you take care of this uncertainty by in the discussion saying that they show "either good agreement or some differences are observed". I do not think that is very accurate. Could you please improve your formulation in such conclusions, abstracts and in the results?*

We understand the Editor point and, consistently, we have omitted all adjective and adverbs providing a qualitative and subjective opinion on our results. Please, see our changes at Lines 14-18 P. 1; 22 P. 13; 31 P. 13; 1 P. 14; 8 P. 14; 27-28 P. 16; 2 P. 17.
* * *
*I also like the suggestions to you seismic moment instead, which can indeed not be calculate directly for all earthquakes, but scaling relations similar to those used also exist to portray seismic moment. Doing that would better show the variation in values and would lead to less bias.*

Although we understand the validity of this suggestion, its feasibility is impossible in our case due to the fact that, for many earthquakes (particularly in historical catalogs but also in old instrumental catalogs), independent assessments of the seismic moments do not exist. This makes impossible the use of seismic moments in our approach, which includes historical and instrumental earthquake catalogs for an entire nation. In other words, the only parameter available for the entire nation over the historical and instrumental periods is the earthquake magnitude and not the seismic moment.
* * *
*In response to Valensise (SC1, p. C7-C9 & comment 16) I think you should also stress more the low probability of occurrence for such large earthquakes. Common people that do not in detail read the whole paper should readily understand this low probability to prevent confusion and potential follow-up mis communication at all levels. The emphasis on such information should thus be done in the main text at multiple key locations and by extending the disclaimer with more information than this being for scientific purposes only (e.g., the missing component of probability of occurrence and risk that is missing in all such figures). Considering the sensitivity of this topic for society I think such additions are useful and required.*

Done. We have added a statement at the end of Intro and in the Acknowledgments. Please, see Lines 22-25 P. 3; 22-27 P. 17.
* * *
*Your new section structure makes it hard for the reader to understand your flow and appreciate what you did. Section 5 describes "Results and discussion" all together in one section. Section 6 then describes a "Statistical test of FLEM values", which is again an analysis of the results. Could you update your structure and section headings a more standard scientific paper setup to facilitate the reader in finding the information needed?*

Done. Please, see Lines 1 P. 12; 23 P. 14.
* * *
*As a follow-up on comment 17 by Valensise:*

*I agree lengths in Fig. 7 and 8 seem to represent the required fault length and there are no isolated cells with too large a magnitude*

*However, in Fig. 9 for the observations I do see quite many of such isolated cells, as I guess these values have ended up only in the cell of the hypocenter. This means that when assessing the difference in Fig. 10,11, and 12 you are also including this missing information and are thus underestimating Mcatalogue and overestimating deltaM.*

Dear Editor, we have re-checked Figs. 7 and 8 and we do not see isolated cells characterized by high FLEM values (and therefore by long faults). In particular, we refer to red cells that are never isolated in Figs. 7 and 8. Please, note that cells with lower FLEM values (orange-yellow to blue cells) are characterized by shorter faults and hence can occur also as isolated. In other words, isolated orange-yellow to blue cells are compatible with the used method whereas isolated red cells would be incompatible (they are not isolated indeed) with our method as pointed out by Valensise et alii. We are obviously available for further improvements and clarifications.
* * *
*- Your sentences can be quite long (e.g. p. 14 l.2-7). I recommend another look at the sentence length and language of the manuscript.*

We have polished a bit the English over the entire manuscript and tried to split all long sentences.
* * *
*- Occasionally (p. 16, l. 15; p.1, l. 14) you still use scaling law instead of scaling relation.*

There are no "scaling laws" in our manuscript. We have found three "scaling equations" that have been changed into "scaling relationships" (Lines 16 P. 10; 22 P. 10; 28 P. 10).

Thanks a lot for your efforts on our manuscript

Sincerely

Andrea Billi and co-authors

---

## Author Response (AR3)

Rome, July 26th, 2019

Dear Editor,

as already discussed with you via email, the following are our changes and responses to your raised points:

(1) Laws vs Relationhips:

We have changed all "laws" into "relationships".

(2) Fig. 9:

Now we understand what you mean. We have included a new statement (page 13, lines 28-31) acknowledging the fact that historical and instrumental earthquakes are reported as punctual epicenters (Fig. 9) even though they were generated by faults that are commonly kilometers-long surfaces. This can lead, in places, to an overestimation of deltaM in Figs 10-12.

(3) Worries expressed by Valensise:

We have now introduced a new statement (see below) at page 11 lines 21-24:

"More generally, using surface faults/ruptures to infer earthquake parameters (at seismogenic depths) may lead to a misestimation. This concept has already been addressed in the previous literature dealing with scaling relationships between fault size and earthquake magnitude (e.g., Wells and Coppersmith, 1994; Leonard, 2010; Thingbaijam et al., 2017)."

Thanks a lot for your assistance

Sincerely

Andrea Billi and co-authors